# Prompt Tuning Transformers for Data Memorization

**Haiyu Wang**
Department of Statistics and Data Science
The Chinese University of Hong Kong
HaiYuWang@link.cuhk.edu.hk

**Yuanyuan Lin**[*]
Department of Statistics and Data Science
The Chinese University of Hong Kong
ylin@sta.cuhk.edu.hk

## Abstract

Prompt tuning has emerged as a powerful parameter-efficient fine-tuning technique, allowing large pretrained Transformers to adapt to downstream tasks by optimizing a small set of prompt embeddings. Despite its empirical success, the extent to which prompt tuning can memorize data remains poorly understood. In this paper, we provide both theoretical and empirical analyses of data memorization ability of prompt-tuned Transformers. Building on recent theoretical frameworks, we derive an upper bound on the required prompt length for exact memorization of finite datasets and establish a trade-off between prompt length and the number of autoregressive generation steps. Specifically, we show that a constant-size Transformer can memorize $n$ input-output pairs with prompts of length $\tilde{O}(\sqrt{nN})$, where $N$ denotes the sequence length. Empirical results further demonstrate that prompt-tuned, randomly initialized Transformers are able to effectively memorize finite datasets. These models also capture the intrinsic low-rank structure of the data, leading to a reduction in the required prompt length. Finally, we analyze how the initialization of the Transformer backbone affects the performance of prompt tuning. Our findings provide new insights into the expressivity, efficiency, and underlying mechanisms of prompt tuning, bridging theoretical memorization limits with observed empirical behaviors.

## 1 Introduction

Large pre-trained Transformer models have become the cornerstone of modern artificial intelligence, exhibiting remarkable capabilities in language understanding, reasoning, and transfer learning [Devlin et al., 2019, Dosovitskiy et al., 2020, Jumper et al., 2021, Raffel et al., 2020, Liu et al., 2021b, Ramesh et al., 2021, Tay et al., 2022, Saharia et al., 2022, Villegas et al., 2022, Peebles and Xie, 2023, Han et al., 2022, Hadi et al., 2023, Naveed et al., 2023, Ji et al., 2025]. To make such massive models more adaptable to downstream applications, parameter-efficient fine-tuning (PEFT) techniques have been proposed as practical alternatives to full fine-tuning, substantially reducing the number of trainable parameters while preserving competitive performance [Hu et al., 2022, Liu et al., 2021a, Lester et al., 2021]. Among these approaches, prompt tuning stands out for its simplicity and generality: it freezes the pretrained backbone while learning only a small set of task-specific prompt embeddings, enabling lightweight yet effective adaptation across diverse domains [Zhou et al., 2022, Ge et al., 2023, Jia et al., 2022, Fang et al., 2023].

**Memorization or generalization?** Despite their impressive empirical success, large language models have reignited a long-standing debate: do they genuinely generalize to unseen inputs, or do they predominantly memorize patterns and examples from pretraining or finetuning corpora? Empirical evidence indicates that LLMs often exhibit a blurred boundary between memorization and generalization—memorizing factual or low-complexity patterns while generalizing in reasoning-intensive or compositional tasks [Wang et al., 2024, Hartmann et al., 2023]. This duality challenges

---

[*]Corresponding Author

our conventional understanding of what it means for a neural network to understand data, raising fundamental questions about the nature of knowledge stored within LLMs. Intuitively, parameter-efficient fine-tuning methods, which updates only a limited number of parameters within otherwise fixed architectures, are expected to possess weaker memorization capacity than fully fine-tuned models. With most of the representational power locked in the frozen backbone, such methods may struggle to encode task-specific information solely through prompt embeddings. This observation naturally leads to a fundamental question: to what extent can prompt tuning memorize data when the underlying Transformer remains fixed?

**Expressivity or vulnerability?** Recent studies have suggested that data memorization should not merely be interpreted as a byproduct of training, but rather as a quantitative indicator of a model's expressive ability. Allen-Zhu and Li [2024] observed that fully trained Transformers can encode roughly two bits of input information per parameter, while theoretical analyses further link this memorization ability to the model's functional richness and representational power [Kim et al., 2022, Mahdavi et al., 2023, Kajitsuka and Sato, 2024]. However, the same ability to store detailed information internally also introduces potential vulnerabilities: memorized data can be extracted via carefully crafted or learned prompts, risking the exposure of sensitive or proprietary information contained in the training corpus [Carlini et al., 2022, Ozdayi et al., 2023]. Therefore, understanding the mechanisms and implications of memorization is crucial not only for characterizing the expressive power of large language models, but also for assessing the privacy risks they inherently pose.

Motivated by these considerations regarding the practical significance of data memorization, and inspired by recent theoretical advances [Wang et al., 2023, Petrov et al., 2023, Hu et al., 2024, Petrov et al., 2024, Nakada et al., 2025], we investigate the expressive ability of prompt tuning in memorizing finite datasets with zero error. Moreover, to disentangle the architectural expressivity of the Transformers from the effects of pretraining, we investigate prompt tuning applied to randomly initialized Transformers. This setting allows us to examine whether prompt embeddings alone can enable memorization when the backbone model provides no task-specific prior knowledge. Our main contributions are summarized as follows:

**Main Contributions** Our main contributions are summarized as follows:

- Building upon the recent theoretical framework in [Nakada et al., 2025], which demonstrated that prompt tuning a Transformer can exactly implement a ReLU feed-forward neural network, we derive the first upper bound on required the prompt length to memorize finite datasets. Specifically, we prove that prompt tuning a constant-size Transformer with the prompt length of $\tilde{O}(\sqrt{nN})$ can memorize $n$ input sequences of length $N$.

- We further establish a theoretical trade-off between the prompt length and number of the intermediate steps during autoregressive generation. When $N = 1$, we show that prompts of length $O(n)$ with $\tilde{O}(1)$ intermediate steps can memorize $n$ input-output pairs, while the same memorization capacity can be achieved with shorter prompts of length $\tilde{O}(\sqrt{n})$ if $\tilde{O}(\sqrt{n})$ intermediate steps are allowed.

- Our experiments demonstrate that prompt-tuned, randomly initialized Transformers can effectively memorize finite datasets, especially when the word embeddings exhibit structural regularity. Moreover, these models are able to capture the intrinsic low-rank structure of the data, which in turn enables a reduction in the required prompt length. Finally, we analyze how the initialization of the Transformer backbone influences the overall performance of prompt tuning.

## 1.1 Related Works

**Theoretical Understanding of Data Memorization** One of the fundamental study on the expressive capacity of Transformers is to see whether Transformers can achieve zero loss on finite input-label pairs, which is named the data memorization ability. Kim et al. [2022] is the first work to study the data memorization ability of Transformers. They proved that Transformers with $\tilde{O}(d + n + \sqrt{nN})$ parameters are able to memorize $N$ sequences of $d$-dimensional tokens with length $n$. Mahdavi et al. [2023] showed that a multi-head self-attention mechanism with $H$ heads

and $O(Hd^2)$ parameters is capable of memorizing $O(Hn)$ data samples. Recently, Kajitsuka and Sato [2023] proved that Transformers with only one single-head self-attention layer possess data memorization ability. However, the required parameters in their work have to be $O(dnM + d^2)$. Chen and Zou [2024] built the results of Transformers with ReLU activation function in the self-attention layers under the assumption each data label is distinct. Kajitsuka and Sato [2024] made an effort to derive the optimal number of parameters needed to memorize given data points. It was shown that Transformers with $\tilde{O}(\sqrt{N})$ parameters can memorize $N$ input sequences of length $n$ in the next-token prediction task and $\tilde{O}(\sqrt{nN})$ parameters in the next-token prediction setting. They further proved that $\tilde{O}(\sqrt{N})$ parameters in the next-token prediction is optimal up to logarithmic factors. Dana et al. [2024] proved that a one-layer attention-only Transformer with $H$ heads each of dimension $d_h$ can memorize $Hd_h + d$ associations.

**Theoretical Understanding of Prompt Tuning** Proposed by [Lester et al., 2021], prompt tuning has emerged as a promising parameter-efficient fine-tuning approach. Wang et al. [2023] analyzed prompt tuning from the lens of universal approximation ability and limitations with finite-depth fixed-weight pretrained Transformers. Their results showed that Transformers with prompts are able to approximate any sequence-to-sequence Lipschitz function. Besides, they constructed a dataset, that cannot be memorized by a single-layer Transformer with prompts of any length. Hu et al. [2024] investigated expressive power of a single-layer and single-head Transformer, showing that prompt tuning is universal approximator and explicitly deriving the lower bound on the length of required prompt tokens. Petrov et al. [2023] formally showed that soft prompt tuning and prefix tuning are more expressive than prompting that only operates in the discrete token space. Besides, they demonstrated that prompt tuning suffers from some structural limitations that hold back the Transformers from forming task-specific attention patterns. In their another work [Petrov et al., 2024], Transformers with prompts were proved to be able to universally approximate continuous functions defined on the hypersphere. However, the number of trainable parameters needed is larger than training from scratch, which showed that prompt tuning might be less efficient. Oymak et al. [2023] developed new statistical foundations for gradient-based prompt tuning, characterized its optimization and generalization dynamics, and explored how it may facilitate attending to context-relevant information. Qiu et al. [2024] showed there exists a finite-size Transformer such that for any computable function, there exists a corresponding prompt following which the Transformer computes the function, meaning that prompt tuning is Turing-complete. Our work is mainly built upon the framework proposed by Nakada et al. [2025], in which they showed that prompt tuning a constant-size Transformer can exactly implement a ReLU feed-forward neural network with the prompt length being determined by the rank of weight matrices in each layer.

## 2 Preliminaries

### 2.1 Transformers

In this section, we formalize some basic concepts about Transformer architecture, which was first proposed by [Vaswani et al., 2017]. In general, Transformers are defined by stacking multiple Transformer layers, each of which consists of a self-attention layer and a feed-forward layer.

**Self-attention Layer:** The core of Transformer architecture is the self-attention mechanism, which mixes information across different positions in the input sequences. The pair-wise dot-product and activation function determine how much focus each token should have on others in a sequence. To be specific, an self-attention layer $\mathcal{F}_{SA} : \mathbb{R}^{D_{in} \times N} \to \mathbb{R}^{D_{in} \times N}$ with $H$ heads is defined as

$$\mathcal{F}_{SA}(\boldsymbol{X}) := \boldsymbol{X} + \sum_{i=1}^{H} \boldsymbol{W}_V^{(i)} \boldsymbol{X} \sigma \left[ (\boldsymbol{W}_K^{(i)} \boldsymbol{X})^\top (\boldsymbol{W}_Q^{(i)} \boldsymbol{X}) \right] \in \mathbb{R}^{D_{in} \times N}, \tag{2.1}$$

where $D_{in} \in \mathbb{N}^+$ is the embedding size, $\boldsymbol{W}_V^{(i)}, \boldsymbol{W}_Q^{(i)}, \boldsymbol{W}_K^{(i)} \in \mathbb{R}^{D_{in} \times D_{in}}$ are the weight matrices, $\sigma : \mathbb{R} \to \mathbb{R}$ is the activation function in the self-attention layer. The self-attention layer computes a weighted average of linearly transformed token representations, with the attention weights determined by the similarity between query and key projections.

**Feed-forward layer:** Self-attention layers deal with the interactions between different tokens, while feed-forward layers provide a non-linear, token-wise transformation, which processes each token

independently. A feed-forward layer $\mathcal{F}_{FF} : \mathbb{R}^{D_{in} \times N} \to \mathbb{R}^{D_{in} \times N}$ is given by

$$\mathcal{F}_{FF}(\boldsymbol{X}) := \boldsymbol{X} + \boldsymbol{W}_2(\sigma'(\boldsymbol{W}_1 \boldsymbol{X})), \tag{2.2}$$

where $D_{hid}$ is the hidden dimension, chosen to be $O(D_{in})$, $\boldsymbol{W}_1 \in \mathbb{R}^{D_{hid} \times D_{in}}$, $\boldsymbol{W}_2 \in \mathbb{R}^{D_{in} \times D_{hid}}$, and $\sigma'$ is the element-wise activation function in the feed-forward layer.

**Remark 2.1.** Let $\sigma_S$ denote the Softmax function (i.e., $[\sigma_S(\boldsymbol{x})]_i = e^{\boldsymbol{x}_i} / \sum_{i=1}^d e^{\boldsymbol{x}_i}$, $i \in [d]$, for any $\boldsymbol{x} \in \mathbb{R}^d$), and $\sigma_R$ denote the ReLU function (i.e., $[\sigma_R(\boldsymbol{x})]_i = \max\{\boldsymbol{x}_i, 0\}$, $i \in [d]$, for any $\boldsymbol{x} \in \mathbb{R}^d$). In existing literature, $\sigma'$ is usually chosen to be ReLU function while $\sigma$ has many distinct options. Yun et al. [2019] and Gu et al. [2021] used Hardmax function for mathematical simplicity; Kajitsuka and Sato [2023] utilized Softmax function and proved that a single self-attention layer with Hardmax function may not be powerful. Recently, Jiao et al. [2025a] proposed to utilize different $\sigma'$, which may help to break the curse of dimensionality. In this paper, we set $\sigma$ and $\sigma'$ both to be element-wise ReLU function. See discussion in Appendix D.5.

Based on the definitions of the self-attention and feed-forward layers, the class of Transformer neural networks can be formally defined as:

$$\mathcal{T}(D_{in}, D_{out}, D_{hid}, H, L) := \left\{ \boldsymbol{T} : \boldsymbol{T} = \mathcal{E}_{out} \circ \mathcal{F}_{FF}^{(L)} \circ \mathcal{F}_{SA}^{(L)} \circ \cdots \circ \mathcal{F}_{FF}^{(1)} \circ \mathcal{F}_{SA}^{(1)} \circ \mathcal{E}_{in} \right\},$$

where $D_{in}$ is the embedding size, $D_{out}$ is the output size, $D_{hid}$ is the hidden dimension, $H$ is the number of heads, $L$ is the number of Transformer layers, each consisting of a self-attention layer and a feed-forward layer. $\mathcal{E}_{out}$ and $\mathcal{E}_{in}$ are two linear affine functions, which are designed to truncate the output and embed the input, respectively. For any sequential data point $\boldsymbol{X} \in \mathbb{R}^{d \times N}$, we have $\mathcal{E}_{in}(\boldsymbol{X}) \in \mathbb{R}^{D_{in} \times N}$, where the embedding size satisfies $D_{in} = O(d)$. If we let the output dimension $D_{out} \leq D_{in}$, then we have $\mathcal{E}_{out}(\boldsymbol{Y}) = \boldsymbol{Y}_{:D_{out},:} \in \mathbb{R}^{D_{out} \times N}$ for any $\boldsymbol{Y} \in \mathbb{R}^{D_{in} \times N}$.

## 2.2 Feed-Forward Neural Networks

We denote $\mathcal{FF}(W, L, \mathbb{R}^d \to \mathbb{R}^{d'})$ as the set of vector-valued functions $\phi : \mathbb{R}^d \to \mathbb{R}^{d'}$ that can be represented by a feed-forward neural network (FFN) with width $\leq W \in \mathbb{N}^+$, depth $\leq L \in \mathbb{N}^+$, and activation function is $\sigma_R$. The width of a FFN refers to the maximum number of neurons in the hidden layers and the depth corresponds to the number of hidden layers. For instance, suppose $\phi : \mathbb{R}^d \to \mathbb{R}^{d'}$ is a vector-valued function realized by a feed-forward neural network activated by $\sigma_R$. Then, $\phi$ can be expressed as

$$\phi = \mathcal{L}_L \circ \sigma_R \circ \mathcal{L}_{L-1} \circ \cdots \circ \sigma_R \circ \mathcal{L}_1 \circ \sigma_R \circ \mathcal{L}_0,$$

where each $\mathcal{L}_\ell$ is an affine linear map given by $\mathcal{L}_\ell(\boldsymbol{x}) := \boldsymbol{W}_\ell \boldsymbol{x} + \boldsymbol{b}_\ell$ for $\ell = 0, 1 \cdots, L$. Here, $\boldsymbol{W}_\ell \in \mathbb{R}^{d_{\ell+1} \times d_\ell}$ and $\boldsymbol{b}_\ell \in \mathbb{R}^{d_\ell+1}$ are the weight matrix and bias term, respectively, with $d_0 = d$, $d_1, \cdots, d_L \in \mathbb{N}^+$, and $d_{L+1} = d'$. Clearly, $\phi \in \mathcal{FF}(W, L, \mathbb{R}^d \to \mathbb{R}^{d'})$, where $W = \max\{d_1, \cdots, d_L\}$.

## 2.3 Autoregressive Generation

Given a Transformer $\boldsymbol{T} \in \mathcal{T}(D_{in}, D_{out}, D_{hid}, H, T)$ for some $D_{in}, D_{out}, D_{hid}, H, T \in \mathbb{N}^+$, which has the following form

$$\boldsymbol{T} = \mathcal{E}_{out} \circ \mathcal{F}_{FF}^{(L)} \circ \mathcal{F}_{SA}^{(L)} \circ \cdots \circ \mathcal{F}_{FF}^{(1)} \circ \mathcal{F}_{SA}^{(1)} \circ \mathcal{E}_{in}.$$

We let $\boldsymbol{g}$ denote the main part of $\boldsymbol{T}$, that is

$$\boldsymbol{g} = \mathcal{F}_{FF}^{(L)} \circ \mathcal{F}_{SA}^{(L)} \circ \cdots \circ \mathcal{F}_{FF}^{(1)} \circ \mathcal{F}_{SA}^{(1)}.$$

For any sequential data $\boldsymbol{X} \in \mathbb{R}^{d \times N}$, and prompts $\boldsymbol{P} \in \mathbb{R}^{D_{in} \times M}$. Denote $\boldsymbol{X}_{emb} = [\boldsymbol{P}, \mathcal{E}_{in}(\boldsymbol{X})] \in \mathbb{R}^{D_{in} \times (M+N)}$. In Autoregressive generation, the last token of the output will be prepended to the original sequence to form the new input. To be specific, suppose that $\boldsymbol{X}_{emb} = [\boldsymbol{x}_1, \cdots, \boldsymbol{x}_M, \boldsymbol{x}_{M+1}, \cdots, \boldsymbol{x}_{M+N}]$, where $\boldsymbol{x}_i = \boldsymbol{P}_{:,i} \in \mathbb{R}^{D_{in}}$ for any $i \in [M]$, and $\boldsymbol{x}_i = \mathcal{E}_{in}(\boldsymbol{X})_{:,i-M}$ for any $i \in \{M+1, \cdots, N+M\}$. Given the generation step $K \in \mathbb{N}^+$, $\boldsymbol{g}$ iteratively generates a sequence of tokens $\boldsymbol{x}_{M+N+1}, \cdots, \boldsymbol{x}_{M+N+t}, \boldsymbol{x}_{M+N+t+1}, \cdots, \boldsymbol{x}_{M+N+K}$ and we have $\boldsymbol{x}_{N+t+1} = \boldsymbol{g}([\boldsymbol{x}_1, \cdots, \boldsymbol{x}_{N+t}])_{:,-1}$. Following the techniques in [Nakada et al., 2025], we need to deal with a certain piece of the generated sequence, which motivates us to give the following definition.

**Definition 2.1.** Given a Transformer $\boldsymbol{T} \in \mathcal{T}(D_{in}, D_{out}, D_{hid}, H, L)$ for some $D_{in}, D_{out}, D_{hid}, H, L \in \mathbb{N}^+$, and the autoregressive generation step $K \in \mathbb{N}^+$. $\boldsymbol{T}$ can be written as $\boldsymbol{T} = \mathcal{E}_{out} \circ \boldsymbol{g} \circ \mathcal{E}_{in}$ with $\boldsymbol{g} = \mathcal{F}_{FF}^{(L)} \circ \mathcal{F}_{SA}^{(L)} \circ \cdots \circ \mathcal{F}_{FF}^{(1)} \circ \mathcal{F}_{SA}^{(1)}$. Given input sequence $\boldsymbol{X} \in \mathbb{R}^{d \times N}$ and prompts $\boldsymbol{P} \in \mathbb{R}^{D_{in} \times M}$ with length $M \in \mathbb{N}^+$, $\boldsymbol{T}$ sequentially generate $K$ tokens according to the autoregressive algorithm, denoted by $\boldsymbol{x}_{M+N+1}, \cdots, \boldsymbol{x}_{M+N+K}$, which satisfy $\boldsymbol{x}_{N+M+i} = \boldsymbol{g}([\boldsymbol{x}_1, \cdots, \boldsymbol{x}_{N+M+i-1}])_{:,-1}$ for $i \in [K]$. Then, for any $K_1, K_2 \in \mathbb{N}^+$ with $1 \leq K_1 \leq K_2 \leq K$, we let $\hat{\boldsymbol{T}}_{\boldsymbol{P}, K_1, K_2}(\boldsymbol{X})$ denote the piece from the $K_1$-th token to the $K_2$-th token of the generated sequence, that is

$$\hat{\boldsymbol{T}}_{\boldsymbol{P}, K_1, K_2}(\boldsymbol{X}) := \mathcal{E}_{out}([\boldsymbol{x}_{M+N+K_1}, \cdots, \boldsymbol{x}_{M+N+K_2}]) \in \mathbb{R}^{D_{out} \times (K_2+1-K_1)}.$$

In particular, we let $\hat{\boldsymbol{T}}_{\boldsymbol{P}, 0, 0}(\boldsymbol{X})$ denote $[\mathcal{E}_{out} \circ \boldsymbol{g} \circ (\boldsymbol{P} + \mathcal{E}_{in}(\boldsymbol{X}))]_{:, M+1:}$.

## 2.4 Prompt Tuning

Prompt tuning aims to design or learn suitable prompts that can guide Transformers to generate desired output. Let $\boldsymbol{X} \in \mathbb{R}^{d \times N}$ be a sequential input. Each column of $\boldsymbol{X}$, that is, $\boldsymbol{X}_{:,i}$, for any $i \in [N]$, is called the $i$-th token of $\boldsymbol{X}$. Let $\boldsymbol{y} \in \mathbb{R}^{D_{out} \times N}$ be the label of $\boldsymbol{X}$. $[\cdot, \cdot]$ represents the horizontal concatenation of matrices. In the following, we give the definition of prompt tuning.

**Definition 2.2** (Prompt Tuning). Let Transformer $\boldsymbol{T} \in \mathcal{T}(D_{in}, D_{out}, D_{hid}, H, L)$ for some $D_{in}, D_{out}, D_{hid}, H, L \in \mathbb{N}^+$ be a pretrained model with frozen parameters. For any downstream task with finetuning dataset $\boldsymbol{S} = \{(\boldsymbol{X}^{(1)}, \boldsymbol{Y}^{(1)}), \cdots, (\boldsymbol{X}^{(n)}, \boldsymbol{Y}^{(n)})\} \subset \mathbb{R}^{d \times N} \times \mathbb{R}^{D_{out} \times N}$, and prompt $\boldsymbol{P} \in \mathbb{R}^{D_{in} \times M}$ of length $M \in \mathbb{N}$, the output of $\boldsymbol{T}$ given input $\boldsymbol{X}^{(i)}$ with prompt $\boldsymbol{P}$ and autoregressive algorithm is defined as

$$\hat{\boldsymbol{Y}}^{(i)} = \hat{\boldsymbol{T}}_{\boldsymbol{P}, K_1, K_2}(\boldsymbol{X}^{(i)}),$$

where $K_2 + 1 - K_1 = N$. Let $\ell : \mathbb{R}^{D_{out} \times N} \times \mathbb{R}^{D_{out} \times N} \to \mathbb{R}_{\geq 0}$ be the loss function. The goal of prompt tuning is to find an optimal prompt $\boldsymbol{P}^*$ such that

$$\boldsymbol{P}^* \in \underset{\boldsymbol{P} \in \mathbb{R}^{D_{in} \times M}, M \in \mathbb{N}}{\arg\min} \sum_{i=1}^{n} \ell\left(\hat{\boldsymbol{Y}}_{:, M+1, :}^{(i)}, \boldsymbol{Y}^{(i)}\right).$$

Technically, we do not approach the above problem above from an optimization perspective. Instead, we directly prove the existence of suitable prompts that enable a certain pretrained Transformer to achieve zero loss on any given finite dataset satisfying some assumptions. Note that prompt tuning only prepends prompt tokens to the original inputs, whereas prefix-tuning inserts such prompt tokens to the input of each self-attention layers (See details in [Petrov et al., 2023]). In this work, we do not distinguish between the two.

# 3 Prompt Tuning Transformers for Data Memorization

In this section, we state the main results of this paper regarding the memorization capacity of Transformers with prompts. Informally, memorization capacity refers to the possibility that models can achieve zero loss on a specific number of arbitrary data points. To be more precise, given $n$ input-label pairs $(\boldsymbol{X}^{(1)}, \boldsymbol{y}^{(1)}), \cdots, (\boldsymbol{X}^{(n)}, \boldsymbol{y}^{(n)}) \subset \mathbb{R}^{d \times N} \times [C]^{1 \times N}$, where $C \in \mathbb{N}^+$ is a constant, representing the vocabulary size. We are interested in constructing an autoregressive Transformer $\boldsymbol{T} : \mathbb{R}^{d \times N} \to [C]^{1 \times N}$ with prompt $\boldsymbol{P}$, and generation step $K_1, K_2$ such that $\hat{\boldsymbol{T}}_{\boldsymbol{P}, K_1, K_2}(\boldsymbol{X}^{(i)}) = \boldsymbol{y}^{(i)}$ holds for any $i \in [n]$. Here, we formally state the assumptions regarding the data to be memorized.

**Assumption 3.1.** Let $(\boldsymbol{X}^{(1)}, \boldsymbol{y}^{(1)}), \cdots, (\boldsymbol{X}^{(n)}, \boldsymbol{y}^{(n)}) \subset \mathbb{R}^{d \times N} \times [C]^{1 \times N}$ be a sequence of input-output pairs, where $d$ is the input dimension, $N$ is the sequence length, and $C \in \mathbb{N}^+$ represents the size of the vocabulary. We treat $d, C$ as constant and assume the following conditions are satisfied:

1. For every $i \in [n]$ and $k \in [N]$, $\boldsymbol{X}_{:,k}^{(i)} \in [0,1]^d$ and $\|\boldsymbol{X}_{:,k}^{(i)}\|_2 \leq r$ for some $r \geq 1$.

2. For every $i, j \in [n]$ and $k, l \in [N]$, either $\boldsymbol{X}_{:,k}^{(i)} = \boldsymbol{X}_{:,l}^{(j)}$ or $\|\boldsymbol{X}_{:,k}^{(i)} - \boldsymbol{X}_{:,l}^{(j)}\|_2 \geq \delta$ holds for some $0 < \delta \leq 1$.

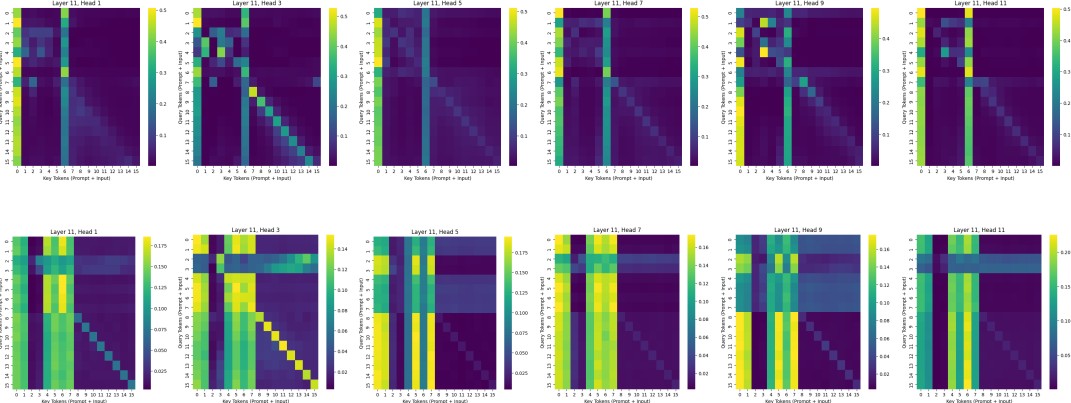

Figure 1: **Attention patterns of prompt tuning Roberta-base [Liu et al., 2019]**. We report the attention patterns of prompt tuning Roberta-base (12 layers, 12 heads) for classification task on SST-2 dataset (first row) and regression task on unstructured random dataset (second row). When prompt tuning on SST-2 dataset, even if we provide a prompt with 8 tokens, the model mainly uses only two of them. However, when prompt tuning on a random dataset consisting of random input-output pairs, the model tends to use more prompt tokens.

3. For every $i \in [n]$, we can not find any $j \in [n]$ with $j \neq i$ such that

$$\boldsymbol{X}^{(i)} = \boldsymbol{X}^{(j)} \quad \text{up to permutations.}$$

4. For every $i \in [n]$, and any $k, l \in [N]$, if $\boldsymbol{X}_{:,k}^{(i)} = \boldsymbol{X}_{:,l}^{(i)}$, we have

$$\boldsymbol{y}_{:,k}^{(i)} = \boldsymbol{y}_{:,l}^{(i)}.$$

Since we only consider finite datasets in this work, we can always find the desired $r$ and $\delta$, meaning that 1) and 2) in Assumption 3.1 are inherently satisfied without imposing extra limitations on the datasets. Mentioned in [Sontag, 1997], without 1) and 2), a linear order of parameters is needed to memorize $n$ arbitrary data points. In order to achieve a sub-linear memorization capacity, 1) and 2) are necessary, which are two common assumptions in existing works [Park et al., 2021, Vardi et al., 2021, Kim et al., 2022, Kajitsuka and Sato, 2023, 2024]. As for 3) and 4) in Assumption 3.1, we know that Transformer architecture is permutation invariant, that is, given any permutation matrix $\boldsymbol{P}, \boldsymbol{T}(\boldsymbol{X}\boldsymbol{P}) = \boldsymbol{T}(\boldsymbol{X})\boldsymbol{P}$ holds for any Transformer $\boldsymbol{T}$. If $\boldsymbol{X}^{(i)} = \boldsymbol{X}^{(j)}$ up to a certain permutation, meaning that $\boldsymbol{T}(\boldsymbol{X}^{(i)}) = \boldsymbol{T}(\boldsymbol{X}^{(j)})$ up to the same permutation. Since in this work, we do not focus on positional encoding, that is why 3) and 4) are assumed to be satisfied. Otherwise, we can easily construct certain datasets that Transformers trivially cannot memorize. In the following, Proposition 3.1 shows that prompt tuning Transformers can memorize single token input-output pairs.

**Proposition 3.1.** *Fix any $n, d \in \mathbb{N}^+$. There exists a Transformer $\boldsymbol{T} \in \mathcal{T}(D_{in}, D_{out}, D_{hid}, H, L)$ such that for any sequence of input-output pairs $(\boldsymbol{x}^{(1)}, y^{(1)}), \cdots, (\boldsymbol{x}^{(n)}, y^{(n)}) \in \mathbb{R}^d \times [C]$ satisfying Assumption 3.1, there exsit a prompt $\boldsymbol{P} \in \mathbb{R}^{D_{in} \times M}$ and $K \in \mathbb{N}^+$ such that*

$$\hat{\boldsymbol{T}}_{\boldsymbol{P}, K, K}(\boldsymbol{x}^{(i)}) = y^{(i)} \quad \text{for any } i \in [n],$$

*where $D_{in} = O(d)$, $D_{out} = O(1)$, $D_{hid} = O(D_{in})$, $H = O(1)$, $L = O(1)$, and $M = \tilde{O}(\sqrt{n})$, $K = \tilde{O}(\sqrt{n})$.*

The proof of Proposition 3.1 is in Appendix D. Our proof depends on the following lemma, which shows that a prompt-tuned Transformer with autoregressive algorithm is able to exactly implement ReLU neural networks with various width and depth by extending the results in [Nakada et al., 2025].

**Lemma 3.1.** *Fix any $W, d \in \mathbb{N}^+$. There exists a Transformer $\boldsymbol{T} \in \mathcal{T}(D_{in}, D_{out}, D_{hid}, H, L)$ such that for any ReLU feed-forward neural network $\boldsymbol{f} \in \mathcal{FF}(W, L, R^d \to R^{d'})$, where $W, L, d, d' \in$*

$\mathbb{N}^+$ with $d' \leq d$, and $n$ inputs $\boldsymbol{X}^{(1)}, \cdots, \boldsymbol{X}^{(n)} \subset \mathbb{R}^{d \times N}$, there exists a prompt $\boldsymbol{P} \in \mathbb{R}^{D_{in} \times M}$, $K_1, K_2 \in \mathbb{N}^+$ such that

$$\hat{\boldsymbol{T}}_{\boldsymbol{P}, K_1, K_2}(\boldsymbol{X}^{(i)}) = [\boldsymbol{f}(\boldsymbol{X}_{:,1}^{(i)}), \cdots, \boldsymbol{f}(\boldsymbol{X}_{:,N}^{(i)})] \quad \text{for any } i \in [n],$$

where $D_{in} = O(W \vee d)$, $D_{out} = O(d')$, $D_{hid} = O(D_{in})$, $H = O(1)$, $L = O(1)$, and $M = O((W \vee d)L)$, $K_1, K_2 = O(NL)$.

In Lemma E.1, the prompt length depends on the width and depth of the target feed-forward neural network, while the number of intermediate steps only depends on the depth. This is because any self-attention layer can capture the interaction between arbitrary token pairs due to the inner product, meaning that self-attention layer is in some sense infinitely "wide" if we plug in long enough prompt tokens. Based on this, we can establish a trade-off between the number of intermediate steps in autoregressive generation between the prompt length, which is summarized in the following corollary.

**Corollary 3.1.** *Fix any $n, d \in \mathbb{N}^+$. There exists a Transformer $\boldsymbol{T} \in \mathcal{T}(D_{in}, D_{out}, D_{hid}, H, L)$ such that for any sequence of input-output pairs $(\boldsymbol{x}^{(1)}, y^{(1)}), \cdots, (\boldsymbol{x}^{(n)}, y^{(n)}) \in \mathbb{R}^d \times [C]$ satisfying Assumption 3.1, there exist a prompt $\boldsymbol{P} \in \mathbb{R}^{D_{in} \times M}$ and $K \in \mathbb{N}^+$ such that*

$$\hat{\boldsymbol{T}}_{\boldsymbol{P}, K, K}(\boldsymbol{x}^{(i)}) = y^{(i)} \quad \text{for any } i \in [n],$$

*where $D_{in} = O(n)$, $D_{out} = O(1)$, $D_{hid} = O(D_{in})$, $H = O(1)$, $L = O(1)$, and $M = \tilde{O}(n)$, $K = \tilde{O}(1)$.*

The result in Corollary 3.1 reveals a trade-off between prompt length and the computational path: a prompt of length $\tilde{O}(\sqrt{n})$ with $\tilde{O}(\sqrt{n})$ intermediate steps and a prompt of length $\tilde{O}(n)$ with $\tilde{O}(1)$ intermediate steps have the same memorization ability, which underscores the advantage of employing longer chains of intermediate computation prior to producing the final answer [Wei et al., 2022]. To extend our result to the case where sequence length $N > 1$, we need to consider the interactions between data tokens instead of only the interaction between data tokens and prompt tokens. This idea results in the following theorem.

**Theorem 3.1.** *Fix any $n, d \in \mathbb{N}^+$. There exists a composition of three Transformers $\boldsymbol{T} = \boldsymbol{T}^{(3)} \circ \boldsymbol{T}^{(2)} \circ \boldsymbol{T}^{(1)}$ with $\boldsymbol{T}^{(i)} \in \mathcal{T}(D_{in}^{(i)}, D_{out}^{(i)}, D_{hid}^{(i)}, H^{(i)}, L^{(i)})$, such that for any sequence of input-output pairs $(\boldsymbol{X}^{(1)}, \boldsymbol{y}^{(1)}), \cdots, (\boldsymbol{X}^{(n)}, \boldsymbol{y}^{(n)}) \in \mathbb{R}^{d \times N} \times [C]^{1 \times N}$ satisfying Assumption 3.1 and $N > 1$, there exist prompts $\boldsymbol{P}^{(i)} \in \mathbb{R}^{D_{in}^{(i)} \times M^{(i)}}$ and $K_1^{(i)}, K_2^{(i)} \in \mathbb{N}$ such that*

$$\hat{\boldsymbol{T}}_{\boldsymbol{P}^{(3)}, K_1^{(3)}, K_2^{(3)}}^{(3)} \circ \hat{\boldsymbol{T}}_{\boldsymbol{P}^{(2)}, K_1^{(2)}, K_2^{(2)}}^{(2)} \circ \hat{\boldsymbol{T}}_{\boldsymbol{P}^{(1)}, K_1^{(1)}, K_2^{(1)}}^{(1)}(\boldsymbol{X}^{(i)}) = \boldsymbol{y}^{(i)} \quad \text{for any } i \in [n],$$

*where $D_{in}^{(i)} = O(d)$, $D_{out}^{(i)} = O(1)$, $D_{hid}^{(i)} = O(D_{in}^{(i)})$, $H^{(i)} = O(1)$, $L^{(i)} = O(1)$, and $M^{(i)} = \tilde{O}(\sqrt{nN})$, $K_1^{(i)}, K_2^{(i)} = \tilde{O}(N \cdot \sqrt{nN})$, for $i = 1, 2, 3$.*

The proof of Theorem of 3.1 is provided in Appendix D, which basically follows the framework in [Kajitsuka and Sato, 2024]. The primary difference between two cases $N = 1$ and $N > 1$ is that we need to include the influence from other data tokens. For example, let $\boldsymbol{X}^{(1)} = [\boldsymbol{x}_1, \boldsymbol{x}_2]$ and $\boldsymbol{X}^{(2)} = [\boldsymbol{x}_1, \boldsymbol{x}_3]$, where $\boldsymbol{x}_2 \neq \boldsymbol{x}_3$. If $\boldsymbol{y}^{(1)}$ and $\boldsymbol{y}^{(2)}$ are their corresponding labels, it is possible that $\boldsymbol{y}_{:,1}^{(1)} \neq \boldsymbol{y}_{:,1}^{(2)}$ even if $\boldsymbol{x}_1$ appears in both sequences, which reflects the real-world fact that the same words in different contexts may carry different meanings. To address this, we need to differentiate $\boldsymbol{x}_1$ in $\boldsymbol{X}^{(1)}$ and $\boldsymbol{X}^{(2)}$ based on the context where it appears by mapping it to different values via self-attention mechanism. Besides, we also need to retain information in originally distinct tokens. This approach is grounded in the concept of "Contextual Mapping", first formulated in [Yun et al., 2019], and further developed in [Kim et al., 2022, Kajitsuka and Sato, 2023, 2024]. In Theorem 3.1, we adopt the technique from [Kajitsuka and Sato, 2024]. In particular, we use Transformer $\boldsymbol{T}^{(i)}$ to map each token into a space with higher dimension to enhance separability. The subsequent layer $\boldsymbol{T}^{(2)}$ is designed to integrate contextual information by simply computing the summation of columns over input sequences. This process ensures that each token is mapped to a unique vector determined by both the token itself and its context: if two tokens differ, or if identical tokens appear in different contexts, the resulting representations will differ as well. One important thing to point out is that in

[Kajitsuka and Sato, 2023], the separation gap between such vectors decays exponentially in terms of the number of data points to be memorized. But in Theorem 3.1. we can keep it as a constant due to the separation in high dimensional space, thereby keeping the required prompt length controllable. Finally, we build $\boldsymbol{T}^{(3)}$ to perform point-wise fitting, mapping the contextualized representations to their corresponding labels.

**Remark 3.1.** $\boldsymbol{T}^{(1)}$ and $\boldsymbol{T}^{(2)}$ in Theorem 3.1 are mainly designed to implement a feed-forward neural network through prompt tuning. According to Lemma E.1, we know the prompt length depends on the rank of $[\boldsymbol{W}, \boldsymbol{b}]$ in each layer of the target feed-forward neural network, where $[\boldsymbol{W}, \boldsymbol{b}]$ denotes the concatenation of the weight matrix and bias term. It is known that neural networks exhibit a low-rank bias, that is, they tend to converge toward solutions with low-rank weight matrices which leads to a possible reduction in prompt length in our setting. One explanation lies in the fact that natural datasets usually present a low-dimensional structure, implying inherently limited intrinsic complexity [Balzano et al., 2025]. As illustrated in Figure 1, prompt tuning indeed favors low-complexity solutions by allowing data tokens to primarily attend to only a small subset of prompt tokens even when longer prompts are provided. Further discussion of this phenomenon can be found in Appendix H.

## 4    Prompt Tuning Random Transformers

The Transformer constructed in Theorem 3.1 is universal across the datasets to be memorized. Its architecture and parameters only depend on the input dimension $d$, while the number of data points only affects the embedding layer. The backbone Transformer contains no task-specific information, rather, it defines a general mechanism governing the interactions between data tokens and prompt tokens. Motivated by this universality, we shift our attention to studying the effectiveness of prompt tuning randomly initialized Transformers. Specifically, we investigate its memorization ability under different word embeddings, how initialization affects the performance and whether datasets with a low-rank structure can reduce the prompt length. This section seeks to understand the inductive bias of Transformer architecture, the fundamental limitations of prompt tuning, and the extent to which prompt tuning can steer the behavior of a model with frozen parameters.

### 4.1    Data Memorization Ability

We conduct experiments to evaluate the memorization ability of prompt tuned random Transformers. Specifically, we employ a two layer randomly initialized Transformer with an embedding size of $512$, matching that of T5-small. The initialization strategy is the default in Pytorch. The data points to be memorized are randomly sampled from the IMDb [Maas et al., 2011] dataset. As our focus is on whether models can memorize finite datasets, we report the training accuracy for dataset sizes of $1600, 2500$, and $3600$. Correspondingly, the prompt length is set to be $40, 50$, and $60$, respectively. Each input sequence is truncated to a length of $8$ and the outputs at the last data token are compared with labels using cross-entropy loss.

We consider two activation functions in each self-attention layer: ReLU and Softmax. The motivation for this choice is that Softmax is widely adopted in practice, while ReLU is used in the construction of Theorem 3.1. For each activation function, we examine two types of word embeddings: those initialized using the T5-small embeddings (frozen), and those that are randomly initialized and also kept untrainable. We compare prompt tuning against embedding-only training of Transformers [Zhong and Andreas, 2024]. Results are shown in Table 1 and a summary of findings is provided below.

**Meaningful word embeddings boost accuracy**    As shown in Table 1, when the word embeddings are initialized with T5-small pretrained embeddings and kept frozen, the resulting accuracy is substantially higher than that with randomly initialized embeddings. This indicates that meaningful pretrained embeddings already encode strong signals that facilitate the mapping from inputs to labels. and are kept untrainable, the accuracy is much higher than random word embeddings. Meaningful word embeddings already contain strong signal mapping from the inputs to labels. In this setting, the influence of prompt tokens appears to serve primarily as a slight calibration rather than a fundamental driver of the model's behavior. This interpretation is supported by the attention patterns visualized in Figure 2: both Softmax and ReLU Transformer automatically allocate most of their attention weights to data tokens, with only minimal focus on prompt tokens.

**ReLU outperforms Softmax** As shown in Table 1, prompt-tuned ReLU Transformers initialized at random consistently outperform their Softmax counterparts under both T5-small and random embedding settings. To further understand this phenomenon, the visualization in Figure 2 offers a plausible explanation. When using T5 embeddings, ReLU-based attention layers can assign disproportionately large weights (often exceeding 10) to data tokens, as they are not constrained by the normalization inherent in Softmax attention. In contrast, Softmax-based attention inevitably distributes its weights across both data and prompt tokens, limiting the emphasis that can be placed on individual data tokens. However, this flexibility of ReLU attention comes with a trade-off: Transformers trained from scratch with ReLU attention often exhibit less stable optimization dynamics, since the lack of normalization may amplify gradient-related instabilities during training [Shen et al., 2023].

Table 1: **Training accuracy** across different activation functions, embeddings and dataset sizes.

| Activation | Prompts | Embedding | Acc(1600) | Acc(2500) | Acc(3600) |
|---|---|---|---|---|---|
| ReLU | ✔ | T5-small | 0.9962 | 0.9944 | 0.9925 |
| ReLU | ✔ | Random | 0.8619 | 0.8216 | 0.8164 |
| ReLU | ✗ | Trainable | 0.9962 | 0.9944 | 0.9947 |
| Softmax | ✔ | T5-small | 0.8888 | 0.8740 | 0.8486 |
| Softmax | ✔ | Random | 0.5375 | 0.5484 | 0.5186 |
| Softmax | ✗ | Trainable | 0.9969 | 0.9928 | 0.9936 |

## 4.2 Random Backbone Still Captures the Low-rank Structure In the Dataset

Based on the observations from Figure 1, 2, 3, and Lemma E.1, the required prompt length appears to depend on the complexity of the downstream tasks. This raises an important question: does this property depend specifically on pretrained LLMs, or can prompt-tuned random Transformer also capture the intrinsic structure of datasets. To investigate this, we examine performance across different prompt lengths using datasets generated by a low-rank feed-forward neural network.

We randomly generate input data $\boldsymbol{X} \in \mathbb{R}^{16 \times 8} \sim N(-1, 4)$, and initialize a ReLU feed-forward neural network $\boldsymbol{f}$ with 8 layers, input dimension 16, and width 32. For the Normal task, $\boldsymbol{f}$ is initialized following the default strategy and kept frozen. For the low-rank mapping task, we replace each concatenation of weight matrix and the bias term, that is, $[\boldsymbol{W}, \boldsymbol{b}]$ by a rank-1 matrix and also make each of them untrainable. For the low-rank input and mapping task, we constrain the input data points to be in a low-rank linear space. The training dataset size is 2000 and test dataset size is 200. As shown in Table 2, prompt tuning a random Transformer can capture the low-rank structure in the dataset and prompts with much less length can even better performance.

Table 2: **MSE Loss** with a low-rank structure in the dataset.

| Dataset ╲ Prompt Length | 10 | 20 | 30 | 40 |
|---|---|---|---|---|
| Normal | 0.3258 | 0.2580 | 0.2674 | 0.2364 |
| Low-rank mapping | 0.0076 | 0.0053 | 0.0026 | 0.0022 |
| Low-rank input and mapping | 0.0038 | 0.0015 | 0.0015 | 0.0007 |

## 4.3 Initialization and Limitation

For mathematical simplicity, let $\boldsymbol{T} = \boldsymbol{\mathcal{F}}_{FF} \circ \boldsymbol{\mathcal{F}}_{SA}$ denote a single layer Transformer consisting of one self-attention layer and one feed-forward layer. We omit the embedding and decoding matrices here and consider a dataset $(\boldsymbol{X}^{(1)}, \boldsymbol{y}^{(1)}), \cdots, (\boldsymbol{X}^{(n)}, \boldsymbol{y}^{(n)}) \subset \mathbb{R}^{d \times N} \times \mathbb{R}^d$ consisting of $n$ input-label pairs to be memorized. Let $\boldsymbol{P} \in \mathbb{R}^{d \times M}$ denote the prompt. If $\boldsymbol{T}([\boldsymbol{P}, \boldsymbol{X}^{(i)}])_{:,-1} = \boldsymbol{y}^{(i)}$ for some $i \in [n]$, this implies that $\boldsymbol{\mathcal{F}}_{SA}([\boldsymbol{P}, \boldsymbol{X}^{(i)}])_{:,-1} \in \boldsymbol{\mathcal{F}}_{FF}^{-1}(\boldsymbol{y}^{(i)})$, where $\boldsymbol{\mathcal{F}}_{FF}^{-1}(\boldsymbol{y}^{(i)}) := \left\{ \boldsymbol{x} \in \mathbb{R}^d : \boldsymbol{\mathcal{F}}_{FF}(\boldsymbol{x}) = \boldsymbol{y}^{(i)} \right\}$. In other words, during the computation of self-attention, the output

representation of the last token must be shifted into the space $\mathcal{F}_{FF}^{-1}(\boldsymbol{y}^{(i)})$. Since all the parameters of $\boldsymbol{T}$ are frozen and further the interactions between data tokens are fixed, this shift can only be achieved through prompt tokens. In [Wang et al., 2023], they explicitly constructed a counterexample showing that regardless of the number of prompt tokens, the data tokens can not be shifted to $\mathcal{F}_{FF}^{-1}(\boldsymbol{y}^{(i)})$. Motivated by this, we can identify several constrained initialization strategies as follows:

- Low-norm $\mathcal{F}_{FF}$: if $\boldsymbol{W}_1$ and $\boldsymbol{W}_2$ in $\mathcal{F}_{FF}$ satisfy $\|\boldsymbol{W}_1\|_2 \cdot \|\boldsymbol{W}_2\|_2 < 1$, then $\mathcal{F}_{FF}$ is invertible [Wang et al., 2023, Behrmann et al., 2019]. As a result, $\mathcal{F}_{FF}^{-1}(\boldsymbol{y}^{(i)})$ remains small.

- Low-rank $\boldsymbol{W}_V$: Since the output of any self-attention layer is a linear combination of the column vectors of $\boldsymbol{W}_V$, constraining the rank of $\boldsymbol{W}_V$ can reduce the diversity of the output.

- Low-rank $\boldsymbol{W}_K$: As shown in [Bhojanapalli et al., 2020], the ranks of $\boldsymbol{W}_K$ and $\boldsymbol{W}_Q$ affect the diversity of the ways tokens interacting with others.

- Low-rank $\boldsymbol{W}_Q$: Similar to the effect of low-rank $\boldsymbol{W}_K$.

As shown in Table 3, low-norm FFN degrades the performance only when the prompt length is small. Among the low-rank variants, low-rank $\boldsymbol{W}_V$ exhibits the lowest accuracy, while low-rank $\boldsymbol{W}_K$ and low-rank $\boldsymbol{W}_Q$ have relatively a slight negative impact on the performance.

Table 3: **Training Accuracy** across different initialization strategies.

| Prompt Length / Initialization | 10 | 20 | 30 | 40 |
|---|---|---|---|---|
| Random | 0.7865 | 0.8263 | 0.8462 | 0.8619 |
| Low-norm FFN | 0.7788 | 0.8119 | 0.8550 | 0.8631 |
| Low-rank $\boldsymbol{W}_V$ | 0.5125 | 0.6625 | 0.5994 | 0.5631 |
| Low-rank $\boldsymbol{W}_K$ | 0.7163 | 0.7356 | 0.7394 | 0.7512 |
| Low-rank $\boldsymbol{W}_Q$ | 0.7506 | 0.7569 | 0.7538 | 0.7481 |

## 5 Conclusion

Our study provides a new perspective on how prompt tuning enables Transformers to memorize and represent data. Rather than viewing prompts merely as task adaptors, we show that they can encode substantial structural information about the dataset itself. Theoretical analysis reveals that the prompt length required for memorization scales sublinearly with the dataset size, while empirical results confirm that prompt-tuned random Transformers can still capture low-rank regularities in data embeddings. These findings suggest that prompt tuning leverages the implicit inductive bias of Transformers to compress information efficiently, even without task-specific supervision.

**Limitations:** Our analysis primarily focuses on simplified Transformer architectures and small datasets, which allow for controlled theoretical and empirical investigation but do not fully capture the complexity of large-scale pretrained models. Extending the current framework to realistic large language model settings, and studying how prompt tuning interacts with pretraining dynamics, remain promising directions for future work.

## Acknowledgments

We thank the anonymous reviewers for their insightful feedback and valuable suggestions that helped improve this paper. Yuanyuan Lin's research was partially supported by the Hong Kong Research Grants Council (No.14304523) and Direct Grants for Research, The Chinese University of Hong Kong.

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

# Appendix

# A   Notation Table

Functions

| | |
|---|---|
| $\boldsymbol{f} : \mathbb{A} \to \mathbb{B}$ | The function $f$ with domain $\mathbb{A}$ and range $\mathbb{B}$ |
| $\boldsymbol{f} \circ \boldsymbol{g}$ | Composition of the functions $f$ and $g$ |
| $\boldsymbol{f}(\boldsymbol{x};\boldsymbol{\theta})$ | A function of $\boldsymbol{x}$ parametrized by $\boldsymbol{\theta}$. (Sometimes we write $f(\boldsymbol{x})$ and omit the argument $\boldsymbol{\theta}$ to lighten notation) |
| $\sigma_R(x)$ | ReLU function, $\max\{x,0\}$ |
| $\sigma_S(\boldsymbol{x})$ | Softmax function, $\sigma_S(\boldsymbol{x})_i = \dfrac{\exp(\boldsymbol{x}_i)}{\sum_{i=1}^{d} \exp(\boldsymbol{x}_i)}$ |
| $\|\boldsymbol{x}\|_p$ | $L^p$ norm of $\boldsymbol{x}$ |
| $\|\boldsymbol{x}\|_2$ | $L^2$ norm of $\boldsymbol{x}$ |
| $\|\boldsymbol{X}\|_2$ | Spectral norm of matrix $\boldsymbol{X}$ |
| $\|\boldsymbol{x}\|_\infty$ | $\infty$ norm of $\boldsymbol{x}$ |
| $x^+$ | Positive part of $x$, i.e., $\max(0,x)$ |
| $x \vee y$ | $\max\{x,y\}$ |
| $\mathcal{F}_{SA}$ | self-attention layer |
| $\mathcal{F}_{FF}$ | Feed-forward layer |
| $\mathcal{T}(D_{in}, D_{out}, D_{hid}, H, L)$ | Transformer neural network class with embedding size $D_{in}$, output dimension $D_{out}$, hidden dimension $h_{hid}$, number of heads $H$, number of layers $L$ |
| $\mathcal{FF}(W, L, \mathbb{R}^d \to \mathbb{R}^{d'})$ | Feed-forward neural network class with width $W$, depth $L$, input dimension $d$ and output dimension $d'$, activation function $\sigma_R$ |
| $\mathcal{R}(W, L, \mathbb{R}^d \to \mathbb{R}^d)$ | Residual feed-forward neural network class with hidden dimension $W$, number of layers $L$, input dimension and output dimension $d$, activation function $\sigma_R$ |

Numbers and Arrays

| | |
|---|---|
| $a$ | A scalar (integer or real) |
| $\boldsymbol{a}$ | A vector |
| $\boldsymbol{A}$ | A matrix |
| $\boldsymbol{I}_n$ | Identity matrix with $n$ rows and $n$ columns |
| $\boldsymbol{I}$ | Identity matrix with dimensionality implied by context |
| $\boldsymbol{e}_i$ | Standard basis vector $[0,\ldots,0,1,0,\ldots,0]$ with a 1 at position $i$ |
| $\boldsymbol{1}_{n \times m}$ | All-one matrix with dimensionality $n \times m$ |

**Sets**

| | |
|---|---|
| $\mathbb{R}$ | The set of real numbers |
| $\mathbb{R}^D$ | The set of $D$-dimensional real vectors |
| $\mathbb{R}^D_{>0}$ | The set of $D$-dimensional positive real vectors |
| $\{0, 1\}$ | The set containing 0 and 1 |
| $\{0, 1, \ldots, n\}$ | The set of all integers between 0 and $n$ |
| $[n]$ | The set of all integers between 1 and $n$, that is, $[n] = \{1, \cdots, n\}$ |

**Indexing**

| | |
|---|---|
| $\boldsymbol{a}_i$ | Element $i$ of vector $\boldsymbol{a}$, with indexing starting at 1 |
| $\boldsymbol{a}_{-1}$ | The last element of vector $\boldsymbol{a}$ |
| $\boldsymbol{A}_{i,j}$ | Element $i, j$ of matrix $\boldsymbol{A}$ |
| $\boldsymbol{A}_{i,:}$ | Row $i$ of matrix $\boldsymbol{A}$ |
| $\boldsymbol{A}_{i:,:}$ | From $i$-th row to the last row of matrix $\boldsymbol{A}$ |
| $\boldsymbol{A}_{:,i}$ | Column $i$ of matrix $\boldsymbol{A}$ |
| $\boldsymbol{A}_{:,i:}$ | from $i$-th column to the last column of matrix $\boldsymbol{A}$ |

**Asymptotics**

| | |
|---|---|
| $f(n) = O(g(n))$ | $f$ grows at most as fast as $g$ for sufficiently large $n$ |
| $f(n) = \tilde{O}(g(n))$ | $f$ grows at most as fast as $g$ for sufficiently large $n$, up to logarithmic factors |
| $f(n) = \Omega(g(n))$ | $f$ grows at least as fast as $g$ for sufficiently large $n$ |
| $f \lesssim g$ | There exists a positive constant $c$ such that $f \leq cg$ holds |

# B   Additional Related Works

Another line of theoretical study on the expressive capacity of Transformers is to consider the approximation ability of Transformers, that is, can Transformers approximate functions that belong to a given function class. The most seminal work by [Yun et al., 2019] provided the first universal approximation theorem for Transformers, showing that any continuous sequence-to-sequence functions defined on a compact domain can be approximated by Transformers to any finite precision. They also extended the results to sparse Transformers [Yun et al., 2020]. Later, Gurevych et al. [2022] established a constructive method, proving that Transformers can approximate piecewise polynomials. Jiang and Li [2024] built their results of approximating continuous functions by shallow Transformers based on the Kolmogorov Representation Theorem. Takakura and Suzuki [2023] provided both approximation and estimation error with $\gamma$-smooth function class under the assumption that the input is infinite dimensional. Similarly, Havrilla and Liao [2024] assumed that the input data has a low-dimensional manifold structure and established approximation results for $\beta$-Hölder continuous functions with Transformers. They also gave an explicit form of the scaling law by utilizing techniques from non-parametric statistics. Kajitsuka and Sato [2023] showed that Transformers with one single-head self-attention layer are able to be a universal approximator by exploring the relationship between the Softmax function in self-attention layers and the Boltzmann operator. Takeshita and Imaizumi [2025] proved that Transformers can efficiently approximate column-symmetric polynomials with respect to the number of parameters. Recently, Jiao et al. [2025a] derived the approximation results of Transformers for Hölder class and Sobolev class under $L^p$-norm, where $p \in [0, +\infty]$. Besides, their another work [Jiao et al., 2025b] showed that Transformers can overcome the curse of dimensionality based on the Kolmogorov-Arnold Representation Theorem. Concurrent work Hu et al. [2025] made an effort to avoid the dependence

on large ReLU feed-forward layers, by proving that self-attention layers alone can approximate a generalized version of ReLU function and hence subsumes any known approximators based on ReLU feed-forward neural networks. Similarly, Liu et al. [2025] proved that a single self-attention layer, preceded by sum-of-linear transformations, is capable of approximating any continuous function on a compact domain under $L^\infty$-norm, highlighting the inherent expressive ability of self-attention mechanism alone.

## C  Construction of Prompts

We basically follow the framework introduced in [Nakada et al., 2025] to construct prompts.

Given prompt $\boldsymbol{P} \in \mathbb{R}^{D_{in} \times M}$, for any $\boldsymbol{X} \in \mathbb{R}^{d \times N}$, let $\boldsymbol{X}_{emb} = [\boldsymbol{P}, \boldsymbol{\mathcal{E}}_{in}(\boldsymbol{X})]$. We have

$$\boldsymbol{X}_{emb} = [\underbrace{\boldsymbol{x}_1, \cdots, \boldsymbol{x}_M}_{\text{prompt tokens}}, \underbrace{\boldsymbol{x}_{M+1}, \cdots, \boldsymbol{x}_{M+N}}_{\text{data tokens}}] \in \mathbb{R}^{D_{in} \times (N+M)},$$

Where $D_{in} \geq d$ is the embedding size, $M$ is the length of the prompt and $N$ is the length of the input sequence. In this work, we set $D_{in} = 4d + 8$. Each prompt token is divided into two parts: guidance embedding and positional embedding The exact form of prompt tokens is defined below

$$\boldsymbol{x}_j := (\underbrace{\boldsymbol{u}_j^\top}_{\text{guidance embedding}}, \underbrace{\boldsymbol{p}_j^\top}_{\text{positional embedding}})^\top = \begin{pmatrix} \boldsymbol{u}_j \\ \boldsymbol{0}_{3d+4} \\ 1 \\ S \\ Sw_j \\ Sj \end{pmatrix} \in \mathbb{R}^{4d+8} \quad \text{for } i \in [M], \qquad \text{(C.1)}$$

where $\boldsymbol{u}_j \in \mathbb{R}^d$ is called the guidance embedding, and $\boldsymbol{p}_j \in \mathbb{R}^{3d+8}$ is the positional encoding, defined as $\boldsymbol{p}_j = \boldsymbol{p}(w_j, j, S) := (\boldsymbol{0}_{3d+4}^T, 1, S, Sw_j, Sj)^T$. The large enough $S > 0$ denotes the scaling factor, which ensures that the positional encoding remains significant relative to guidance embedding. The constant 1 serves as a bias term, while the zeros in $\boldsymbol{p}_j$ function as a temporal memory to store intermediate variables because of the skip connection. Guidance embeddings aim to guide the Transformers, which store the task-specific information. The integer $w_j \in \mathbb{N}^+$ works as an indicator to force Transformers to focus on certain prompt tokens in different autoregressive generation steps. Given the same scaling factor $S$ in prompt tokens, data tokens $\boldsymbol{x}_{M+1}, \cdots, \boldsymbol{x}_{M+N}$ are defined as follows

$$\boldsymbol{x}_{M+i} := (\boldsymbol{X}_{:,i}^\top, \boldsymbol{p}(0, M+i, S)^\top)^\top = \begin{pmatrix} \boldsymbol{X}_{:,i}^\top \\ \boldsymbol{0}_{3d+4} \\ 1 \\ S \\ 0 \\ S \cdot (M+i) \end{pmatrix} \in \mathbb{R}^{4d+8} \quad \text{for } i \in [N]. \qquad \text{(C.2)}$$

## D  Proofs of Section 3

This section provides all the detailed proofs in Section 3.

### D.1  Proof of Lemma 3.1

Lemma 3.1 shows that prompt tuning a Transformer can exactly implement a ReLU feed-forward neural network, that is, given the same input, the output is the same. Our results are built upon [Nakada et al., 2025], by extending their results to the case where the width of the target ReLU feed-forward neural network can be smaller than the input dimension.

**Lemma D.1** (Restatement of Lemma 3.1). *Fix any $W, d \in \mathbb{N}^+$. There exists a Transformer $\boldsymbol{T} \in \mathcal{T}(D_{in}, D_{out}, D_{hid}, H, L)$ such that for any ReLU feed-forward neural network $\boldsymbol{f} \in \mathcal{FF}(W, L, R^d \rightarrow R^{d'})$, where $W, L, d, d' \in \mathbb{N}^+$ with $d' \leq d$, and $n$ inputs $\boldsymbol{X}^{(1)}, \cdots, \boldsymbol{X}^{(n)} \subset \mathbb{R}^{d \times N}$, there exists a prompt $\boldsymbol{P} \in \mathbb{R}^{D_{in} \times M}$, $K_1, K_2 \in \mathbb{N}^+$ such that*

$$\hat{\boldsymbol{T}}_{\boldsymbol{P}, K_1, K_2}(\boldsymbol{X}^{(i)}) = [\boldsymbol{f}(\boldsymbol{X}^{(i)}_{:,1}), \cdots, \boldsymbol{f}(\boldsymbol{X}^{(i)}_{:,N})] \quad \text{for any } i \in [n],$$

*where $D_{in} = O(W \vee d)$, $D_{out} = O(d')$, $D_{hid} = O(D_{in})$, $H = O(1)$, $L = O(1)$, and $M = O((W \vee d)L)$, $K_1, K_2 = O(NL)$.*

*Proof of Lemma 3.1.* Suppose $\boldsymbol{f}$ has the following form

$$\boldsymbol{f} = \mathcal{L}_L \circ \sigma_R \circ \mathcal{L}_{L-1} \circ \cdots \mathcal{L}_1 \circ \sigma_R \circ \mathcal{L}_0(\boldsymbol{x}),$$

where $\mathcal{L}_\ell(\boldsymbol{x}) = \boldsymbol{W}_\ell \boldsymbol{x} + \boldsymbol{b}_\ell$ with $\boldsymbol{W}_\ell \in \mathbb{R}^{d_{\ell+1} \times d_\ell}$ and $\boldsymbol{b}_\ell \in \mathbb{R}^{d_{\ell+1}}$ for $\ell = 0, 1, \cdots, L$. In the following, we consider two cases: **Case 1**: $W \geq d$, **Case 2**: $W < d$, respectively.

**Case 1:** $W \geq d$  Let $\hat{\boldsymbol{W}}_\ell$ for $\ell \in [L] \cup \{0\}$ denote a series of matrices that incorporate the bias terms $\boldsymbol{b}_\ell$ into multiplication of matrices with dimension $\mathbb{R}^{((W+1) \times (W+1))}$, which are defined as

$$\hat{\boldsymbol{W}}_\ell = \begin{pmatrix} \boldsymbol{W}_\ell & \boldsymbol{b}_\ell & \boldsymbol{0} \\ \boldsymbol{0} & 1 & \boldsymbol{0} \\ \boldsymbol{0} & \boldsymbol{0} & \boldsymbol{0} \end{pmatrix} \in \mathbb{R}^{(W+1) \times (W+1)} \quad \text{for } i \in [L] \cup \{0\}.$$

With $\hat{\boldsymbol{W}}_\ell$ in hand, we define a new feed-forward neural network $\hat{\boldsymbol{f}}$ as

$$\hat{\boldsymbol{f}}(\boldsymbol{x}) = \hat{\mathcal{L}}_L \circ \sigma_R \circ \hat{\mathcal{L}}_{L-1} \circ \cdots \hat{\mathcal{L}}_1 \circ \sigma_R \circ \hat{\mathcal{L}}_0(\boldsymbol{x}),$$

where $\hat{L}_\ell(\boldsymbol{x}) := \hat{\boldsymbol{W}}_\ell(\boldsymbol{x})$. It is clear that $\boldsymbol{f} \in \mathcal{FF}(W+1, L, \mathbb{R}^{W+1} \rightarrow \mathbb{R}^{W+1})$. For any $\boldsymbol{X} = [\boldsymbol{x}_1, \cdots, \boldsymbol{x}_N] \in \mathbb{R}^{d \times N}$, let $\hat{\boldsymbol{X}} = [\hat{\boldsymbol{x}}_1, \cdots, \hat{\boldsymbol{x}}_N] \in \mathbb{R}^{(W+1) \times N}$, where $\hat{\boldsymbol{x}}_i = \left[\boldsymbol{x}_i^\top, 1, \boldsymbol{0}_{(W-d)}^\top\right]$ for $i \in [N]$. It is direct to verity that

$$\hat{\boldsymbol{f}}(\hat{\boldsymbol{x}}_i)_{1:d'} = \boldsymbol{f}(\boldsymbol{x}_i),$$

and

$$\hat{\boldsymbol{f}}(\hat{\boldsymbol{x}}_i)_{1:d'+1} = \left[\boldsymbol{f}(\boldsymbol{x}_i)^\top, 1\right]^\top.$$

By applying Lemma E.1 to $\hat{\boldsymbol{f}}$, we know that there exists a Transformer $\boldsymbol{T} \in \mathcal{T}(D_{in}, D_{out}, D_{hid}, H, L)$, and prompt $\boldsymbol{P} \in \mathbb{R}^{D_{in} \times M}$, $K_i \in \mathbb{N}^+$ such that

$$\hat{\boldsymbol{T}}_{\boldsymbol{P}, K_i, K_i}(\boldsymbol{X}) = \hat{\boldsymbol{f}}(\hat{\boldsymbol{x}}_i)_{1:d'} = \boldsymbol{f}(\boldsymbol{x}_i)^\top \quad \text{for any } i \in [N],$$

or by using different $\mathcal{E}_{out}$, we have

$$\hat{\boldsymbol{T}}_{\boldsymbol{P}, K_i, K_i}(\boldsymbol{X}) = \hat{\boldsymbol{f}}(\hat{\boldsymbol{x}}_i)_{1:d'+1} = \left[\boldsymbol{f}(\boldsymbol{x}_i)^\top, 1\right]^\top \quad \text{for any } i \in [N],$$

where $K_{i+1} = K_i + 1$. The proof of **Case 1** is completed by pointing out that $D_{in} = O(W)$, $D_{out} = O(d)$, $D_{hid} = O(D_{in})$, and $H = O(1)$, $L = O(1)$, and $M = O((W+1)(L+1))$, $K_i = O(N(L+2))$.

**Case 2:** $W < d$  Similar to **Case 1**, let $\hat{\boldsymbol{W}}_\ell$ for $\ell \in [L] \cup \{0\}$ denote a series of matrices that incorporate the bias terms $\boldsymbol{b}_\ell$ into multiplication of matrices with dimension $\mathbb{R}^{((d+1) \times (d+1))}$, which are defined as

$$\hat{\boldsymbol{W}}_\ell = \begin{pmatrix} \boldsymbol{W}_\ell & \boldsymbol{b}_\ell & \boldsymbol{0} \\ \boldsymbol{0} & 1 & \boldsymbol{0} \\ \boldsymbol{0} & \boldsymbol{0} & \boldsymbol{0} \end{pmatrix} \in \mathbb{R}^{(d+1) \times (d+1)} \quad \text{for } i \in [L] \cup \{0\}.$$

With $\hat{W}_\ell$ in hand, we define a new feed-forward neural network $\hat{f}$ as

$$\hat{f}(x) = \hat{\mathcal{L}}_L \circ \sigma_R \circ \hat{\mathcal{L}}_{L-1} \circ \cdots \hat{\mathcal{L}}_1 \circ \sigma_R \circ \hat{\mathcal{L}}_0(x),$$

where $\hat{L}_\ell(x) := \hat{W}_\ell(x)$. It is clear that $f \in \mathcal{FF}(d+1, L, R^{d+1} \to \mathbb{R}^{d+1})$. For any $X = [x_1, \cdots, x_N] \in \mathbb{R}^{d \times N}$, let $\hat{X} = [\hat{x}_1, \cdots, \hat{x}_N] \in \mathbb{R}^{(d+1) \times N}$, where $\hat{x}_i = [x_i^\top, 1]^\top$ for $i \in [N]$. It is direct to verify that

$$\hat{f}(\hat{x}_i)_{1:d'} = f(x_i),$$

and

$$\hat{f}(\hat{x}_i)_{1:d'+1} = [f(x_i)^\top, 1]^\top.$$

By applying Lemma E.1 to $\hat{f}$, we know that there exists a Transformer $T \in \mathcal{T}(D_{in}, D_{out}, D_{hid}, H, L)$, and prompt $P \in \mathbb{R}^{D_{in} \times M}$, $K_i \in \mathbb{N}^+$ such that

$$\hat{T}_{P,K_i,K_i}(X) = \hat{f}(\hat{x}_i)_{1:d'} = f(x_i)^\top \quad \text{for any } i \in [N],$$

or by using different $\mathcal{E}_{out}$, we have

$$\hat{T}_{P,K_i,K_i}(X) = \hat{f}(\hat{x}_i)_{1:d'+1} = [f(x_i)^\top, 1]^\top \quad \text{for any } i \in [N],$$

where $K_{i+1} = K_i + 1$. The proof of **Case 2** is completed by pointing out that $D_{in} = O(W)$, $D_{out} = O(d)$, $D_{hid} = O(D_{in})$, and $H = O(1)$, $L = O(1)$, and $M = O((d+1)(L+1))$, $K_i = O(N(L+2))$.

Finally, we have proved that for any feed-forward neural network $f \in \mathcal{FF}(W, L, \mathbb{R}^d \to \mathbb{R}^{d'})$, there exists Transformer $\mathcal{T}(D_{in}, D_{out}, D_{hid}, H, L)$ such that

$$\hat{T}_{P,K_1,K_2}(X) = [f(X_{:,1}), \cdots, f(X_{:,N})] \quad \text{for any } X \in [0,1]^{d \times N},$$

where $D_{in} = O(W \vee d)$, $D_{out} = O(d)$, $D_{hid} = O(D_{in})$, $H = O(1)$, $L = O(1)$, and $M = O((W \vee d)L)$, $K_1, K_2 = O(NL)$. The proof is completed by considering $\hat{T}_{P,K_1,K_2}(X^{(i)})$ for $i \in [n]$. $\qquad\square$

## D.2  Proof of Proposition 3.1

**Proposition D.1** (Restatement of Proposition 3.1). *Fix any $n, d \in \mathbb{N}^+$. There exists a Transformer $T \in \mathcal{T}(D_{in}, D_{out}, D_{hid}, H, L)$ such that for any sequence of input-output pairs $(x^{(1)}, y^{(1)}), \cdots, (x^{(n)}, y^{(n)}) \in \mathbb{R}^d \times [C]$ satisfying Assumption 3.1, there exsit a prompt $P \in \mathbb{R}^{D_{in} \times M}$ and $K \in \mathbb{N}^+$ such that*

$$\hat{T}_{P,K,K}(x^{(i)}) = y^{(i)} \quad \text{for any } i \in [n],$$

*where $D_{in} = O(d)$, $D_{out} = O(1)$, $D_{hid} = O(D_{in})$, $H = O(1)$, $L = O(1)$, and $M = \tilde{O}(\sqrt{n})$, $K = \tilde{O}(\sqrt{n})$.*

*Proof of Proposition 3.1.* According to Assumption 3.1, we know that $x^{(i)} \in [0,1]^d$ with $\|x^{(i)}\|_2 \le r$, and $\|x^{(i)} - x^{(j)}\|_2 \ge \delta$ for some $r \ge 1$, $0 < \delta \le 1$. By applying Lemma E.2 to $(x^{(1)}, y^{(1)}), \cdots, (x^{(n)}, y^{(n)})$, there exists a feed-forward neural network $f \in \mathcal{T}(W, L, \mathbb{R}^d \to \mathbb{R})$ with $W = O(1)$, $L = \tilde{O}(\sqrt{n})$ such that $f(x^{(i)}) = y^{(i)}$. Then, by applying Lemma 3.1 to $f$, there exists a Transformer $T \in \mathcal{T}(D_{in}, D_{out}, D_{hid}, H, L)$, and prompt $P \in \mathbb{R}^{D_{in} \times M}$, $K_i \in \mathbb{N}^+$ such that

$$T_{P,K_i,K_i}(x^{(i)}) = y^{(i)} \quad \text{for any } i \in [n],$$

where $O(D_{in}) = O(d)$, $D_{out} = O(1)$, $D_{hid} = O(D_{in})$, $H = O(1)$, $L = O(1)$, and $M = \tilde{O}(\sqrt{n})$, $K_i = \tilde{O}(\sqrt{n})$. $\qquad\square$

### D.3 Proof of Corollary 3.1

The proof of Corollary 3.1 is similar to that of Proposition 3.1, where we only need to replace Lemma E.2 by Lemma E.3.

**Corollary D.1** (Restatement of Corollary 3.1). *Fix any $n, d \in \mathbb{N}^+$. There exists a Transformer $\boldsymbol{T} \in \mathcal{T}(D_{in}, D_{out}, D_{hid}, H, L)$ such that for any sequence of input-output pairs $(\boldsymbol{x}^{(1)}, y^{(1)}), \cdots, (\boldsymbol{x}^{(n)}, y^{(n)}) \in \mathbb{R}^d \times [C]$ satisfying Assumption 3.1, there exist a prompt $\boldsymbol{P} \in \mathbb{R}^{D_{in} \times M}$ and $K \in \mathbb{N}^+$ such that*

$$\hat{\boldsymbol{T}}_{\boldsymbol{P}, K, K}(\boldsymbol{x}^{(i)}) = y^{(i)} \quad \text{for any } i \in [n],$$

*where $D_{in} = O(n)$, $D_{out} = O(1)$, $D_{hid} = O(D_{in})$, $H = O(1)$, $L = O(1)$, and $M = \tilde{O}(n)$, $K = \tilde{O}(1)$.*

*Proof of Corollary 3.1.* According to Assumption 3.1, we know that $\boldsymbol{x}^{(i)} \in [0,1]^d$ with $\|\boldsymbol{x}^{(i)}\|_2 \leq r$, and $\|\boldsymbol{x}^{(i)} - \boldsymbol{x}^{(j)}\|_2 \geq \delta$ for some $r \geq 1$, $0 < \delta \leq 1$. By applying Lemma E.3 to $(\boldsymbol{x}^{(1)}, y^{(1)}), \cdots, (\boldsymbol{x}^{(n)}, y^{(n)})$, there exists a feed-forward neural network $\boldsymbol{f} \in \mathcal{T}(W, L, \mathbb{R}^d \to \mathbb{R})$ with $W = O(n)$, $L = \tilde{O}(1)$ such that $\boldsymbol{f}(\boldsymbol{x}^{(i)}) = y^{(i)}$. Then, by applying Lemma 3.1 to $\boldsymbol{f}$, there exists a Transformer $\boldsymbol{T} \in \mathcal{T}(D_{in}, D_{out}, D_{hid}, H, L)$, and prompt $\boldsymbol{P} \in \mathbb{R}^{D_{in} \times M}$, $K \in \mathbb{N}^+$ such that

$$\boldsymbol{T}_{\boldsymbol{P}, K, K}(\boldsymbol{x}^{(i)}) = y^{(i)} \quad \text{for any } i \in [n],$$

where $O(D_{in}) = O(n)$, $D_{out} = O(1)$, $D_{hid} = O(D_{in})$, $H = O(1)$, $L = O(1)$, and $M = \tilde{O}(n)$, $K = \tilde{O}(1)$. $\qquad \square$

### D.4 Proof of Theorem 3.1

**Theorem D.1** (Restatement of Theorem 3.1). *Fix any $n, d \in \mathbb{N}^+$. There exists a composition of three Transformers $\boldsymbol{T} = \boldsymbol{T}^{(3)} \circ \boldsymbol{T}^{(2)} \circ \boldsymbol{T}^{(1)}$ with $\boldsymbol{T}^{(i)} \in \mathcal{T}(D_{in}^{(i)}, D_{out}^{(i)}, D_{hid}^{(i)}, H, L)$, such that for any sequence of input-output pairs $(\boldsymbol{X}^{(1)}, \boldsymbol{y}^{(1)}), \cdots, (\boldsymbol{X}^{(n)}, \boldsymbol{y}^{(n)}) \in \mathbb{R}^{d \times N} \times [C]^{1 \times N}$ satisfying Assumption 3.1 and $N > 1$, there exist prompts $\boldsymbol{P}^{(i)} \in \mathbb{R}^{D_{in}^{(i)} \times M^{(i)}}$ and $K_1^{(i)}, K_2^{(i)} \in \mathbb{N}$ such that*

$$\hat{\boldsymbol{T}}^{(3)}_{\boldsymbol{P}^{(3)}, K_1^{(3)}, K_2^{(3)}} \circ \hat{\boldsymbol{T}}^{(2)}_{\boldsymbol{P}^{(2)}, K_1^{(2)}, K_2^{(2)}} \circ \hat{\boldsymbol{T}}^{(1)}_{\boldsymbol{P}^{(1)}, K_1^{(1)}, K_2^{(1)}} (\boldsymbol{X}^{(i)}) = \boldsymbol{y}^{(i)} \quad \text{for any } i \in [n],$$

*where $D_{in}^{(i)} = O(d)$, $D_{out}^{(i)} = O(1)$, $D_{hid}^{(i)} = O(D_{in}^{(i)})$, $H^{(i)} = O(1)$, $L^{(i)} = O(1)$, and $M^{(i)} = \tilde{O}(\sqrt{nN})$, $K_1^{(i)}, K_2^{(i)} = \tilde{O}(N \cdot \sqrt{nN})$, for $i = 1, 2, 3$.*

*Proof of Theorem 3.1.* Our proof basically follows [Kajitsuka and Sato, 2024]. **Step 1:** For any $\boldsymbol{X} \in \mathbb{R}^{d \times N}$, define function $\boldsymbol{m}(\boldsymbol{x}) : \mathbb{R}^d \to \mathbb{N}$ as counting the number of occurences of $\boldsymbol{x}$ in $\boldsymbol{X}$, that is, $\boldsymbol{m}(\boldsymbol{x}) = |\{k \in [N] : \boldsymbol{X}_{:,k} = \boldsymbol{x}\}|$. Let $\boldsymbol{m}_i$ denote the corresponding $\boldsymbol{m}(\boldsymbol{x})$ of $\boldsymbol{X}^{(i)}$. Firstly, we show that for any dataset $\{(\boldsymbol{X}^{(i)}, \boldsymbol{y}^{(i)})\}_{i=1}^n$ satisfying Assumption 3.1, there exists a subset $\mathbb{A} \subset \mathbb{R}^d$ with $|\mathbb{A}| \leq n$ such that for any $i, j \in [n]$, there exists $\boldsymbol{x} \in \mathbb{A}$ such that

$$\boldsymbol{m}_i(\boldsymbol{x}) \neq \boldsymbol{m}_j(\boldsymbol{x}),$$

which means that we can only use a set containing less than $n$ elements to differentiate each $\boldsymbol{X}^{(i)}$. To prove this fact, we can see that the case when $n = 1$ is obvious. We assume that the case for $n = k$ is correct, and prove the case where $n = k + 1$. Let $(\boldsymbol{X}^{(1)}, \boldsymbol{y}^{(1)}), \cdots, (\boldsymbol{X}^{(k+1)}, \boldsymbol{y}^{(k+1)}) \subset \mathbb{R}^{d \times N} \times [C]^{1 \times N}$ be a sequence of input-output pairs, which satisfy Assumption 3.1. Then, by applying the assumption about $n = k$ to the first $k$ data points, we know that there exists a subset $\mathbb{A} \subset \mathbb{R}^d$ with $|\mathbb{A}| \leq k$ such that for any $i \neq j \in [n]$, there exists $\boldsymbol{x} \in \mathbb{A}$ such that $\boldsymbol{m}_i(\boldsymbol{x}) \neq \boldsymbol{m}_j(\boldsymbol{x})$. If there exists $i \in [k]$ such that for any $\boldsymbol{x} \in \mathbb{A}$, the following holds

$$\boldsymbol{m}_i(\boldsymbol{x}) = \boldsymbol{m}_{k+1}(\boldsymbol{x}).$$

Then, we can find an element $\boldsymbol{x} \in \mathbb{R}^d \backslash \mathbb{A}$ such that $\boldsymbol{m}_i(\boldsymbol{x}) \neq \boldsymbol{m}_{k+1}(\boldsymbol{x})$ due to the fact that $\boldsymbol{X}^{(i)} \neq \boldsymbol{X}^{(j)}$ under any permutations. Subsequently, we define a new set $\mathbb{A}' = \mathbb{A} \cup \{\boldsymbol{x}\}$, which is the desired set for the case where $n = k + 1$ and $|\mathbb{A}'| \leq k + 1$ clearly. The induction is comleted.

**Step 2: Construction of $T^{(1)}$** In this step, we present the construction of $T^{(1)}$. Let $S$ denote the set introduced in **Step 1**. Let $g : S \to [|S|]$ be an arbitrary bijective function, which maps each token in $S$ to a unique positive integer less than $|S|$. For each token $x \in S$, we consider mapping it to an element in a high-dimension space, in which they are separable. With this motivation in hand, we define $\hat{x}^{(i)}$ for $i \in [n]$ by

$$\hat{x}^{(i)} := \sum_{1 \leq k \leq N, X_{:,k}^{(i)} \in S} e_{g(X_{:,k}^{(i)})},$$

where $e_{g(X_{:,k}^{(i)})} \in \{0,1\}^{|S|}$ is a one-hot vector with 1 in the $g(X_{:,k}^{(i)})$-th position. According to Assumption 3.1 and definition of $S$, for any $i \neq j \in [n]$, we can always either find $k, l \in [N]$ such that $X_{:,k}^{(i)} \neq X_{:,l}^{(j)} \in S$ or $X_{:,k}^{(i)} = X_{:,l}^{(j)} \in S$ but $m_i(X_{:,k}^{(i)}) \neq m_j(X_{:,l}^{(j)})$. As a result, we have the following fact

$$\left\| \hat{x}^{(i)} - \hat{x}^{(j)} \right\|_2^2 \geq 1, \tag{D.1}$$

holds for any $i \neq j \in [n]$, and the norm of each $\hat{x}^{(i)}$ is upper bounded by

$$\left\| \hat{x}^{(i)} \right\|_2 \leq \sum_{1 \leq k \leq N, X_{:,k}^{(i)} \in S} \left\| e_{g(X_{:,k}^{(i)})} \right\|_2 \leq N.$$

By applying Lemma E.5 to $\hat{x}^{(1)}, \cdots, \hat{x}^{(n)}$, there exists a unit vector $v \in \mathbb{R}^{|S|}$ such that

$$\frac{1}{n^2} \sqrt{\frac{8}{\pi |S|}} \left\| \hat{x}^{(i)} - \hat{x}^{(j)} \right\|_2 \leq \left| v^\top (\hat{x}^{(i)} - \hat{x}^{(j)}) \right| \leq \left\| \hat{x}^{(i)} - \hat{x}^{(j)} \right\|_2 \tag{D.2}$$

holds for any $i, j \in [n]$. Let $h$ be the function $h : S \to \mathbb{Z}$, $x \to \lceil n^2 |S| \sqrt{\pi} v_{g(x)} \rceil$ and $\hat{v} := \left( \lceil n^2 |S| \sqrt{\pi} v_1 \rceil, \cdots, \lceil n^2 |S| \sqrt{\pi} v_{|S|} \rceil \right) \in \mathbb{N}^{|S|}$. The motivation of introducing $\hat{v}$ is to use an integer vector to approximate $n^2 |S| \sqrt{\pi} v$ and make it convenient for later techniques based on memorization of integer labels. It is clear that

$$\left\| n^2 |S| \sqrt{\pi} v - \hat{v} \right\|_2 \leq \sum_{i=1}^{|S|} 1 = |S|. \tag{D.3}$$

For any $X^{(i)}$, we have

$$\sum_{1 \leq k \leq N, X_{:,k}^{(i)} \in S} h(X_{:,k}^{(i)}) = \sum_{1 \leq k \leq N, X_{:,k}^{(i)} \in S} \lceil n^2 |S| \sqrt{\pi} v_{g(X_{:,k}^{(i)})} \rceil$$

$$= \sum_{1 \leq k \leq N, X_{:,k}^{(i)} \in S} \hat{v}^\top \cdot e_{g(X_{:,k}^{(i)})}$$

$$= \hat{v}^\top \cdot \hat{x}^{(i)}.$$

We point out that $\sum_{1 \leq k \leq N, X_{:,k}^{(i)} \in S} h(X_{:,k}^{(i)})$ can be viewed as the integrated information of the sequence $X^{(i)}$. The explanation is as follows. For any $i \neq j \in [n]$, we have

$$\left| \sum_{1 \leq k \leq N, X_{:,k}^{(i)} \in S} h(X_{:,k}^{(i)}) - \sum_{1 \leq k \leq N, X_{:,k}^{(j)} \in S} h(X_{:,k}^{(j)}) \right| \tag{D.4}$$

$$= \left| \hat{v}^\top \cdot (\hat{x}^{(i)} - \hat{x}^{(j)}) \right| \tag{D.5}$$

$$\geq \left| n^2 |S| \sqrt{\pi} v^\top (\hat{x}^{(i)} - \hat{x}^{(j)}) \right| - \left| (n^2 |S| \sqrt{\pi} v - \hat{v})^\top (\hat{x}^{(i)} - \hat{x}^{(j)}) \right| \tag{D.6}$$

$$> 2\sqrt{|S|} \left\| \hat{x}^{(i)} - \hat{x}^{(j)} \right\|_2 - \sqrt{|S|} \left\| \hat{x}^{(i)} - \hat{x}^{(j)} \right\|_2 \tag{D.7}$$

$$= \sqrt{|S|} \left\| \hat{x}^{(i)} - \hat{x}^{(j)} \right\|_2 \tag{D.8}$$

$$\geq \sqrt{|S|}, \tag{D.9}$$

where the last inequality is derived from Eq.D.1. Besides, the second to last inequality is derived from Eq.D.3, and the following fact

$$\left| n^2 |S| \sqrt{\pi} \boldsymbol{v}^\top (\hat{\boldsymbol{x}}^{(i)} - \hat{\boldsymbol{x}}^{(j)}) \right| = n^2 |S| \sqrt{\pi} \left| \boldsymbol{v}^\top (\hat{\boldsymbol{x}}^{(i)} - \hat{\boldsymbol{x}}^{(j)}) \right|$$

$$\geq n^2 |S| \sqrt{\pi} \cdot \frac{1}{n^2} \sqrt{\frac{8}{\pi |S|}} \left\| \hat{\boldsymbol{x}}^{(i)} - \hat{\boldsymbol{x}}^{(j)} \right\|_2$$

$$> 2\sqrt{|S|} \left\| \hat{\boldsymbol{x}}^{(i)} - \hat{\boldsymbol{x}}^{(j)} \right\|_2,$$

where the first inequality is based on Eq.D.2. Up to now, we have constructed a function $\boldsymbol{h}$ such that $\sum_{1 \leq k \leq N, \boldsymbol{X}^{(i)}_{:,k} \in S} \boldsymbol{h}(\boldsymbol{X}^{(i)}_{:,k})$ are different with each other and the difference between them is large. In the meantime it is straightforward to verify that $|\boldsymbol{h}(\boldsymbol{x})| \leq 2n^2 |S| \sqrt{\pi}$, and we can further derive

$$\sum_{1 \leq k \leq N, \boldsymbol{X}^{(i)}_{:,k} \in S} \boldsymbol{h}(\boldsymbol{X}^{(i)}_{:,k}) \leq 2n^4 \sqrt{\pi}, \tag{D.10}$$

where we use the fact $|S| \leq n$. With function $\boldsymbol{h}$ in hand, we define a function $\phi : \mathbb{R}^d \to \mathbb{R}$ such that

$$\phi(\boldsymbol{x}) := \begin{cases} \boldsymbol{h}(\boldsymbol{x}) & \boldsymbol{x} \in S, \\ 0 & \text{otherwise.} \end{cases}$$

Notice that the possible number of inputs for the function $\phi$ is at most $nN$, and all outputs are less than or equal to $\lceil 2n^2 |S| \sqrt{\pi} \rceil \leq \lceil 2n^3 \sqrt{\pi} \rceil$. Then, we consider use a feed-forward neural network to implement $\phi$. By applying Lemma E.2, we know there exists a feed-forward neural network $\boldsymbol{f}_1 \in \mathcal{FF}(W_1, L_1, R^d \to \mathbb{R})$ such that for any $i \in [n]$ and $k \in [N]$

$$\boldsymbol{f}_1(\boldsymbol{X}^{(i)}_{:,k}) = \begin{cases} \boldsymbol{h}(\boldsymbol{X}^{(i)}_{:,k}) & \boldsymbol{X}^{(i)}_{:,k} \in S, \\ 0 & \text{otherwise.} \end{cases}$$

If we let $C_1 = \lceil 2n^3 \sqrt{\pi} \rceil$, and $R_1 = 20r(nN)^2 \delta^{-1} \sqrt{\pi d}$, we have

$$W_1 = O(1), L_1 = O\left( \sqrt{n \log n} + \sqrt{\frac{n}{\log n}} \cdot \max\{\log R_1, \log C_1\} \right) = \tilde{O}(\sqrt{n}).$$

In the following, we verify that $\boldsymbol{f}_1$ can represent the information of the input sequences. For any $i, j \in [n]$ with $i \neq j$, we have

$$\left| \sum_{k=1}^N \boldsymbol{f}_1(\boldsymbol{X}^{(i)}_{:,k}) - \sum_{k=1}^N \boldsymbol{f}_1(\boldsymbol{X}^{(j)}_{:,k}) \right| = \left| \sum_{1 \leq k \leq N, \boldsymbol{X}^{(i)}_{:,k} \in S} \boldsymbol{h}(\boldsymbol{X}^{(i)}_{:,k}) - \sum_{1 \leq k \leq N, \boldsymbol{X}^{(j)}_{:,k} \in S} \boldsymbol{h}(\boldsymbol{X}^{(j)}_{:,k}) \right| \tag{D.11}$$

$$\geq \sqrt{|S|} \geq 1, \tag{D.12}$$

where the last inequality if from Eq.D.4. Up to now, we have already constructed a feed-forward neural nework $\boldsymbol{f}_1$, which aims to capture the information of the whole sequence. In the following, we show that we also can construct a feed-forward neural network $\boldsymbol{f}_2$ that can remain the information of individual tokens. Let $\mathcal{V}$ denote the set that contains all the tokens appearing in the dataset, that is, $\mathcal{V} = \{\boldsymbol{X}^{(i)}_{:,k} : i \in [n], k \in [N]\}$. According to Assumption 3.1, we know that there exists $r \geq 1$ and $0 < \delta \leq 1$ such that for any $\boldsymbol{x}_i, \boldsymbol{x}_j \in \mathcal{V}$ with $i \neq j$, we have $\|\boldsymbol{x}_i - \boldsymbol{x}_j\|_2 \geq \delta$ and $\|\boldsymbol{x}_i\|_2 \leq r$. Based on this, we apply Lemma E.4 to $\mathcal{V}$ and gain a feed-forward neural network $\boldsymbol{f}_2 \in \mathcal{FF}(O(1), O(1), \mathbb{R}^d \to \mathbb{R})$ such that

$$0 \leq \boldsymbol{f}_2(\boldsymbol{X}^{(i)}_{:,k}) \leq 10r(nN)^2 \delta^{-1} \sqrt{\pi d},$$

for any $i \in [n]$ and $k \in [N]$, and

$$\left| \boldsymbol{f}_2(\boldsymbol{X}^{(i)}_{:,k}) - \boldsymbol{f}_2(\boldsymbol{X}^{(j)}_{:,l}) \right| \geq 2, \tag{D.13}$$

for any $i, j \in [n]$ and $k, l \in [N]$ with $\boldsymbol{X}^{(i)}_{:,k} \neq \boldsymbol{X}^{(j)}_{:,l}$. Then, we consider integrate $\boldsymbol{f}_1$ and $\boldsymbol{f}_2$ into one single feed-forward neural network, which can both capture the information of the sequence but also that of the tokens. Besides, we also augment the output dimension by one more and pad it by $0$, which is used as a temporary memory. Let $\boldsymbol{F}_1 : \mathbb{R}^d \to \mathbb{R}^3$, which takes the input $\boldsymbol{x} \in \mathbb{R}^d$ and outputs

$$\boldsymbol{F}_1(\boldsymbol{x}) = [\boldsymbol{f}_1(\boldsymbol{x}), \boldsymbol{f}_2(\boldsymbol{x}), 0]^\top .$$

Since we know that

$$\boldsymbol{f}_1 \in \mathcal{FF}(O(1), \tilde{O}(\sqrt{n}), \mathbb{R}^d \to \mathbb{R}),$$
$$\boldsymbol{f}_2 \in \mathcal{FF}(O(1), O(1), \mathbb{R}^d \to \mathbb{R}),$$

which means that $\boldsymbol{F}_1 \in \mathcal{FF}(O(1), \tilde{O}(\sqrt{n}), \mathbb{R}^d \to \mathbb{R}^3)$. We assume that $d \geq 3$. By applying Lemma 3.1 to $\boldsymbol{F}_1$, we know that there exists a Transformer $\boldsymbol{T}^{(1)}_1 \in \mathcal{T}(D^{(1)}_{in}, D^{(1)}_{out}, D^{(1)}_{hid}, H^{(1)}, L^{(1)})$, and prompt $\boldsymbol{P}^{(1)} \in \mathbb{R}^{D^{(1)}_{in} \times M^{(1)}}$, $K^{(1)}_1, K^{(1)}_2 \in \mathbb{N}^+$ such that

$$\hat{\boldsymbol{T}}^{(1)}_{\boldsymbol{P}^{(1)}, K^{(1)}_1, K^{(1)}_2}(\boldsymbol{X}^{(i)}) = \begin{pmatrix} \boldsymbol{F}_1(\boldsymbol{X}^{(i)}_{:,1})^\top & \boldsymbol{F}_1(\boldsymbol{X}^{(i)}_{:,2})^\top & \cdots & \boldsymbol{F}_1(\boldsymbol{X}^{(i)}_{:,N})^\top \\ 1 & 1 & \cdots & 1 \end{pmatrix}$$

for any $i \in [n]$. We have $D^{(1)}_{in} = O(d)$, $D^{(1)}_{out} = O(1)$, $D^{(1)}_{hid} = O(D^{(1)}_{in})$, $H^{(1)} = O(1)$, $L^{(1)} = O(1)$, and $M^{(1)} = \tilde{O}(\sqrt{n})$, $K^{(1)}_1, K^{(1)}_2 = \tilde{O}(N \cdot \sqrt{n})$. The construction of $\boldsymbol{T}_1$ is completed.

**Step 3: Construction of $\boldsymbol{T}^{(2)}$** In this step, we aim to construct Transformer $\boldsymbol{T}^{(2)}$ to integrate the information of whole the sequence. $\boldsymbol{T}^{(2)}$ consists of one self-attention layer and one feed-forward layer, that is, $\boldsymbol{T}^{(2)} = \mathcal{F}_{FF} \circ \mathcal{F}_{SA}$. Define

$$\boldsymbol{W}_Q = \boldsymbol{W}_K = \begin{pmatrix} 0 & 0 & 0 & 0 \\ 0 & 0 & 0 & 0 \\ 0 & 0 & 0 & 0 \\ 0 & 0 & 0 & 1 \end{pmatrix},$$

and

$$\boldsymbol{W}_V = \begin{pmatrix} 0 & 0 & 0 & 0 \\ 0 & 0 & 0 & 0 \\ 1 & 0 & 0 & 0 \\ 0 & 0 & 0 & 0 \end{pmatrix}.$$

As for the weight matrices in $\mathcal{F}_{FF}$, we let

$$\boldsymbol{W}_1 = \boldsymbol{W}_2 = \boldsymbol{0},$$

which means that $\mathcal{F}_{FF}$ is just an identity mapping with skip connection. Given any $\boldsymbol{X} = \begin{pmatrix} \boldsymbol{X}_{1,1} & \boldsymbol{X}_{1,2} & \cdots & \boldsymbol{X}_{1,N} \\ \boldsymbol{X}_{2,1} & \boldsymbol{X}_{2,2} & \cdots & \boldsymbol{X}_{2,N} \\ \vdots & \vdots & \cdots & 0 \\ 1 & 1 & \cdots & 1 \end{pmatrix} \in \mathbb{R}^{4 \times N}$ with the the 3-th element of each column being zero, and

the 4-th element being 1. We know that the output of $\boldsymbol{T}^{(2)}$ given input $\boldsymbol{X}$ is

$$
\boldsymbol{T}^{(2)}(\boldsymbol{X}) = \mathcal{F}_{FF} \circ \mathcal{F}_{SA}(\boldsymbol{X})
$$
$$
= \mathcal{F}_{SA}(\boldsymbol{X})
$$
$$
= \boldsymbol{X} + \begin{pmatrix} 0 & 0 & 0 & 0 \\ 0 & 0 & 0 & 0 \\ 1 & 0 & 0 & 0 \\ 0 & 0 & 0 & 0 \end{pmatrix} \boldsymbol{X} \begin{pmatrix} 1 & 1 & \cdots & 1 \\ 1 & 1 & \cdots & 1 \\ \vdots & \vdots & \vdots & \vdots \\ 1 & 1 & \cdots & 1 \end{pmatrix}
$$
$$
= \boldsymbol{X} + \begin{pmatrix} 0 & 0 & 0 & 0 \\ 0 & 0 & 0 & 0 \\ 1 & 0 & 0 & 0 \\ 0 & 0 & 0 & 0 \end{pmatrix} \left( \begin{pmatrix} \sum_{k\in[N]} \boldsymbol{X}_{1,k} \\ \sum_{k\in[N]} \boldsymbol{X}_{2,k} \\ 0 \\ n \end{pmatrix} \cdots \begin{pmatrix} \sum_{k\in[N]} \boldsymbol{X}_{1,k} \\ \sum_{k\in[N]} \boldsymbol{X}_{2,k} \\ 0 \\ n \end{pmatrix} \right)
$$
$$
= \boldsymbol{X} + \begin{pmatrix} 0 & 0 & \cdots & 0 \\ 0 & 0 & \cdots & 0 \\ \sum_{k\in[N]} \boldsymbol{X}_{1,k} & \sum_{k\in[N]} \boldsymbol{X}_{1,k} & \cdots & \sum_{k\in[N]} \boldsymbol{X}_{1,k} \\ 0 & 0 & \cdots & 0 \end{pmatrix}
$$
$$
= \begin{pmatrix} \boldsymbol{X}_{1,1} & \boldsymbol{X}_{1,2} & \cdots & \boldsymbol{X}_{1,N} \\ \boldsymbol{X}_{2,1} & \boldsymbol{X}_{2,2} & \cdots & \boldsymbol{X}_{2,N} \\ \sum_{k\in[N]} \boldsymbol{X}_{1,k} & \sum_{k\in[N]} \boldsymbol{X}_{1,k} & \cdots & \sum_{k\in[N]} \boldsymbol{X}_{1,k} \\ 1 & 1 & \cdots & 1 \end{pmatrix}
$$

In particular, let $\boldsymbol{s}_k^{(i)}$ denote the $k$-th column of the output of $\boldsymbol{T}^{(2)}$ given input $\hat{\boldsymbol{T}}^{(1)}_{\boldsymbol{P}^{(1)},K_1^{(1)},K_2^{(1)}}(\boldsymbol{X}^{(i)})$, which can be calculated as

$$
\boldsymbol{s}_k^{(i)} := \begin{pmatrix} \boldsymbol{f}_1(\boldsymbol{X}_{:,k}^{(i)}) \\ \boldsymbol{f}_2(\boldsymbol{X}_{:,k}^{(i)}) \\ \sum_{k\in[N]} \boldsymbol{f}_1(\boldsymbol{X}_{:,k}^{(i)}) \\ 1 \end{pmatrix}.
$$

We claim that $\boldsymbol{s}_k^{(i)}$ can help us to differentiate different tokens or the same tokens in different contexts. For any $i, j \in [n]$, and $k, l \in [N]$, if $\boldsymbol{X}_{:,k}^{(i)} \neq \boldsymbol{X}_{:,l}^{(j)}$, according to Eq.(D.13), we know that

$$
\left| \boldsymbol{f}_2(\boldsymbol{X}_{:,k}^{(i)}) - \boldsymbol{f}_2(\boldsymbol{X}_{:,l}^{(j)}) \right| \geq 2.
$$

Besides, for any $i \in [n]$, and $k \neq l \in [N]$, if $\boldsymbol{X}_{:,k}^{(i)} = \boldsymbol{X}_{:,l}^{(i)}$, we immediatetly have $\boldsymbol{s}_k^{(i)} = \boldsymbol{s}_l^{(i)}$. On the other hand, for any $i \neq j \in [n]$, according to assumption 3.1, we have $\boldsymbol{X}^{(i)} \neq \boldsymbol{X}^{(j)}$ under any permutation. Then, Eq.(D.11) implies

$$
\left| \sum_{k=1}^{n} \boldsymbol{f}_1(\boldsymbol{X}_{:,k}^{(i)}) - \sum_{k=1}^{n} \boldsymbol{f}_1(\boldsymbol{X}_{:,k}^{(i)}) \right| \geq 1.
$$

Thus, by incoporating the above three cases, the difference between any two arbitrary columns of the output of $\boldsymbol{T}^{(2)}$ has the following form

$$
\left\| \boldsymbol{s}_k^{(i)} - \boldsymbol{s}_l^{(j)} \right\|_2 = \begin{cases} \geq 2 & \boldsymbol{X}_{:,k}^{(i)} \neq \boldsymbol{X}_{:,l}^{(j)}, \\ 0 & i = j, \ \boldsymbol{X}_{:,k}^{(i)} = \boldsymbol{X}_{:,l}^{(j)}, \\ \geq 1 & i \neq j, \ \boldsymbol{X}_{:,k}^{(i)} = \boldsymbol{X}_{:,l}^{(j)}, \end{cases}
$$

where we use the basic fact

$$\left\| \boldsymbol{s}_{:,k}^{(i)} - \boldsymbol{s}_{:,l}^{(j)} \right\|_2 = \left\| \begin{pmatrix} \boldsymbol{f}_1(\boldsymbol{X}_{:,k}^{(i)}) \\ \boldsymbol{f}_2(\boldsymbol{X}_{:,k}^{(i)}) \\ \sum_{t\in[N]} \boldsymbol{f}_1(\boldsymbol{X}_{:,t}^{(i)}) \\ 1 \end{pmatrix} - \begin{pmatrix} \boldsymbol{f}_1(\boldsymbol{X}_{:,l}^{(j)}) \\ \boldsymbol{f}_2(\boldsymbol{X}_{:,l}^{(j)}) \\ \sum_{t\in[N]} \boldsymbol{f}_1(\boldsymbol{X}_{:,t}^{(i)}) \\ 1 \end{pmatrix} \right\|_2$$

$$\geq \min \left\{ \left| \boldsymbol{f}_2(\boldsymbol{X}_{:,k}^{(i)}) - \boldsymbol{f}_2(\boldsymbol{X}_{:,l}^{(j)}) \right|, \left| \sum_{t\in[N]} \boldsymbol{f}_1(\boldsymbol{X}_{:,t}^{(i)}) - \sum_{t\in[N]} \boldsymbol{f}(\boldsymbol{X}_{:,t}^{(j)}) \right| \right\}.$$

As for the upper bound of the norm of $\boldsymbol{s}_k^{(i)}$, we can compute it as

$$\left| \boldsymbol{s}_{:,n}^{(i)} \right\|_2 = \left\| \begin{pmatrix} \boldsymbol{f}_1(\boldsymbol{X}_{:,k}^{(i)}) \\ \boldsymbol{f}_2(\boldsymbol{X}_{:,k}^{(i)}) \\ \sum_{t\in[N]} \boldsymbol{f}_1(\boldsymbol{X}_{:,t}^{(i)}) \\ 1 \end{pmatrix} \right\|_2$$

$$\leq \left| \boldsymbol{f}_1(\boldsymbol{X}_{:,k}^{(i)}) \right| + \left| \boldsymbol{f}_2(\boldsymbol{X}_{:,k}^{(i)}) \right| + \left| \sum_{t\in[N]} \boldsymbol{f}_2(\boldsymbol{X}_{:,t}^{(i)}) \right|$$

$$\leq \lceil 2n^3\sqrt{\pi} \rceil + 10r(nN)^2\delta^{-1}\sqrt{\pi d} + n \cdot \lceil 2n^3\sqrt{\pi} \rceil + 1$$

$$\leq 21rn^4N^2\delta^{-1}\sqrt{\pi d}.$$

It we let $\boldsymbol{P}^{(2)} \in \mathbb{R}^{D_{in}^{(2)} \times M^{(2)}}$, where $M^{(2)} = 0$, and $K_1^{(2)} = K_2^{(2)} = 0$ we have

$$\hat{\boldsymbol{T}}_{\boldsymbol{P}^{(2)}, K_1^{(2)}, K_2^{(2)}}^{(2)} \circ \hat{\boldsymbol{T}}_{\boldsymbol{P}^{(1)}, K_1^{(1)}, K_2^{(1)}}^{(3)} (\boldsymbol{X}^{(i)})_{:,k} = \boldsymbol{s}_k^{(i)} \quad \text{for any } i \in [n],$$

which completes the construction of $\boldsymbol{T}^{(2)}$.

**Step 3: Construction of $\boldsymbol{T}^{(3)}$**   In this step, we construct $\boldsymbol{T}^{(3)}$ to map each $\boldsymbol{s}_k^{(i)}$ to its corresponding label $\boldsymbol{y}_{:,k}^{(i)}$. Let $R_3 = 20 \cdot 21rn^4N^2\delta^{-1}\sqrt{\pi d} \cdot (nN)^2 \cdot 1 \cdot \sqrt{\pi d}$. By applying Lemma E.2 to input-output pairs $(\boldsymbol{s}_k^{(i)}, \boldsymbol{y}_{:,k}^{(i)})$ for $i \in [n]$ and $k \in [N]$, there exists a feed-forward neural network $\boldsymbol{f}_3 : \mathbb{R}^4 \to \mathbb{R}$ with width $W_3 = O(1)$, and depth $L_3 = $

$$O(\sqrt{nN \log(nN)} + \sqrt{\frac{nN}{\log nN}} \cdot \max\{\log R_3, \log C\}) = \tilde{O}(\sqrt{nN}),$$

such that

$$\boldsymbol{f}_3(\boldsymbol{s}_k^{(i)}) = \boldsymbol{y}_{:,k}^{(i)} \quad \text{for any } i \in [n], k \in [N].$$

Then, we apply Lemma 3.1 to $\boldsymbol{f}_3$ and gain a Transformer $\boldsymbol{T}^{(3)} \in \mathcal{T}(D_{in}^{(3)}, D_{out}^{(3)}, D_{hid}^{(3)}, H^{(3)}, L^{(3)})$, and prompt $\boldsymbol{P}^{(3)} \in \mathbb{R}^{D_{in}^{(3)} \times M^{(3)}}, K_1^{(3)}, K_2^{(3)} \in \mathbb{N}^+$ such that

$$\hat{\boldsymbol{T}}_{\boldsymbol{P}^{(3)}, K_1^{(3)}, K_2^{(3)}}^{(3)} \left( \left[ \boldsymbol{s}_{:,1}^{(i)}, \cdots, \boldsymbol{s}_{:,N}^{(i)} \right] \right) = \boldsymbol{y}^{(i)},$$

for any $i \in [n]$, where $D_{in}^{(3)} = O(1), D_{out}^{(3)} = O(1), D_{hid}^{(3)} = O(D_{in}^{(3)}), H^{(3)} = O(1), L^{(3)} = O(1)$, and $M^{(3)} = \tilde{O}(\sqrt{nN}), K_1^{(3)}, K_2^{(3)} = \tilde{O}(N \cdot \sqrt{nN})$.

**Step 4: Put every thing together**   Based on our analysis above, we have proved that There exist $\boldsymbol{T} = \boldsymbol{T}^{(3)} \circ \boldsymbol{T}^{(2)} \circ \boldsymbol{T}^{(1)}$ with

$$\boldsymbol{T}^{(1)} \in \mathcal{T}(O(d), O(1), O(d), O(1), O(1)),$$
$$\boldsymbol{T}^{(2)} \in \mathcal{T}(O(1), O(1), O(1), O(1), O(1)),$$
$$\boldsymbol{T}^{(3)} \in \mathcal{T}(O(1), O(1), O(1), O(1), O(1)),$$

and

$$\boldsymbol{P}^{(1)} \in \mathbb{R}^{O(d) \times \tilde{O}(\sqrt{n})},$$
$$\boldsymbol{P}^{(2)} \in \mathbb{R}^{O(1) \times O(1)},$$
$$\boldsymbol{P}^{(3)} \in \mathbb{R}^{O(1) \times \tilde{O}(\sqrt{nN})},$$

and

$$K_1^{(1)}, K_2^{(1)} = \tilde{O}(N \cdot \sqrt{n}),$$
$$K_1^{(2)}, K_2^{(2)} = O(1),$$
$$K_1^{(3)}, K_2^{(3)} = \tilde{O}(N \cdot \sqrt{nN})$$

such that

$$\hat{\boldsymbol{T}}^{(3)}_{\boldsymbol{P}^{(3)},K_1^{(3)},K_2^{(3)}} \circ \hat{\boldsymbol{T}}^{(2)}_{\boldsymbol{P}^{(2)},K_1^{(2)},K_2^{(2)}} \circ \hat{\boldsymbol{T}}^{(1)}_{\boldsymbol{P}^{(1)},K_1^{(1)},K_2^{(1)}}(\boldsymbol{X}^{(i)}) = \boldsymbol{y}^{(i)} \quad \text{for any } i \in [n],$$

which completes the proof of Theorem 3.1. $\qquad\square$

## D.5 The Limitation of a Single ReLU Self-attention Layer

In Theorem 3.1, the activation function used in each self-attention layer is ReLU. Although ReLU-based Transformers are not commonly utilized in practice, empirical studies [Shen et al., 2023, Wortsman et al., 2023] have demonstrated that, with appropriate normalization techniques, they can achieve competitive performance across a range of tasks. The Softmax function converts pairwise dot products into strictly positive attention weights, enabling each token to attend to all others in the absence of explicit masking. However, for theoretical analysis, it is often necessary to restrict attention to specific tokens. To achieve such deterministic token interactions, existing studies typically replace Softmax with Hardmax function or constrain it to perform column averaging operation. In contrast, since ReLU function zeroes out all negative inputs, it provides a more explicit mechanism to control token interactions (see details in [Nakada et al., 2025]). Nevertheless, this does not imply that ReLU-based self-attention is theoretically more expressive than its Softmax-based counterpart.

In this following, we study the limitation of a single layer ReLU self-attention to distinguish the same token in different contexts. Specifically, we want to know for any sequential inputs $\boldsymbol{X}^{(1)}, \boldsymbol{X}^{(2)} \in \mathbb{R}^{d \times N}$ with $\boldsymbol{X}^{(1)}_{:,k} = \boldsymbol{X}^{(j)}_{:,l}$ for some $k, l \in [N]$, and $\boldsymbol{X}^{(1)} \neq \boldsymbol{X}^{(2)}$ under any column permutation, whether we can find a ReLU self-attention layer $\boldsymbol{\mathcal{F}}_{SA}$ such that

$$\boldsymbol{\mathcal{F}}_{SA}(\boldsymbol{X}^{(1)})_{:,k} \neq \boldsymbol{\mathcal{F}}_{SA}(\boldsymbol{X}^{(2)})_{:,l}.$$

According to Theorem 2 in [Kajitsuka and Sato, 2023], a single Softmax self-attention layer is able to achieve this property. However, in the following, we construct a counterexample to show there exist $\boldsymbol{X}^{(1)} \neq \boldsymbol{X}^{(2)}$ with $\boldsymbol{X}^{(1)}_{:,2} = \boldsymbol{X}^{(2)}_{:,2}$ such that any single ReLU self-attention layer can not differentiate the second token in $\boldsymbol{X}^{(1)}$ and $\boldsymbol{X}^{(2)}$.

Fix any $d \in \mathbb{N}^+$. Let $\boldsymbol{X}^{(1)} = (\alpha_1 \boldsymbol{v}, \alpha_2 \boldsymbol{v}, \alpha_3 \boldsymbol{v}) \in \mathbb{R}^{d \times 3}$ and $\boldsymbol{X}^{(2)} = (\alpha_4 \boldsymbol{v}, \alpha_2 \boldsymbol{v}, \alpha_5 \boldsymbol{v})$, with $\boldsymbol{v} \in \mathbb{R}^d$ and $\alpha_1, \alpha_2, \alpha_3, \alpha_4, \alpha_5 \in \mathbb{R}_{>0}$. Let $\sigma_R$ denote the ReLU function and $\boldsymbol{\mathcal{F}}_{SA}$ be an arbitrary single self-attention layer with $H$ heads, which has the following form,

$$\boldsymbol{\mathcal{F}}_{SA}(\boldsymbol{X}) := \boldsymbol{X} + \sum_{i=1}^{H} \boldsymbol{W}_V^{(i)} \boldsymbol{X} \sigma_R \left[ (\boldsymbol{W}_K^{(i)} \boldsymbol{X})^T (\boldsymbol{W}_Q^{(i)} \boldsymbol{X}) \right] \in \mathbb{R}^{D_{in} \times N}.$$

Through direct verification and according to the definition of ReLU function, the second column of the outputs of $\mathcal{F}_{\text{SA}}$ given input $\boldsymbol{X}^{(1)}$ and $\boldsymbol{X}^{(2)}$ are

$$\boldsymbol{\mathcal{F}}_{\text{SA}}(\boldsymbol{X}^{(1)})_{:,2} = \boldsymbol{X}^{(1)}_{:,2} + \sum_{i=1}^{H} \left( \alpha_2 \alpha_1^2 \sigma_R \langle \boldsymbol{W}_Q^{(i)} \boldsymbol{v}, \boldsymbol{W}_K^{(i)} \boldsymbol{v} \rangle \boldsymbol{W}_V^{(i)} \boldsymbol{v} + \alpha_2 \alpha_3^2 \sigma_R (\langle \boldsymbol{W}_Q^{(i)} \boldsymbol{v}, \boldsymbol{W}_K^{(i)} \boldsymbol{v} \rangle) \boldsymbol{W}_V^{(i)} \boldsymbol{v} \right.$$
$$\left. + \alpha_2^3 \sigma_R (\langle \boldsymbol{W}_Q^{(i)} \boldsymbol{v}, \boldsymbol{W}_K^{(i)} \boldsymbol{v} \rangle) \boldsymbol{W}_V^{(i)} \boldsymbol{v} \right)$$
$$= \sum_{i=1}^{H} \left( \alpha_2 (\alpha_1^2 + \alpha_2^2 + \alpha_3^2) \right) \sigma_R (\langle \boldsymbol{W}_Q^{(i)} \boldsymbol{v}, \boldsymbol{W}_K^{(i)} \boldsymbol{v} \rangle) \boldsymbol{W}_V^{(i)} \boldsymbol{v},$$

$$\boldsymbol{\mathcal{F}}_{SA}(\boldsymbol{X}^{(2)})_{:,2} = \boldsymbol{X}^{(2)}_{:,2} + \sum_{h=1}^{H} \left( \alpha_2 \alpha_4^2 \sigma_R (\langle \boldsymbol{W}_Q^{(i)} \boldsymbol{v}, \boldsymbol{W}_K^{(i)} \boldsymbol{v} \rangle) \boldsymbol{W}_V^{(i)} \boldsymbol{v} + \alpha_2 \alpha_5^2 \sigma_R (\langle \boldsymbol{W}_Q^{(i)} \boldsymbol{v}, \boldsymbol{W}_K^{(i)} \boldsymbol{v} \rangle) \boldsymbol{W}_V^{(i)} \boldsymbol{v} \right.$$
$$\left. + \alpha_2^3 \sigma_R (\langle \boldsymbol{W}_Q^{(i)} \boldsymbol{v}, \boldsymbol{W}_K^{(i)} \boldsymbol{v} \rangle) \boldsymbol{W}_V^{(i)} \boldsymbol{v} \right)$$
$$= \sum_{h=1}^{H} \left( \alpha_2 (\alpha_4^2 + \alpha_2^2 + \alpha_5^2) \right) \sigma_R (\langle \boldsymbol{W}_Q^{(i)} \boldsymbol{v}, \boldsymbol{W}_K^{(i)} \boldsymbol{v} \rangle) \boldsymbol{W}_V^{(i)} \boldsymbol{v}.$$

If $\alpha_1^2 + \alpha_3^2 = \alpha_4^2 + \alpha_5^2$, we always have $\boldsymbol{\mathcal{F}}_{SA}(\boldsymbol{X}^{(1)})_{:,2} = \boldsymbol{\mathcal{F}}_{SA}(\boldsymbol{X}^{(2)})_{:,2}$, even if $(\alpha_1, \alpha_2, \alpha_3) \neq (\alpha_4, \alpha_2, \alpha_5)$.

## E  Supporting Lemmas

**Lemma E.1** (Theorem C.2 in [Nakada et al., 2025]). *For any $L$ layer ReLU feed-forward neural network $\boldsymbol{f}$, which has the following form*

$$\boldsymbol{f}(\boldsymbol{x}) = \boldsymbol{\mathcal{L}}_L \circ \sigma_R \circ \boldsymbol{\mathcal{L}}_{L-1} \circ \cdots \boldsymbol{\mathcal{L}}_1 \circ \sigma_R \circ \boldsymbol{\mathcal{L}}_0(\boldsymbol{x}) \quad \text{for any } \boldsymbol{x} \in \mathbb{R}^d,$$

*where $\boldsymbol{\mathcal{L}}_\ell(\boldsymbol{x}) := \boldsymbol{W}_\ell \boldsymbol{x}$ with $\boldsymbol{W}_\ell \in \mathbb{R}^{d \times d}$. There exists a Transformer $\boldsymbol{T} \in \mathcal{T}(D_{in}, D_{out}, D_{hid}, H, L)$ with all the parameters only depend on $d$, and $D_{in} = O(d)$, $D_{out} = O(d)$, $D_{hid} = O(D_{in})$, $H = O(1)$, $L = O(1)$ satisfying: for any $n$ data points $\{\boldsymbol{x}_i\}_{i=1}^n \subset [0,1]^d$, there exists a prompt $\boldsymbol{P} \in \mathbb{R}^{D_{in} \times M}$ such that*

$$\hat{\boldsymbol{T}}_{P, K_i, K_i} = \boldsymbol{f}(\boldsymbol{x}_i),$$

*where $K_i = n \cdot (L+1) + i$ for $i \in [n]$, and $M = O(\sum_{\ell=0}^{L} \text{rank}(\boldsymbol{W}_\ell)) \leq O(d \cdot (L+1))$.*

**Lemma E.2** (Lemma C.1, [Kajitsuka and Sato, 2024]). *Let $n, m, d, c \in \mathbb{N}^+$ with $n \leq m$, and $r \geq 1$, $0 < \delta \leq 1$. Let $y^{(1)}, \cdots, y^{(n)} \in [C]$ be a set of $n$ labels, and $\boldsymbol{x}^{(1)}, \cdots, \boldsymbol{x}^{(m)} \in \mathbb{R}^d$ be a set of $m$ inputs such that $\boldsymbol{x}^{(i)} \in [0,1]^d$ with $\|\boldsymbol{x}^{(i)}\|_2 \leq r$ for any $i \in [m]$, and $\|\boldsymbol{x}^{(i)} - \boldsymbol{x}^{(j)}\|_2 \geq \delta$ for any $i, j \in [m]$ with $i \neq j$. Denote $R := 20rm^2\delta^{-1}\sqrt{\pi d}$. Then, there exists a feed-forward neural network $\boldsymbol{f} : \mathbb{R}^d \to \mathbb{R}$ with width $W = O(1)$, depth $L =$*

$$O\left( \sqrt{n \log n} + \sqrt{\frac{n}{\log n}} \cdot \max\{\log(R), \log(C)\} \right),$$

*such that $\boldsymbol{f}(\boldsymbol{x}^{(i)}) = y^{(i)}$ for every $i \in [n]$, and $\boldsymbol{f}(\boldsymbol{x}^{(i)}) = 0$ for any $i \in [m] \backslash [n]$.*

**Lemma E.3.** *Let $n, m, d, C \in \mathbb{N}^+$ with $n \leq m$, and $r \geq 1$, $0 < \delta \leq 1$. Let $y^{(1)}, \cdots, y^{(n)}$ be a set of $n$ labels and $\boldsymbol{x}^{(1)}, \cdots, \boldsymbol{x}^{(m)} \in \mathbb{R}^d$ be a set of $m$ inputs such that $\boldsymbol{x}^{(i)} \in [0,1]^d$ with $\|\boldsymbol{x}^{(i)}\|_2 \leq r$ for any $i \in [m]$, and $\|\boldsymbol{x}^{(i)} - \boldsymbol{x}^{(j)}\|_2 \geq \delta$ for any $i, j \in [m]$ with $i \neq j$. Denote $R := 20rm^2\delta^{-1}\sqrt{\pi d}$. Then, there exsits a feed-forward neural network $\boldsymbol{f} : \mathbb{R}^d \to \mathbb{R}$ with width $W = O(n)$, depth $L =$*

$$O(\max\{\log R, \log C\}),$$

*such that $\boldsymbol{f}(\boldsymbol{x}^{(i)}) = y^{(i)}$ for every $i \in [n]$, and $\boldsymbol{f}(\boldsymbol{x}^{(i)}) = 0$ for every $i \in [m] \backslash [n]$.*

*Proof of Lemma E.3.* In Lemma E.2, we notice that the width of the feed-forward neural network is a constant, while the depth depends on the number of data points to be memorized. This result can be transformed into the one where the width depends on the number of data points while depth is constant up to logarithmic factors. Firstly, by applying Lemma E.4 to $\{\boldsymbol{x}^{(i)}\}_{i=1}^{n}$, there exists a feed-forward neural network $\boldsymbol{f}$ with width and depth both $O(1)$ such that the inputs are mapped to $\mathbb{R}$ while approximately remains their original distance. Let $x^{(i)} \in \mathbb{R}$ denote $\boldsymbol{f}(\boldsymbol{x}^{(i)})$. Next, let $B \in [\lfloor\sqrt{n}\rfloor] \setminus \{1\}$ be an arbitrary integer. We divide data points $x^{(1)}, \cdots, x^{(n)}$ into $\frac{n}{B^2}$ subsets, each of which has a size of $B^2$. We denote these subsets as $\boldsymbol{I}_1, \cdots, \boldsymbol{I}_{\frac{n}{B^2}}$. We apply Lemma E.2 to these subsets respectively to get $\frac{n}{B^2}$ feed-forward neural networks $\boldsymbol{f}_1, \cdots, \boldsymbol{f}_{\frac{n}{B^2}}$, each of which satisfies

$$
\boldsymbol{f}_i(x^{(j)}) = \begin{cases} y^{(j)} & x^{(j)} \in \boldsymbol{I}_i, \\ 0 & \text{Otherwise.} \end{cases}
$$

Let $\boldsymbol{F}$ denote the concatenation of $\boldsymbol{f}_1 \circ \boldsymbol{f}, \cdots, \boldsymbol{f}_{\frac{N}{B^2}} \circ \boldsymbol{f}$, which takes input $\boldsymbol{x}^{(i)}$ and outputs

$$
\boldsymbol{F}(\boldsymbol{x}^{(i)}) = \begin{pmatrix} \boldsymbol{f}_1(x^{(i)}) \\ \vdots \\ \boldsymbol{f}_{\frac{n}{B^2}}(x^{(i)}) \end{pmatrix}.
$$

There is only one entry of $\boldsymbol{F}(\boldsymbol{x}^{(i)})$ not zero and equals to $y^{(i)}$. It is clear that $\boldsymbol{f} \in \mathcal{FF}(O(1), O(1), \mathbb{R}^d \to \mathbb{R})$, and $\boldsymbol{f}_i \in \mathcal{FF}(O(1), O(\max\{\log R, \log C\}), \mathbb{R} \to \mathbb{R})$ for any $i \in [\frac{n}{B^2}]$. Then, we know that

$$
\boldsymbol{F} \in \mathcal{FF}(O(n), O(\max\{\log R, \log C\}), \mathbb{R}^d \to \mathbb{R}),
$$

which completes the proof by letting $\boldsymbol{f} = \boldsymbol{F}$. $\qquad\square$

**Lemma E.4** (Lemma A.2, [Vardi et al., 2021]). *Let $n \in \mathbb{N}^+$, and $r \geq 1$, $0 < \delta \leq 1$. Let $\boldsymbol{x}^{(1)}, \cdots, \boldsymbol{x}^{(n)} \in \mathbb{R}^d$ be $n$ inputs such that $\boldsymbol{x}^{(i)} \in [0,1]^d$ with $\|\boldsymbol{x}^{(i)}\|_2 \leq r$ for any $i \in [m]$, and $\|\boldsymbol{x}^{(i)} - \boldsymbol{x}^{(j)}\|_2 \geq \delta$ for any $i, j \in [m]$ with $i \neq j$. Then, there exists a feed-forward neural network $\boldsymbol{f} \in \mathcal{FF}(O(1), O(1), \mathbb{R}^d \to \mathbb{R})$, such that $0 \leq \boldsymbol{f}(\boldsymbol{x}^{(i)}) \leq 10rn^2\delta^{-1}\sqrt{\pi d}$ for every $i \in [N]$ and $\left|\boldsymbol{f}(\boldsymbol{x}^{(i)}) - \boldsymbol{f}(\boldsymbol{x}^{(j)})\right| \geq 2$ for every $i, j \in [n]$ with $i \neq j$.*

**Lemma E.5** ([Park et al., 2021]). *Let $d \in \mathbb{N}^+$. Then, for any finite subset $S \subset \mathbb{R}^d$, there exists a unit vector $\boldsymbol{v} \in \mathbb{R}^d$ such that*

$$
\frac{1}{|S|^2}\sqrt{\frac{8}{\pi d}} \|\boldsymbol{x} - \boldsymbol{x}'\|_2 \leq \left|\boldsymbol{v}^\top(\boldsymbol{x} - \boldsymbol{x}')\right| \leq \|\boldsymbol{x} - \boldsymbol{x}'\|_2
$$

*holds for any $\boldsymbol{x}, \boldsymbol{x}' \in S$.*

# F Prompt Tuning Transformers to Exactly Implement Residual Feed-Forward Neural Networks

Residual feed-forward neural network was proposed by [He et al., 2016], which is widely used existing works [Yun et al., 2019] as a substitute of non-residual feed-forward neural networks. We define the class of residual neural networks as

$$
\mathcal{R}(W, L, \mathbb{R}^d \to \mathbb{R}^d) := \left\{ \boldsymbol{f} : \boldsymbol{f} = \boldsymbol{F}_L \circ \boldsymbol{F}_{L-1} \circ \cdots \circ \boldsymbol{F}_1(\boldsymbol{x}), \boldsymbol{x} \in \mathbb{R}^d \right\}.
$$

There are three parameters to describe a residual neural network. $d$ is the input dimension and the output dimension. $F_i(\boldsymbol{x}) := \boldsymbol{x} + \boldsymbol{W}_i^2 \sigma_R(\boldsymbol{W}_i^1 \boldsymbol{x} + \boldsymbol{b}_i^1) + \boldsymbol{b}_i^2$, where $\boldsymbol{W}_i^1 \in \mathbb{R}^{W \times d}$, $\boldsymbol{b}_i^1 \in \mathbb{R}^W$ and $\boldsymbol{W}_i^2 \in \mathbb{R}^{d \times W}$, $\boldsymbol{b}_i^2 \in \mathbb{R}^d$. $\sigma_R$ represents the element-wise ReLU function. $W$ is called the width of $\mathcal{R}$ and $L$ is called the depth. We provide a lemma that shows any residual feed-forward neural network can be realized by a non-residual one.

**Lemma F.1.** *For any residual neural network $\boldsymbol{g} \in \mathcal{R}(W, L, \mathbb{R}^d \to \mathbb{R}^d)$ with some $d, W, L \in \mathbb{N}^+$, then $\boldsymbol{g} \in \mathcal{FF}(W + 2d, L, \mathbb{R}^d \to \mathbb{R}^d)$.*

*Proof.* Our proof basically follows [Jiao et al., 2025a]. Given any residual neural network $\boldsymbol{g} \in \mathcal{R}(W, L, \mathbb{R}^d \to \mathbb{R}^d)$, which can be written as $\boldsymbol{g}(\boldsymbol{x}) = \boldsymbol{F}_L \circ \cdots \circ \boldsymbol{F}_1(\boldsymbol{x})$. Firstly, without loss of generality, we show that ReLU feed-forward neural networks can implement $\boldsymbol{F}_1$. We define

$$\boldsymbol{W}_1 = \begin{pmatrix} \boldsymbol{W}_1^1 \\ \boldsymbol{I}_{d \times d} \\ -\boldsymbol{I}_{d \times d} \end{pmatrix} \in \mathbb{R}^{(W+2d) \times d}, \quad \boldsymbol{b}_1 = \begin{pmatrix} \boldsymbol{b}_1^1 \\ \boldsymbol{0} \\ \boldsymbol{0} \end{pmatrix} \in \mathbb{R}^{W+2d},$$

$$\boldsymbol{W}_2 = \begin{pmatrix} \boldsymbol{W}_1^2 & \boldsymbol{I}_{d \times d} & -\boldsymbol{I}_{d \times d} \end{pmatrix} \in \mathbb{R}^{d \times (W+2d)}, \boldsymbol{b}_2 = \boldsymbol{b}_1^2 \in \mathbb{R}^d.$$

It is straightforward to verify that

$$\boldsymbol{W}_2 \sigma_R(\boldsymbol{W}_1 \boldsymbol{x} + \boldsymbol{b}_1) + \boldsymbol{b}_2 = \boldsymbol{W}_2 \begin{pmatrix} \sigma_R(\boldsymbol{W}_1^1 \boldsymbol{x} + \boldsymbol{b}_1) \\ \sigma_R(\boldsymbol{x}) \\ \sigma_R(-\boldsymbol{x}) \end{pmatrix} + \boldsymbol{b}_2$$

$$= \boldsymbol{W}_1^2(\sigma_R(\boldsymbol{W}_1^1 \boldsymbol{x} + \boldsymbol{b}_1^1)) + \sigma_R(\boldsymbol{x}) - \sigma_R(-\boldsymbol{x}) + \boldsymbol{b}_2$$

$$= \boldsymbol{x} + \boldsymbol{W}_1^2 \sigma_R(\boldsymbol{W}_1^1 \boldsymbol{x} + \boldsymbol{b}_1^1) + \boldsymbol{b}_1^2 = \boldsymbol{F}_1(\boldsymbol{x}).$$

where we use the fact that $\sigma_R(\boldsymbol{x}) - \sigma_R(-\boldsymbol{x}) = \boldsymbol{x}$. Let $\boldsymbol{f}_1$ denote $\boldsymbol{W}_2(\sigma_R(\boldsymbol{W}_1(\boldsymbol{x})) + \boldsymbol{b}_1) + \boldsymbol{b}_2$, which is a non-residual feed-forward neural network with width $W + 2d$ and depth 1. Similarly, we define $\boldsymbol{f}_i$ in the same manner which implements $\boldsymbol{F}_i$, respectively. In the last, By composing $\{\boldsymbol{f}_i\}_{i \in [L]}$, we have a feed-forward neural network $\boldsymbol{f} = \boldsymbol{f}_L \circ \boldsymbol{f}_{L-1} \circ \cdots \circ \boldsymbol{f}_1$, and the width of which is $W + 2d$, while the depth is $L$, that is, $\boldsymbol{f} \in \mathcal{FF}(W + 2d, L, \mathbb{R}^d \to \mathbb{R}^d)$. The proof is completed by pointing out that $\boldsymbol{f}(\boldsymbol{x}) = \boldsymbol{g}(\boldsymbol{x})$ for any $\boldsymbol{x} \in \mathbb{R}^d$. $\square$

We present the extension of Lemma 3.1, where the feed-forward neural network is replaced by a residual one.

**Lemma F.2.** *Fix any $W, d \in \mathbb{N}^+$. There exsits a Transformer $\boldsymbol{T} \in \mathcal{T}(D_{in}, D_{out}, D_{hid}, H, L)$ such that for any residual ReLU feed-forward neural network $\boldsymbol{g} \in \mathcal{R}(W, L, R^d \to R^d)$ for some $W, L, d \in \mathbb{N}^+$, and $n$ inputs $\boldsymbol{X}^{(1)}, \cdots, \boldsymbol{X}^{(n)} \subset \mathbb{R}^{d \times N}$. There exsits a prompt $\boldsymbol{P} \in \mathbb{R}^{D_{in} \times M}$, $K_1, K_2 \in \mathbb{N}^+$ such that*

$$\hat{\boldsymbol{T}}_{\boldsymbol{P}, K_1, K_2}(\boldsymbol{X}) = [\boldsymbol{g}(\boldsymbol{X}_{:,1}^{(i)}), \cdots, \boldsymbol{g}(\boldsymbol{X}_{:,N}^{(i)})] \quad \text{for any } i \in [n],$$

*where $D_{in} = O(W + 2d)$, $D_{out} = O(1)$, $D_{hid} = O(D_{in})$, $H = O(1)$, $L = O(1)$, and $M = O((W + d)L)$, $K_1, K_2 = O(NL)$.*

*Proof.* According to Proposition F.1, there exists a feed-forward ReLU neural network $\boldsymbol{f} \in \mathcal{FF}(W + 2d, L, \mathbb{R}^d \to \mathbb{R}^d)$ such that $\boldsymbol{f}(\boldsymbol{x}) = \boldsymbol{g}(\boldsymbol{x})$ for any $\boldsymbol{x} \in \mathbb{R}^d$. This proof is completed by applying Lemma 3.1 to $\boldsymbol{f}$. $\square$

## G   Data Memorization with Real Labels

In Section 3, we focus on integer labels, which can be regarded as a classification problem. It is easy to extend our results to real labels by adopting methods in existing literature. As proposed in [Vardi et al., 2021], when the output range is bounded, we can partition the output range into $\epsilon$-length intervals, each interval can be treated as a class. Then we reduce the problem to a data memorization with $O(\frac{1}{\epsilon})$ classes. However, this method can only achieve $\epsilon$-error instead of zero loss. Although [Hu et al., 2024] show that there exists a trade-off between the width and depth of neural networks, their results are established on $\epsilon$-error. In this section, we first show that it is easy to build a ReLU neural network with width $n$ and depth 1, which maps input vectors to real labels. Our proof is modified from [Jiao et al., 2025a], where we use a ReLU neural network instead of a residual neural network.

**Lemma G.1.** *Given any $d, n \in \mathbb{N}^+$. Let $(\boldsymbol{x}^{(1)}, y^{(1)}), \cdots, (\boldsymbol{x}^{(n)}, y^{(n)}) \subset \mathbb{R}^d \times [0, 1]$ be the input-label pairs with $\|\boldsymbol{x}^{(i)} - \boldsymbol{x}^{(j)}\|_2 \geq \delta$ for every $i \neq j \in [n]$ and $\|\boldsymbol{x}^{(i)}\|_2 \leq r$ for every $i \in [n]$. Then, there exists a feed-forward neural network $\boldsymbol{f} \in \mathcal{FF}(O(n), O(1), \mathbb{R}^d \to \mathbb{R})$ such that $\boldsymbol{f}(\boldsymbol{x}^{(i)}) = y^{(i)}$ for any $i \in [n]$.*

*Proof.* According to Lemma E.5, there exists $\boldsymbol{v} \in \mathbb{R}^d$ such that $\boldsymbol{v}^\top \boldsymbol{x}^{(i)}$ are distinct. Let $K > 0$ be determined later. We define

$$\boldsymbol{W}_i^{(1)} = K \begin{pmatrix} 1 \\ 1 \\ 1 \end{pmatrix} \boldsymbol{v}^\top, \quad \boldsymbol{b}_i^{(1)} = \begin{pmatrix} -K\boldsymbol{v}^\top \boldsymbol{x}^{(i)} - 1 \\ -K\boldsymbol{v}^\top \boldsymbol{x}^{(i)} \\ -K\boldsymbol{v}^\top \boldsymbol{x}^{(i)} + 1 \end{pmatrix}, \quad , \boldsymbol{W}_i^{(2)} = y^{(i)}(1, -2, 1), \quad \boldsymbol{b}_i^{(2)} = \boldsymbol{0}.$$

where $\boldsymbol{W}_i^{(1)} \in \mathbb{R}^{3 \times d}$, $\boldsymbol{b}_i^{(1)} \in \mathbb{R}^3$, $\boldsymbol{W}_i^{(2)} \in \mathbb{R}^{1 \times 3}$, and $\boldsymbol{b}_i^{(2)} \in \mathbb{R}$.

It is straightforward to verify that

$$\boldsymbol{W}_i^{(2)} \sigma_R(\boldsymbol{W}_i^{(1)} \boldsymbol{x} + \boldsymbol{b}_i^{(1)}) + \boldsymbol{b}_i^{(2)}$$
$$= \boldsymbol{y}^{(i)} \left( \sigma_R(K\boldsymbol{v}^\top (\boldsymbol{x} - \boldsymbol{x}^{(i)}) - 1) - 2\sigma_R(K\boldsymbol{v}^\top (\boldsymbol{x} - \boldsymbol{x}^{(i)})) + \sigma_R(K\boldsymbol{v}^\top (\boldsymbol{x} - \boldsymbol{x}^{(i)}) + 1) \right)$$
$$= \boldsymbol{y}^{(i)} I_i(\boldsymbol{x}),$$

where $I_i(\boldsymbol{x})$ satisfies $I_i(\boldsymbol{x}^{(i)}) = 1$ and $I_i(\boldsymbol{x}) = 0$ if $|\boldsymbol{v}^\top (\boldsymbol{x} - \boldsymbol{x}^{(i)})| \geq 1/K$. We choose $K > \frac{2}{\min_{i \neq j} |\boldsymbol{v}^\top (\boldsymbol{x}^{(j)} - \boldsymbol{x}^{(i)})|}$. Define

$$\boldsymbol{W}^{(1)} = \begin{pmatrix} \boldsymbol{W}_1^{(1)} \\ \vdots \\ \boldsymbol{W}_n^{(1)} \end{pmatrix}, \quad \boldsymbol{b}^{(1)} = \begin{pmatrix} \boldsymbol{b}_1^{(1)} \\ \vdots \\ \boldsymbol{b}_n^{(1)} \end{pmatrix}, \quad \boldsymbol{W}^{(2)} = (\boldsymbol{W}_1^{(2)}, \cdots, \boldsymbol{W}_n^{(2)}), \quad \boldsymbol{b}^{(2)} = 0,$$

where $\boldsymbol{W}^{(1)} \in \mathbb{R}^{3n \times d}$, $\boldsymbol{b}^{(1)} \in \mathbb{R}^{3n}$, $\boldsymbol{W}^{(2)} \in \mathbb{R}^{1 \times 3n}$, $\boldsymbol{b}^{(2)} \in \mathbb{R}$. Let

$$f(\boldsymbol{x}) = \boldsymbol{W}^{(2)} \sigma_R(\boldsymbol{W}^{(1)} \boldsymbol{x} + \boldsymbol{b}^{(1)}) + \boldsymbol{b}^{(2)}$$
$$= \sum_{i=1}^n \boldsymbol{y}^{(i)} I_i(\boldsymbol{x}).$$

The proof is completed by verifying that $f(\boldsymbol{x}^{(i)}) = y^{(i)}$ and $f \in \mathcal{FF}(O(n), O(1), \mathbb{R}^d \to \mathbb{R})$. $\qquad \square$

Next, we present the trade-off between the width and depth of the feed-froward neural networks that are constructed to memorize datasets with real labels. In [Yun et al., 2019], their results rely on piecewise linear functions which yield approximation error by ReLU function and [Hu et al., 2024] also face the same problem. Our following result is novel since it does not cause any extra error and exactly achieves the zero loss.

**Lemma G.2.** *Given any $d, n \in \mathbb{Z}^+$. Let $(\boldsymbol{x}^{(1)}, y^{(1)}), \cdots, (\boldsymbol{x}^{(n)}, y^{(n)}) \subset \mathbb{R}^d \times [0, 1]$ be the input-label pairs with $\|\boldsymbol{x}^{(i)} - \boldsymbol{x}^{(j)}\| \geq \delta$ for every $i \neq j \in [n]$ and $\|\boldsymbol{x}^{(i)}\| \leq r$ for every $i \in [n]$. Then, there exists a feed-forward neural network $\boldsymbol{f} \in \mathcal{FF}(O(1), O(n), \mathbb{R}^d \to \mathbb{R})$ such that $\boldsymbol{f}(\boldsymbol{x}^{(i)}) = y^{(i)}$ for any $i \in [n]$.*

*Proof of Lemma G.2.* According to Lemma E.4, there exists a feed-forward neural network $\boldsymbol{F} : \mathbb{R}^d \to \mathbb{R}$ with width $O(1)$ and depth $O(1)$ such that $|\boldsymbol{F}(\boldsymbol{x}^{(i)}) - \boldsymbol{F}(\boldsymbol{x}^{(j)})| \geq 2$ and $\boldsymbol{F}(\boldsymbol{x}^{(i)}) \geq 0$ for any $i \neq j \in [n]$. Let $x^{(i)}$ denote the output of $\boldsymbol{F}$ given input $\boldsymbol{x}^{(i)}$. Let $\boldsymbol{f}_i(x) := x + \boldsymbol{W}_2^{(i)} \sigma_R(\boldsymbol{W}_1^{(i)} x + \boldsymbol{b}_1^{(i)}) + \boldsymbol{b}_2^{(i)}$, where

$$\boldsymbol{W}_1^{(i)} = \begin{pmatrix} 1 \\ 1 \\ 1 \end{pmatrix}, \quad \boldsymbol{b}_1^{(i)} = \begin{pmatrix} -x^{(i)} - 1 \\ -x^{(i)} \\ -x^{(i)} + 1 \end{pmatrix}, \quad \boldsymbol{W}_2^{(i)} = (y^{(i)} - x^{(i)} - 4)(1, -2, 1), \quad \boldsymbol{b}_2^{(i)} = 0,$$

and $\boldsymbol{W}_1^{(i)} \in \mathbb{R}^{3 \times 1}, \boldsymbol{b}_1^{(i)} \in \mathbb{R}^3, \boldsymbol{W}_2^{(i)} \in \mathbb{R}^{1 \times 3}, \boldsymbol{b}_2^{(i)} \in \mathbb{R}$. It is direct to verify that

$$
\begin{aligned}
\boldsymbol{f}_i(x) &= x + \boldsymbol{W}_2^{(i)} \sigma_R(\boldsymbol{W}_1^{(i)} x + \boldsymbol{b}_1^{(i)}) + \boldsymbol{b}_2^{(i)} \\
&= x + (y^{(i)} - x^{(i)} - 4)\Big( \sigma_R(x - x^{(i)} - 1) - 2\sigma_R(x - x^{(i)}) + \sigma_R(x - x^{(i)} + 1) \Big) \\
&= \begin{cases} y^{(i)} - 4 & \text{if } x = x^{(i)}, \\ x & \text{if } |x - x^{(i)}| \geq 1. \end{cases}
\end{aligned}
$$

Define $\boldsymbol{f}_{n+1}(x) = x + 0\sigma_R(0 \cdot x + 0) + 4$. Let $\hat{\boldsymbol{f}} = \boldsymbol{f}_{n+1} \circ \boldsymbol{f}_n \circ \boldsymbol{f}_{n-1} \circ \cdots \circ \boldsymbol{f}_1 \in \mathcal{R}(3, n+1, \mathbb{R} \to \mathbb{R})$. Since $y^{(i)} \in [0, 1]$, we can verify that

$$
\hat{\boldsymbol{f}}(x^{(i)}) = y^{(i)} \quad \text{for any } i \in [n].
$$

By applying Lemma F.1 to $\hat{\boldsymbol{f}}$, there exists a feed-forward neural network $\boldsymbol{f}' \in \mathcal{FF}(5, n+1, \mathbb{R} \to \mathbb{R})$ such that $\boldsymbol{f}'(x^{(i)}) = y^{(i)}$ for any $i \in [n]$. Let $\boldsymbol{f}$ denote the composition of $\boldsymbol{f}'$ and $\boldsymbol{F}$. It is clear that $\boldsymbol{f} \in \mathcal{FF}(O(1), O(n)), \mathbb{R}^d \to \mathbb{R}$, which completes the proof. $\qquad \square$

Note that in order to obtain the similar trade-off between prompt length and the number of intermediate steps in Section 3, we need to prove that there exists a ReLU neural network with depth $\tilde{O}(\sqrt{n})$ that can memorize $n$ data points with real labels. However, we only derive a weaker version with $O(n)$ depth, which is still an open problem for future research. We provide similar Theorems of prompt tuning Transformers for data memorization with real labels without proof since the proofs are basically the same as that of Theorem 3.1.

**Theorem G.1.** *Fix any $d \in \mathbb{N}^+$. There exists a composition of three Transformers $\boldsymbol{T} = \boldsymbol{T}^{(3)} \circ \boldsymbol{T}^{(2)} \circ \boldsymbol{T}^{(1)}$ with $\boldsymbol{T}^{(i)} \in \mathcal{T}(D_{in}^{(i)}, D_{out}^{(i)}, D_{hid}^{(i)}, H^{(i)}, L^{(i)})$, such that for any sequence of input-output pairs $(\boldsymbol{X}^{(1)}, \boldsymbol{y}^{(1)}), \cdots, (\boldsymbol{X}^{(n)}, \boldsymbol{y}^{(n)}) \in \mathbb{R}^{d \times N} \times [C]^{1 \times N}$ satisfying Assumption 3.1 and $N > 1$, there exist prompts $\boldsymbol{P}^{(i)} \in \mathbb{R}^{D_{in}^{(i)} \times M^{(i)}}$ and $K_1^{(i)}, K_2^{(i)} \in \mathbb{N}$ such that*

$$
\hat{\boldsymbol{T}}_{\boldsymbol{P}^{(3)}, K_1^{(3)}, K_2^{(3)}}^{(3)} \circ \hat{\boldsymbol{T}}_{\boldsymbol{P}^{(2)}, K_1^{(2)}, K_2^{(2)}}^{(2)} \circ \hat{\boldsymbol{T}}_{\boldsymbol{P}^{(1)}, K_1^{(1)}, K_2^{(1)}}^{(1)}(\boldsymbol{X}^{(i)}) = \boldsymbol{y}^{(i)} \quad \text{for any } i \in [n],
$$

*where $D_{in}^{(i)} = O(d)$, $D_{out}^{(i)} = O(1)$, $D_{hid}^{(i)} = O(D_{in})$, $H^{(i)} = O(1)$, $L^{(i)} = O(1)$, and $M^{(i)} = O(n)$, $K_1^{(i)}, K_2^{(i)} = O(N \cdot n)$, for $i = 1, 2, 3$.*

**Theorem G.2.** *Fix any $d, n \in \mathbb{N}^+$. There exists a composition of three Transformers $\boldsymbol{T} = \boldsymbol{T}^{(3)} \circ \boldsymbol{T}^{(2)} \circ \boldsymbol{T}^{(1)}$ with $\boldsymbol{T}^{(i)} \in \mathcal{T}(D_{in}^{(i)}, D_{out}^{(i)}, D_{hid}^{(i)}, H^{(i)}, L^{(i)})$, such that for any sequence of input-output pairs $(\boldsymbol{X}^{(1)}, \boldsymbol{y}^{(1)}), \cdots, (\boldsymbol{X}^{(n)}, \boldsymbol{y}^{(n)}) \in \mathbb{R}^{d \times N} \times [C]^{1 \times N}$ satisfying Assumption 3.1 and $N > 1$, there exist prompts $\boldsymbol{P}^{(i)} \in \mathbb{R}^{D_{in}^{(i)} \times M^{(i)}}$ and $K_1^{(i)}, K_2^{(i)} \in \mathbb{N}$ such that*

$$
\hat{\boldsymbol{T}}_{\boldsymbol{P}^{(3)}, K_1^{(3)}, K_2^{(3)}}^{(3)} \circ \hat{\boldsymbol{T}}_{\boldsymbol{P}^{(2)}, K_1^{(2)}, K_2^{(2)}}^{(2)} \circ \hat{\boldsymbol{T}}_{\boldsymbol{P}^{(1)}, K_1^{(1)}, K_2^{(1)}}^{(1)}(\boldsymbol{X}^{(i)}) = \boldsymbol{y}^{(i)} \quad \text{for any } i \in [n],
$$

*where $D_{in}^{(i)} = O(n \vee d)$, $D_{out}^{(i)} = O(1)$, $D_{hid}^{(i)} = O(D_{in})$, $H^{(i)} = O(1)$, $L^{(i)} = O(1)$, and $M^{(i)} = O(n)$, $K_1^{(i)}, K_2^{(i)} = O(N)$, for $i = 1, 2, 3$.*

## H Low-Rank Bias of Prompt Tuning

Deep neural networks, despite their enormous parameter counts, often display an implicit preference for learning functions of low effective complexity. One notable manifestation of this phenomenon is the low-rank biasthe empirical tendency of neural networks to produce representations, weight matrices, or inputoutput mappings that are approximately low-rank, even in the absence of explicit rank constraints. This bias reflects a form of structural simplicity that naturally arises from standard optimization and regularization procedures such as stochastic gradient descent, weight decay, and early stopping [Gunasekar et al., 2018, Huh et al., 2021].

In the case of linear models, theoretical analyses have shown that gradient descent implicitly minimizes the nuclear norm, thereby converging to low-rank solutions. This phenomenon extends

to deep nonlinear networks, where empirical studies reveal that the singular value spectra of trained weight matrices and activation covariances decay rapidlysuggesting that most of the representational variance is captured by a small number of dominant modes. Similar low-rank patterns also emerge in self-attention mechanisms, where effective attention maps are often confined to low-dimensional subspaces.

From a theoretical perspective, the low-rank bias can be viewed as a form of implicit regularization, whereby the dynamics of stochastic gradient descent favor smoother and more compressive mappings. This implicit regularization provides a partial explanation for the strong generalization ability of overparameterized neural networks: low-rank solutions reduce model complexity and improve robustness to input perturbations. However, this same bias can also limit expressivitytasks requiring high-rank or highly entangled feature interactions may be more difficult to capture under such constraints.

An equally important factor lies in the geometry of the data distribution itself. Real-world data—such as images, language, audio, and other structured signals—typically lie on or near a low-dimensional manifold embedded within a high-dimensional ambient space. This manifold hypothesis implies that, although input representations are high-dimensional, the intrinsic degrees of freedom governing them are much smaller. Neural networks trained via gradient-based optimization may thus naturally adapt to this underlying manifold structure, leading to the emergence of low-rank patterns in their learned parameters and representations.

In the following, we prove that under certain assumption, prompt tuning does can capture the low-rank structure in the dataset, which leads to a reduction in prompt length. The prompt length does not depend on the number of data points to be memorzied, while mainly determined by the number of classes.

**Theorem H.1.** *Fix any $n, d \in \mathbb{N}^+$. There exists a Transformer $\boldsymbol{T} \in \mathcal{T}(D_{in}, D_{out}, D_{hid}, H, L)$ such that for any sequence of input-output pairs $(\boldsymbol{x}^{(1)}, y^{(1)}), \cdots, (\boldsymbol{x}^{(n)}, y^{(n)}) \in \mathbb{R}^d \times [C]$ satisfying Assumption 3.1, and we assume that $y^{(1)}, \cdots, y^{(n)}$ have at most $m$ different values with $n = m \cdot k$. Without loss of generality, we denote the $m$ different values as $v^{(1)}, \cdots, v^{(m)}$. Define the set $\mathbb{Y}^{(i)} := \{\boldsymbol{x}^{(j)} : y^{(j)} = v^{(i)}\}$, which contains all the inputs whose labels are the same. We assume that for any $i \in [m]$, there exsits a vector $\boldsymbol{z}^{(i)} \in \mathbb{R}^d$ such that for any $\boldsymbol{x} \in \mathbb{Y}^{(i)}$, we have $\boldsymbol{x} = c \cdot \boldsymbol{z}^{(i)}$ for some distinct $c \in \mathbb{R}_{>0}$. Then, there exist a prompt $\boldsymbol{P} \in \mathbb{R}^{D_{in} \times M}$ and $K \in \mathbb{N}^+$ such that*

$$\hat{\boldsymbol{T}}_{\boldsymbol{P}, K, K}(\boldsymbol{x}^{(i)}) = y^{(i)} \quad \text{for any } i \in [n],$$

*where $D_{in} = O(n)$, $D_{out} = O(1)$, $D_{hid} = O(D_{in})$, $H = O(1)$, $L = O(1)$, and $M = O(m)$, $K = O(1)$.*

*Proof of Theorem H.1.* According to our assumption, we know that the whole dataset $(\boldsymbol{x}^{(1)}, y^{(1)}), \cdots, (\boldsymbol{x}^{(n)}, y^{(n)})$ can be divided into $m$ subsets. in each of which, labels are the same. Without loss of generality, we assume that all the $\boldsymbol{x}^{(i)}$ are listed in order in terms of their labels and we let $\boldsymbol{x}^{(i)} = c^{(i)} \boldsymbol{z}^{(i)}$. According to Lemma E.5, there exists $\boldsymbol{v} \in \mathbb{R}^d$ such that $\boldsymbol{v}^\top \boldsymbol{x}^{(i)}$ are distinct. Let $K > 0$ be determined later. We define

$$\boldsymbol{W}_i^{(1)} = K \begin{pmatrix} 1 \\ 1 \\ 1 \end{pmatrix} \boldsymbol{v}^\top, \quad \boldsymbol{b}_i^{(1)} = \begin{pmatrix} -K\boldsymbol{v}^\top \boldsymbol{x}^{(i)} - c^{(i)} \\ -K\boldsymbol{v}^\top \boldsymbol{x}^{(i)} \\ -K\boldsymbol{v}^\top \boldsymbol{x}^{(i)} + c^{(i)} \end{pmatrix}, \quad \boldsymbol{W}_i^{(2)} = \frac{1}{c^{(i)}} \boldsymbol{y}^{(i)}(1, -2, 1), \quad \boldsymbol{b}_i^{(2)} = 0.$$

where $\boldsymbol{W}_i^{(1)} \in \mathbb{R}^{3 \times d}$, $\boldsymbol{b}_i^{(1)} \in \mathbb{R}^3$, $\boldsymbol{W}_i^{(2)} \in \mathbb{R}^{1 \times 3}$, and $\boldsymbol{b}_i^{(2)} \in \mathbb{R}$.

It is straightforward to verify that

$$\boldsymbol{W}_i^{(2)} \sigma_R(\boldsymbol{W}_i^{(1)} \boldsymbol{x} + \boldsymbol{b}_i^{(1)}) + \boldsymbol{b}_i^{(2)}$$
$$= \frac{1}{c^{(i)}} \boldsymbol{y}^{(i)} \left( \sigma_R(K\boldsymbol{v}^\top(\boldsymbol{x} - \boldsymbol{x}^{(i)}) - c^{(i)}) - 2\sigma_R(K\boldsymbol{v}^\top(\boldsymbol{x} - \boldsymbol{x}^{(i)})) + \sigma_R(K\boldsymbol{v}^\top(\boldsymbol{x} - \boldsymbol{x}^{(i)}) + c^{(i)}) \right)$$
$$= \frac{1}{c^{(i)}} \boldsymbol{y}^{(i)} I_i(\boldsymbol{x}),$$

where $I_i(\boldsymbol{x})$ satisfies $I_i(\boldsymbol{x}^{(i)}) = c^{(i)}$ and $I_i(\boldsymbol{x}) = 0$ if $|\boldsymbol{v}^\top(\boldsymbol{x} - \boldsymbol{x}^{(i)})| \geq |c^{(i)}|/K$. Let $c = \max\{|c^{(1)}|, \cdots, |c^{(n)}|\}$. We choose $K > \frac{2c}{\min_{i \neq j} |\boldsymbol{v}^\top(\boldsymbol{x}^{(j)} - \boldsymbol{x}^{(i)})|}$. Define

$$\boldsymbol{W}^{(1)} = \begin{pmatrix} \boldsymbol{W}_1^{(1)} \\ \vdots \\ \boldsymbol{W}_n^{(1)} \end{pmatrix}, \quad \boldsymbol{b}^{(1)} = \begin{pmatrix} \boldsymbol{b}_1^{(1)} \\ \vdots \\ \boldsymbol{b}_n^{(1)} \end{pmatrix}, \quad \boldsymbol{W}^{(2)} = (\boldsymbol{W}_1^{(2)}, \cdots, \boldsymbol{W}_n^{(2)}), \quad \boldsymbol{b}^{(2)} = 0,$$

where $\boldsymbol{W}^{(1)} \in \mathbb{R}^{3n \times d}$, $\boldsymbol{b}^{(1)} \in \mathbb{R}^{3n}$, $\boldsymbol{W}^{(2)} \in \mathbb{R}^{1 \times 3n}$, $\boldsymbol{b}^{(2)} \in \mathbb{R}$. Let
$$\boldsymbol{f}(\boldsymbol{x}) = \boldsymbol{W}^{(2)} \sigma_R(\boldsymbol{W}^{(1)} \boldsymbol{x} + \boldsymbol{b}^{(1)}) + \boldsymbol{b}^{(2)}$$
$$= \sum_{i=1}^n \frac{1}{c^{(i)}} \boldsymbol{y}^{(i)} I_i(\boldsymbol{x}).$$

It is direct to verify that $\boldsymbol{f}(\boldsymbol{x}^{(i)}) = y^{(i)}$ and $\boldsymbol{f} \in \mathcal{FF}(O(n), O(1), \mathbb{R}^d \to \mathbb{R})$ by using the fact
$$\sigma_R(-c^{(i)}) + \sigma_R(c^{(i)}) = c^{(i)},$$
$$|\boldsymbol{v}^\top(\boldsymbol{x} - \boldsymbol{x}^{(i)})| \geq \min_{i \neq j} |\boldsymbol{v}^\top(\boldsymbol{x}^{(j)} - \boldsymbol{x}^{(i)})| \geq \frac{|c^{(i)}|}{2c} \cdot \min_{i \neq j} |\boldsymbol{v}^\top(\boldsymbol{x}^{(j)} - \boldsymbol{x}^{(i)})| = \frac{|c^{(i)}|}{K}.$$

As for the ranks, we point out that
$$\text{rank}([\boldsymbol{W}^{(1)}, \boldsymbol{b}^{(1)}]) \leq 3m,$$
$$\text{rank}([\boldsymbol{W}^{(2)}, \boldsymbol{b}^{(2)}]) = 1.$$

According to Lemma E.1 and Lemma 3.1, there exsits a Transformer $\boldsymbol{T} \in \mathcal{T}(D_{in}, D_{out}, D_{hid}, H, L)$, prompt $\boldsymbol{P} \in \mathbb{R}^{D_{in} \times M}$, and $K \in \mathbb{N}^+$ such that
$$\hat{\boldsymbol{T}}_{\boldsymbol{P},K,K}(\boldsymbol{x}^{(i)}) = y^{(i)},$$
where $D_{in} = O(n)$, $D_{out} = O(1)$, $D_{hid} = O(D_{in})$, $H = O(1)$, $L = O(1)$, and $M = O(m)$, $K = O(1)$. The proof is completed. $\qquad\square$

# I Experimental Details

## I.1 Setup Details

All the experiments are conducted on one NVIDIA T4 GPU. Our code is based on standard PyTorch modules.

**Figure 1** We randomly sample 1000 samples from SST-2 dataset [Socher et al., 2013], which are truncated to a length of 8. The length of the prompts prepended to the inputs is also 8. Number of training epochs is 1000, leanring rate is 0.005. Optimizer is AdamW [Loshchilov and Hutter, 2017]. We use the Roberta-base (12 heads and 12 layers) implementation of Hugginface [Wolf et al., 2019]. We plot the average attention patterns over all the training samples of heads in the 10-th layer.

**Table 1** We randomly sample $1600, 2500$, and $3600$ data points from IMDb dataset [Maas et al., 2011] and the corresponding prompt length is set to be $40, 50$, and $60$, which is roughly the square root of the dataset size. Number of training epochs is 100, learning rate is 0.001. Optimizer is AdamW. We use a two-layer hand-crafted Transformer architecture, in which the activation function in each self-attention layer can be ReLU or Softmax, number of heads is 8 and hidden dimension is 4 times the embedding size 512.

**Table 2** We randomly sample 2000 input sequential data $\boldsymbol{X}^{(i)} \in \mathbb{R}^{16 \times 8}$ from distribution $N(-1, 4)$. Then, we initialize a ReLU feed-forward neural network with 8 layers and width 32 following three strategies: default initialization strategy in PyTorch, replacing $[\boldsymbol{W}, \boldsymbol{b}]$ in each layer by a rank-1 matrix, and initializing low-rank $[\boldsymbol{W}, \boldsymbol{b}]$ together with a rank-1 embedding layer. To achieve a low-rank structure, we use the fact that $rank(AB) \leq \min\{rank(A), rank(B)\}$. Number of training epochs is 100, learning rate is 0.001. Optimizer is AdamW. The backbone is a one-layer one-head Transformer with ReLU activation, embedding size 128, hidden dimension $4 \cdot 128$. Prompt length is set to be $10, 20, 30$ and $40$.

**Table 3** We randomly sample 1600 data points from IMDb dataset the set the prompt length to be 40. The backbone is a two-layer ReLU Transformer with random word embeddings. Number of training epochs is 100, learning rate is 0.001. Optimizer is AdamW. To initialize low-norm FFN, we utilize the SVD of $\boldsymbol{W}$ and modify its spetral norm. To initialize low-rank $\boldsymbol{W}_V$, $\boldsymbol{W}_K$, $\boldsymbol{W}_Q$, we set them to be a multiplication of two low-rank matrices.

## I.2 Additional Experimental Results

In this section, we present some additional experimental results.

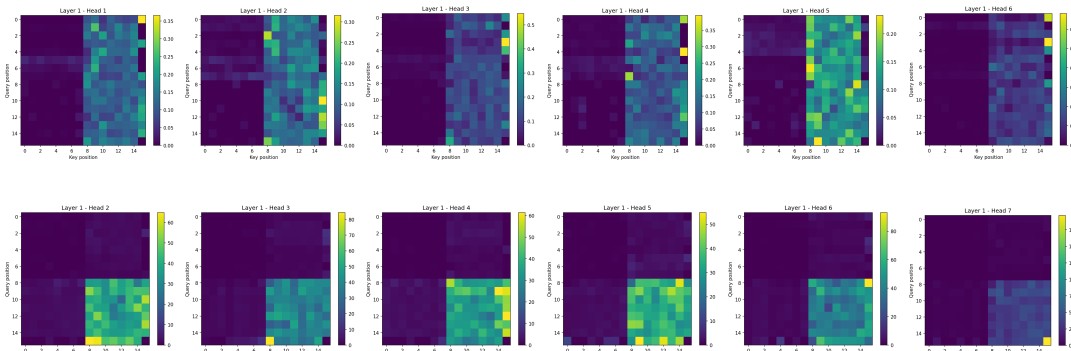

Figure 2: Attention patterns from head 1 to head 6 of random ReLU, Softmax Transformer averaged over 100 samples from IMDb dataset, using word embeddings from T5-small. The input sequence length is 16 where the first 8 tokens are prompt tokens and the remaining 8 are data tokens. The first row displays attention patterns of random Softmax Transformer, and the second row corresponds to the ReLU Transformer. Due to the normalization effect in Softmax function, the attention weights on data tokens in the random Softmax transformer are diluted by the presence of prompt tokens. In contrast, the random ReLU Transformer can assign significantly larger attention weights to data tokens.

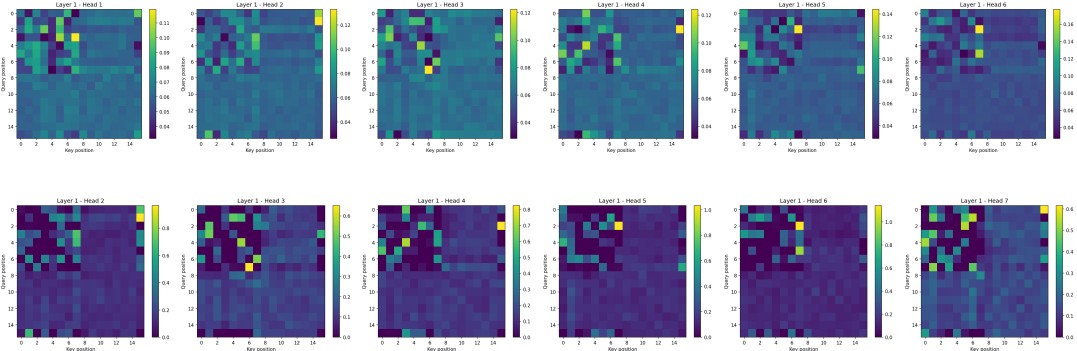

Figure 3: Attention patterns from head 1 to head 6 of random ReLU, Softmax Transformer averaged over 100 samples from IMDb dataset, using random word embeddings. The input sequence length is 16 where the first 8 tokens are prompt tokens and the remaining 8 are data tokens. The first row displays attention patterns of random Softmax Transformer, and the second row corresponds to the ReLU Transformer. Both Softmax and ReLU Transformer tend to assign equal attention weights to prompt tokens and data tokens.

