# OpenReview forum: "Prompt Tuning Transformers for Data Memorization"
_NeurIPS.cc/2025/Conference — NeurIPS 2025 poster_

### Official Review · Reviewer_xVXV · 2025-06-03

**Clarity:** 1
**Significance:** 3
**Originality:** 3
**Rating:** 3
**Confidence:** 2

**Summary:**

This paper studies the role that prompts (and prompt tuning) plays in the expressive capacity of transformers. The main paper is fully theoretical with results that outline how to simulate ReLU networks with transformers. They use this framework to show that a constant size transformer can memorize any N sequential data points of length n with prompts whose lengths are $O(sqrt(nN))$ in additional to other theoretical points.

**Questions:**

- In line 373 you write the prompt length is $O(nN)$ but in theorem 4.1 you claim it is $O(sqrt(nN))$. Is this a typo?
- Please see additional questions in the "Strengths and Weaknesses" section under "Quality" and "Clarity" for additional questions.

**Ethical Concerns:**

["NO or VERY MINOR ethics concerns only"]

**Final Justification:**

As detailed in my post-rebuttal discussion, I will raise my score to a 3 for two reasons:
- As the paper currently stands, and by the reviewing guidelines, the results in this paper are not easy to follow/reproducible without the suggested reorganization provided by the authors. The concern w.r.t. paper organization of both theoretical/emprical results which were shared by myself and reviewer cRMN would require a substantial reorganization ensure this paper is both understandable/reproducible. At the time of submission, the clarity and organization of the paper was below par.
- I outline some raised concerns during rebuttal below. The theme here is that in practice no one "tunes" their prompts when querying their language model. Instead, practically, people study "memorization" (achieving zero loss on a model input) by iteratively querying language models on large training corpuses [1]. The authors have written that data memorization is a problem pertaining to privacy, but study a randomly initialized model's ability to "memorize" prompts: this is a setting where there is no privacy concern. There is a clear disconnect between what is being studied, and the practical utility of it in informing the true limits of model's ability to memorize a prompt. If instead, the framing was around studying the relationship between model initialization schemes as they pertain to the expressive power of prompts, I think that would be a more straightforward/well-motivated story given the current experimental setup.
  -  As the authors point out "We admit that prompt tuning may not be necessarily designed to implement the simulation of ReLU feed-forward neural networks, some other potential effects will be studied in the future"
  -  reviewer QVYe points out "It is unclear what, if any, practical takeaway there is here for those who are still using prompt tuning today"
  -  reviewer URsx points out "Using prompt tuning as a means to memorize arbitrary data seems somewhat unnatural. It would strengthen the paper to include a discussion of real-world scenarios where such a setup is relevant, to better motivate the problem."

**Limitations:**

Concisely compiling the limitations in a single subsection in the conclusion would be helpful for the reader.

**Paper Formatting Concerns:**

- line 957 type: dose --> does

**Quality:**

2

**Strengths And Weaknesses:**

**Quality:**

- I notice that one of the main claims in the abstract is line 11: "Our findings, supported by experiments,...", but no where in the main paper are experiments. The experiments are included in the appendix, but I find it challenging to know which experiments are mean to support which theoretical results in the main paper.
- I notice in the experimental description in Appendix H.1 line 956: "Our experiments do not follow Proposition 3.1 exactly in several aspects. First, we discard the Autoregressive algorithm 1 because we find it does not improve performance too much.". Then why include algorithm 1? And why not modify Proposition 3.1 to exactly match your supporting experiments?

**Clarity:**

- I found the organization of this paper a little tricky to follow since it claims to contain both theoretical and experimental results in the abstract, but the experimental results are not logically organized following the respective theoretical result section in the main text but rather sequestered to the appendix.
- I did not fully understand what the concrete relationship between section 3 and section 4 is. I understand that transformers can simulate ReLU networks. I also understand that prompts can be structured in such a way that the input output pair exhibits a loss of 0 (memorization), but if/why is this specific to simulating ReLU networks?

**Significance:**

- The motivation for this line of work is: better understanding the role that model context (prompts) play for improving the expressive power of transformers. This style of inquiry helps shape the understanding of transformer failure modes and limits.

**Originality:**

- The scope of this paper is out of my main area of expertise, therefore after reviewing the related work, it seems the authors have built on existing frameworks to simulate ReLU networks using transformers but have introduced new theorems and lemmas.

---

> ### Author Rebuttal · Authors · 2025-07-31
>
> Thank you for taking the time to review our paper and sharing your insightful comments! We address some of your concerns below.
>
> > **W1. I notice that one of the main claims in the abstract is line 11: "Our findings, supported by experiments,...", but no where in the main paper are experiments. The experiments are included in the appendix, but I find it challenging to know which experiments are mean to support which theoretical results in the main paper.**
>
> We apologize for the organization of this paper, and this will be modified in our revised version following your comments. We provide a brief outline of our experiments here:
>
> - Table 1 and Table 2 study the Transformer simulation for deep ReLU neural networks and shallow ones, finding that deep neural networks are much easier to simulate than shallow ones. This experiment supports Proposition 3.1 in our work.
>
> - Table 3 explores the relationship between the Transformer simulation for shallow ReLU neural network and the embedding size, finding that the embedding size becomes the bottleneck rather than the prompt length when simulating shallow ReLU neural networks. This supports our Remark 3.4.
>
> - Table 4 and 5 show that the prompt length is sharply reduced when simulating ReLU neural networks with a low-rank structure in the weight matrices. Only $1/10$ prompt length is needed to achieve similar performance. This verifies our discussion in Section G.
>
> - Table 6 and 7 compare the performance of prompt tuning and training from scratch on a real-world dataset when the embedding size is chosen to be $256$, finding that prompt tuning can only achieve inferior performance. This also verifies our discussion in Section G. This supports our Theorem 4.1.
>
> - Table 8 and 9 compare the performance of prompt tuning and training from scratch on a real-world dataset when the embedding size is chosen to be $16$, finding that prompt tuning performs better than training from scratch when the embedding size is small.
>
> > **W2. I notice in the experimental description in Appendix H.1 line 956: "Our experiments do not follow Proposition 3.1 exactly in several aspects. First, we discard the Autoregressive algorithm 1 because we find it does not improve performance too much.". Then why include algorithm 1? And why not modify Proposition 3.1 to exactly match your supporting experiments?**
>
> Autoregressive algorithm is the key technique that can help us prove the data memorization ability of prompt tuning a fixed size Transformer and it shows the powerful parameter sharing property of Transformer architecture. However, we find that the computational burden is unbearable in practice and that is why we did not utilize the same setting in Proposition 3.1. It is important to note that we always use a simpler setting in our experiments than those in theories and find that the results are already satisfying. Since our theories are all constructive, meaning that gradient-based optimization methods may not always find the required weight configurations. We hope the reviewer could understand the gap between theoretical proofs and experiments is inevitable. Actually, our work provide a possible analysis of prompt tuning, which may explain its success in practice. Some insights coming from our theoretical comtributions may guide the empirical study, such as, low-rank structure can reduce the prompt length, etc. To fully understand data memorization ability of prompt tuning, other factors like the optimization dynamics have to be considered in the meantime, which is left for future research and we will include this in our limitation section.
>
> > **W3. I found the organization of this paper a little tricky to follow since it claims to contain both theoretical and experimental results in the abstract, but the experimental results are not logically organized following the respective theoretical result section in the main text but rather sequestered to the appendix.**
>
> We apologize for the confusion caused by the organization of this paper. We will polish it in our revised version, and we provide an outline of our experiments in the answer of **W1**, hoping this may help the reviewer understand our paper. Thanks for your time and patience again. Every comment and feedback will make our paper clearer and better.
>
> > **W4. I did not fully understand what the concrete relationship between section 3 and section 4 is. I understand that transformers can simulate ReLU networks. I also understand that prompts can be structured in such a way that the input output pair exhibits a loss of 0 (memorization), but if/why is this specific to simulating ReLU networks?**
>
> In our work, Section 3 explores the Transformer simulation for ReLU feed-forward neural networks, that is, given the same input, there exist proper prompts such that the Transformer can output the same output as the target ReLU feed-forward neural network. In Section 4, we study the data memorization ability of prompt-tuning Transformers. The connection between Section 3 and Section 4 is that in the analysis of data memorization, ReLU feed-forward neural networks are necessary, prompt tuning here replaces all the ReLU feed-forward neural networks. We admit that prompt tuning may not be necessarily designed to implement the simulation of ReLU feed-forward neural networks, some other potential effects will be studied in the future. We will provide a discussion about the effects of prompt tuning in the revised version.
>
> > **Q1. In line 373 you write the prompt length is $\tilde{O}(nN)$ but in theorem 4.1 you claim it is $\tilde{O}(\sqrt{nN})$ . Is this a typo?**
>
> Yes, this is indeed a typo, and the correct one is $\tilde{O}(\sqrt{nM})$. We apologize for this mistake and this will be modified in our revised version.

---

> > ### Comment · Reviewer_xVXV · 2025-08-03
> >
> > Thank you for the response, I will increase my score to a 3 for two reasons:
> >
> > 1. As the paper currently stands, and by the reviewing guidelines, the results in this paper are not easy to follow/reproducible without the suggested reorganization provided by the authors. The concern w.r.t. paper organization of both theoretical/emprical results which were shared by myself and reviewer cRMN would require a substantial reorganization ensure this paper is both understandable/reproducible. At the time of submission, the clarity and organization of the paper was below par.
> > 2. As the authors point out "We admit that prompt tuning may not be necessarily designed to implement the simulation of ReLU feed-forward neural networks, some other potential effects will be studied in the future", reviewer QVYe points out "It is unclear what, if any, practical takeaway there is here for those who are still using prompt tuning today" and, reviewer URsx points out "Using prompt tuning as a means to memorize arbitrary data seems somewhat unnatural. It would strengthen the paper to include a discussion of real-world scenarios where such a setup is relevant, to better motivate the problem." The theme here is that in practice no one "tunes" their prompts when querying their language model. Instead, practically, people study "memorization" (achieving zero loss on a model input) by iteratively querying language models on large training corpuses [1]. The authors have written that data memorization is a problem pertaining to privacy, but study a randomly initialized model's ability to "memorize" prompts: this is a setting where there is no privacy concern. There is a clear disconnect between what is being studied, and the practical utility of it in informing the true limits of model's ability to memorize a prompt. If instead, the framing was around studying the relationship between model initialization schemes as they pertain to the expressive power of prompts, I think that would be a more straightforward/well-motivated story given the current experimental setup.
> >
> > [1] Quantifying Memorization Across Neural Language Models, ICLR, 2023.

---

> > > ### Author Response · Authors · 2025-08-05
> > >
> > > Thank you for your insightful comments and for raising the score.
> > >
> > > - As for the organization of this paper, we apologize again for this. We agree with the reviewer's comment that the way we organize the theoretical and empirical results has caused some unnecessary ambiguity. We will reorganize our paper in the final version such that each theoretical theorem matches its corresponding empirical verification. Thanks for the reviewer's suggestions, which will make our paper clearer and better.
> > >
> > > - In this work, we study the data memorization ability of prompt tuning mainly to understand the expressive power of prompt tuning theoretically, which can be viewed as an extension of the Transformer approximation capacity. The issues about data privacy are just secondary, and we do not focus on that. Since we proved that prompt-tuning Transformers do have data memorization ability, this can strengthen the motivation to develop privacy-preserving prompt-tuning methods in practice. In fact, some papers have already observed this problem [1]. As the reviewer said, in our experiment setup, we utilize a randomly initialized Transformer with weights being frozen and only tune the parameters in prompts. Actually, our goal is to show that the data memorization ability of an imperfect Transformer with prompts already exists, and one can further naturally infer that a well-pretrained Transformer can have a more powerful data memorization ability with prompts. We agree with the reviewer's statement that there is no privacy concern in this setting, because this paper was not written to study the privacy problem at all.
> > >
> > > Your thoughtful feedback not only helped us address the concerns you raised but also deepened our understanding of the data memorization ability of real-world Transformer models. We are deeply grateful for your valuable insights and for raising the score.
> > >
> > > ## References
> > >
> > > [1] T. Nakai, K. Oonishi and T. Higashi, "Does Prompt-tuning Enhance Privacy in Large Language Models?," 2024 Annual Computer Security Applications Conference Workshops (ACSAC Workshops), Honolulu, HI, USA, 2024,

---

### Official Review · Reviewer_URsx · 2025-06-29

**Clarity:** 3
**Significance:** 3
**Originality:** 2
**Rating:** 4
**Confidence:** 4

**Summary:**

The paper studies the memorization capacity of transformers using prompts. It shows that a constant-size transformer with ReLU attention can memorize $N$ token-wise separable sequences of length $n$, using prompts of length $\tilde{O}(\sqrt{nN})$.

**Questions:**

Question:

- Regarding the statement *"if the target ReLU neural network presents the low-rank structure, the required prompt length is sharply reduced"*:
  - Is this an empirical observation without a formal proof?
  - Since the ReLU network is constructed from the data, what does the assumption of a "low-rank structure" imply about the underlying data to be memorized?
- It would be helpful to include a brief proof sketch in the main text to aid understanding.

- minor comments:
  - The term *Autoregressive Generalization* is somewhat unclear to me, as it is not obvious what is being generalized in this context.
  - In the abstract, it is claimed that *“a constant-size Transformer with prompts ... can memorize any $N$ input sequences of length $n$.”* However, as stated in the main text, this requires Assumption 4.1 (token-wise separability). I suggest clarifying this condition in the abstract to avoid overstatement.

Please also refer to the weaknesses. I am happy to revise my score based on the authors’ rebuttal and the discussion with other reviewers.

**Ethical Concerns:**

["NO or VERY MINOR ethics concerns only"]

**Final Justification:**

After considering the rebuttal, I have updated my score to 4. My recommendation is based on the following points:

Significance: The authors explained the reasonableness of their assumptions (e.g., ReLU attention) and discussed how their setting relates to real-world scenarios.

Technical contribution: The authors extended the results of [16] to a more general case where the width of the simulated ReLU network can be a small constant. While this is a meaningful step, the novelty remains somewhat limited.

**Limitations:**

yes

**Quality:**

3

**Strengths And Weaknesses:**

Strengths:
- The paper provides a theoretical analysis of the memorization capabilities of transformers with prompt tuning.  It shows that a constant-size transformer with ReLU attention can memorize any amount of data, as long as the prompt length grows in the order of $\tilde O(\sqrt {nN})$. The observation that the required prompt length can potentially be reduced in the presence of low-rank structure is also interesting.

Weaknesses:

- The proof technique primarily builds on two existing results: (1) the simulation of ReLU neural networks via prompt-tuning transformers [Nakada et al., 2025], and (2) the use of $\tilde{O}(\sqrt{nN})$-size deep ReLU networks to memorize $N$ labels [Kajitsuka and Sato, 2024]. As a result, the technical originality appears somewhat limited. It would be helpful if the authors could more clearly explain the core novelty of their contribution.
- Using prompt tuning as a means to memorize arbitrary data seems somewhat unnatural. It would strengthen the paper to include a discussion of real-world scenarios where such a setup is relevant, to better motivate the problem.
- The results are derived specifically for transformers with ReLU attention, rather than the more commonly used softmax attention. This limits the applicability of the findings to standard transformer architectures.

---

> ### Author Rebuttal · Authors · 2025-07-26
>
> Thank you for taking the time to review our paper and sharing your insightful comments! We address some of your concerns below.
>
> > **W1. The proof technique primarily builds on two existing results: (1) the simulation of ReLU neural networks via prompt-tuning transformers [Nakada et al., 2025], and (2) the use of $\tilde{O}(\sqrt{nN})$-size deep ReLU networks to memorize $N$ labels [Kajitsuka and Sato, 2024]. As a result, the technical originality appears somewhat limited. It would be helpful if the authors could more clearly explain the core novelty of their contribution.**
>
> Our contribution can be summarized as follows:
>
> - We theoretically verify the expressive ability of prompt tuning from the data memorization perspective. To our best knowledge, [1] was the first to study this problem. However, their results focus on $\epsilon$ error rather than zero error and they only provided the lower bound of the prompt length needed for data memorization. We are the first to give an upper bound.
>
> - We theoretically prove that there exists a prompt length and computational efficiency trade-off. Longer prompts lead to less iterations in the autoregressive algorithm.
>
> - We provide a possible explanation of why prompt tuning is efficient in terms of prompt length by connecting prompt tuning with the low-rank structure in the ReLu feed-forward neural networks that are simulated.
>
> - We extend the results in [16] to a more general case where the width of the ReLU feed-forward neural network to be simulated can be a small constant and can be smaller than the data dimension, which is not trivial. Besides,  we show that ReLU Transformers can implement contextual mapping under the prompt tuning setting.
>
> > **W2. Using prompt tuning as a means to memorize arbitrary data seems somewhat unnatural. It would strengthen the paper to include a discussion of real-world scenarios where such a setup is relevant, to better motivate the problem.**
>
> - Data Memorization is closely related to data privacy issues. If prompt tuning has the data memorization ability, this means that prompt tuning is not inherently privacy-preserving. Proper safeguards (e.g., DP, federated learning) should be applied to ensure data privacy when using prompt tuning in sensitive applications. Some empirical works have studied this problem [6,7,8].
>
> - Data Memorization studies the expressive ability of Transformers to achieve infinite error on discrete datasets, which is an extension of the finite precision approximation results provided by [2,3].
>
> - Data Memorization is a stepping stone towards quantifying the model's ability to generalize to new data [4,5].
>
>
> We will provide a detailed discussion in our revised version.
>
> > **W3. The results are derived specifically for transformers with ReLU attention, rather than the more commonly used softmax attention. This limits the applicability of the findings to standard transformer architectures.**
>
> In theoretical analysis, Softmax Transformers are notoriously difficult to deal with. Many existing works resort to the Hardmax function for mathematical simplicity, which can be approximated by the Softmax function [1, 2, 3,]. Besides, some other works utilize the uniform Transformers, in which the self-attention layers are designed to implement the column average operation [9, 10]. ReLU Transformers are also widely used in theoretical works [13,14]. Although ReLU Transformers are not the most common one in practice, some works [11,12] compared the performance of ReLU Transformers and Softmax Transformers on various tasks, finding that ReLU Transformers have the potential to surpass Softmax Transformers on certain tasks. Due to the clear mathematical property of ReLU function and its close relationship with ReLU feed-forward neural networks, we can achieve exactly zero error instead of $\varepsilon$-error in [1]. Although ReLU Transformers are mathematically simpler, it does not mean that they are more powerful. In Proposition 4.1, we prove that a single layer ReLU Transformer can not implement contextual mapping while a single layer Softmax Transformer can (see Theorem 2 in [15]). Extending our results to Softmax Transformers is a valuable attempt, which is left for our future research.
>
> > **Q1. Regarding the statement "if the target ReLU neural network presents the low-rank structure, the required prompt length is sharply reduced": Is this an empirical observation without a formal proof? Since the ReLU network is constructed from the data, what does the assumption of a "low-rank structure" imply about the underlying data to be memorized?**
>
> - Yes, it is an empirical observation, but it can also be theoretically proved. Please see section G in our work. Briefly speaking, the prompt length depends on the rank of the concatenation of the weight matrices and bias terms in each layer of the ReLU feed-forward network to be simulated (i.e., rank($[W_i,b_i]$)). If we let each $[W_i,b_i] = UT^{\top}$, where $U,T$ are rank-1 matrices, our results in Tables 4 and 5 show that $1/10$ prompt length can achieve comparable results. In fact, our results provide a possible explanation of the efficiency of prompt tuning, that is, Transformers with relatively short prompts (less than 100) already present powerful performance (see figure 3a in [17]).
>
> - Some works show that there exists a low-rank structure in the weight matrices in ReLU feed-forward neural networks trained with certain regularization [18]. Besides, some other works propose to approximate weight matrices with low-rank matrices without a decline in accuracy [19,20]. We assume that prompt tuning can implicitly capture this low-rank structure during training, which leads to the satisfying performance of short prompts. This may be a possible explanation for the success of prompt tuning. We did not prove that the ReLU feed-forward neural networks in our construction may be low-rank under some assumptions about the data to be memorized, and the implication about data is not clear up to now, but it is a very interesting direction for future research. However, existing related works are scarce. [21] study the approximation ability of swish neural networks and [22] consider orthogonally separable data from an optimization perspective. Thank the reviewer for this very insightful comment. We will further study that in the future.
>
> > **Q2. It would be helpful to include a brief proof sketch in the main text to aid understanding.**
>
> We apologize for the confusion caused by our organization of this paper. We will include a proof sketch in our revised version to make it clearer.
>
> > **Q3. The term Autoregressive Generalization is somewhat unclear to me, as it is not obvious what is being generalized in this context./ In the abstract, it is claimed that “a constant-size Transformer with prompts ... can memorize any
>  input sequences of length.” However, as stated in the main text, this requires Assumption 4.1 (token-wise separability). I suggest clarifying this condition in the abstract to avoid overstatement.**
>
> In fact, "Autoregressive Generalization" should be written as "Autoregressive Generation". We will correct this typo in our revised version. Since we consider discrete finite datasets, the token-wise separability is naturally satisfied, and this is a standard assumption in existing works [10,15]. This assumption does not impose extra limitations on the data but only provides a convenience for mathematics.
>
> ## References
>
>  [1] Hu, Jerry Yao-Chieh, et al. Fundamental limits of prompt tuning transformers: Universality, capacity and efficiency.
>
> [2] Chulhee Yun, et al. Are transformers universal approximators of sequence-to-sequence functions?
>
> [3] Wang, Yihan, et al. Universality and limitations of prompt tuning.
>
> [4] Chiyuan Zhang, et al. Understanding deep learning requires rethinking generalization.
>
> [5] Preetum Nakkiran, et al. Deep double descent: where bigger models and more data hurt.
>
> [6] Xie, Shangyu, et al. Does prompt-tuning language model ensure privacy?.
>
> [7] Li, Yansong, et al. Privacy-preserving prompt tuning for large language model services.
>
> [8] Hong, Junyuan, et al. Dp-opt: Make large language model your privacy-preserving prompt engineer.
>
> [9] Jiao, Yuling, et al. Transformers Can Overcome the Curse of Dimensionality: A Theoretical Study from an Approximation Perspective.
>
> [10] Kajitsuka, Tokio, and Issei Sato. On the optimal memorization capacity of transformers.
>
> [11] Shen, Kai, et al. A study on relu and softmax in transformer.
>
> [12] Wortsman, Mitchell, et al. Replacing softmax with relu in vision transformers.
>
> [13] Havrilla, Alexander, et al. Understanding scaling laws with statistical and approximation theory for transformer neural networks on intrinsically low-dimensional data.
>
> [14] Bai, Yu, et al. Transformers as statisticians: Provable in-context learning with in-context algorithm selection.
>
> [15] Kajitsuka, Tokio, and Issei Sato. Are transformers with one layer self-attention using low-rank weight matrices universal approximators?.
>
> [16] Nakada, Ryumei, et al. A Theoretical Framework for Prompt Engineering: Approximating Smooth Functions with Transformer Prompts.
>
> [17] Lester, Brian, Rami Al-Rfou, and Noah Constant. The power of scale for parameter-efficient prompt tuning.
>
> [18] Kuzborskij, et al. Low-rank bias, weight decay, and model merging in neural networks.
>
> [19] Jian Xue, et al. Restructuring of deep neural network acoustic models with
> singular value decomposition.
>
> [20] Xiyu Yu, et al. On compressing deep models by
> low rank and sparse decomposition.
>
> [21] Zimeng Li, et all.Approximation to Smooth Functions by Low-Rank Swish Networks.
>
> [22] Bui Thi Mai, et al. The inductive bias of relu networks on orthogonally separable data.

---

> > ### Comment · Reviewer_URsx · 2025-08-04
> >
> > Thank you for the detailed rebuttal. I will raise my score to 4. Just a thought for improving clarity: since you mentioned that the statement “low-rank structure reduces the required prompt length” can be theoretically proved, why not present it formally as a theorem in the paper?

---

> > > ### Author Response · Authors · 2025-08-05
> > >
> > > Your thoughtful feedback not only helped us address the concerns you raised but also deepened our understanding of the data memorization ability of real-world Transformer models. We are deeply grateful for your valuable insights and for giving us the opportunity to refine and strengthen our work. We will provide a formal theorem in our final version to prove that the low-rank structure can reduce the prompt length, following the reviewer's suggestion.
> > >
> > > Thank you once again for your insightful comments and for raising the score.

---

### Official Review · Reviewer_cRMN · 2025-07-03

**Clarity:** 2
**Significance:** 2
**Originality:** 2
**Rating:** 4
**Confidence:** 2

**Summary:**

This paper investigates how much data a constant-size Transformer can memorize when it is adapted only through prompt tuning. Building on and generalizing earlier work, the authors prove that prompts of length $\tilde{O}(\sqrt{nN})$ are sufficient for a Transformer to memorize $N$ sequences of length $n$, even when the simulated ReLU network has only constant width. They also provide a theoretical trade-off between prompt length and the number of autoregressive steps required, and they offer appendix-level experiments suggesting that low-rank structure in the target network can further shorten prompts.

**Questions:**

1. A compact table showing memorization accuracy versus prompt length—and a simple plot of the length–efficiency trade-off—would greatly strengthen the empirical link to your theory.

2. Could you discuss concrete datasets or tasks where token-wise separateness and consistent labeling approximately hold, and comment on what happens when they fail?

3. A small table contrasting your new upper bounds with those of Kajitsuka & Sato (2024), Hu et al. (2024), and Nakada et al. (2025) would help readers quantify the improvement across different regimes.

4. Why does accuracy peak at prompt length 50 but drop at 60 as observed in Table 8? Please clarify why a longer prompt does not monotonically improve accuracy. A short analysis or ablation would make the empirical trend more convincing.

5. On lines 208–209 the displayed equation should end with a comma, and the following where should start with a lowercase “w” because it continues the same sentence.

**Ethical Concerns:**

["NO or VERY MINOR ethics concerns only"]

**Final Justification:**

The paper gives a clear theory result by showing the first upper bound on prompt length for constant-size Transformers and explaining the trade-off between prompt length and efficiency. As reviewer xVXV and I both said, the paper needs to be reorganized to make the theory and experiments easier to follow.

**Limitations:**

Yes.

**Paper Formatting Concerns:**

No.

**Quality:**

2

**Strengths And Weaknesses:**

**Strengths**

1. The paper improves on prior work by providing the first explicit upper bound on prompt length for sequence-to-sequence memorization and by establishing a clear trade-off between prompt length and efficiency. These contributions extend Nakada et al. (2025) to constant-width networks and move beyond earlier works that offered only exponential lower bounds or relied on wide feed-forward layers.

2. Rigorous and well-documented analysis.
Formal definitions, lemmas, and proofs are presented with care, and the architecture of the prompt construction is spelled out in detail. A thorough Related-Work section situates the contribution within both theoretical and empirical literature, helping readers see exactly how this work fits into the evolving prompt-tuning landscape.

3. By showing when large embedding dimensions are unavoidable and how low-rank structure can shrink prompt length, the paper offers concrete guidance for designing prompt-tuning strategies. The results also strengthen the argument that fixed, pretrained Transformers can solve new tasks through prompts alone, a point of practical relevance for resource-constrained adaptation.

**Weaknesses**

1. All substantive experiments are relegated to the appendix; the body contains no figures, tables, or quantitative results. This omission makes it hard for readers to gauge how well the theory matches practice and does not meet common expectations for reproducibility.

2. Theoretical guarantees hinge on strong assumptions such as token-wise separateness and consistent labeling, yet the manuscript offers no empirical check on how often these assumptions hold or how performance degrades when they do not. Consequently, the practical impact of the results remains unclear.

3. Dense notation, many nested definitions, and the absence of illustrative diagrams or worked examples make the paper hard to follow for non-experts. Moreover, the intricate prompt design is not accompanied by a step-by-step example, which would help readers replicate the construction.

---

> ### Author Rebuttal · Authors · 2025-07-26
>
> We thank the reviewer for their thorough response! We address some of your concerns below.
>
> > **W1 and Q1. All substantive experiments are relegated to the appendix; the body contains no figures, tables, or quantitative results. This omission makes it hard for readers to gauge how well the theory matches practice and does not meet common expectations for reproducibility. /  compact table showing memorization accuracy versus prompt length—and a simple plot of the length–efficiency trade-off—would greatly strengthen the empirical link to your theory.**
>
> We apologize for the organization of this paper, and this will be modified in our revised version following your comments. We provide a brief outline of our experiments here:
>
> - Table 1 and Table 2 study the Transformer simulation for deep ReLU neural networks and shallow ones, finding that deep neural networks are much easier to simulate than shallow ones. This experiment supports Proposition 3.1 in our work.
>
> - Table 3 explores the relationship between the Transformer simulation for shallow ReLU neural network and the embedding size, finding that the embedding size becomes the bottleneck rather than the prompt length when simulating shallow ReLU neural networks. This supports our Remark 3.4.
>
> - Table 4 and 5 show that the prompt length is sharply reduced when simulating ReLU neural networks with a low-rank structure in the weight matrices. Only $1/10$ prompt length is needed to achieve similar performance. This verifies our discussion in Section G.
>
> - Table 6 and 7 compare the performance of prompt tuning and training from scratch on a real-world dataset when the embedding size is chosen to be $256$, finding that prompt tuning can only achieve inferior performance. This also verifies our discussion in Section G. This supports our Theorem 4.1.
>
> - Table 8 and 9 compare the performance of prompt tuning and training from scratch on a real-world dataset when the embedding size is chosen to be $16$, finding that prompt tuning performs better than training from scratch when the embedding size is small.
>
> Here, we provide the table showing the accuracy versus prompt length. We will reorganize our paper and provide more experimental results in the main text following the reviewer's suggestions.
>
> | Embedding size |   Prompt length | Acc |
> |--------------------|-------------------|------|
> |256| 0| 1.0000|
> | 256| 40| 0.9800|
> | 256| 50| 0.9756|
> | 256| 60| 0.9731|
> |16| 0| 0.8281|
> | 16| 40| 0.9206|
> | 16| 50| 0.9381|
> | 16| 60| 0.9350|
>
>
>
> > **W2 and Q2. Could you discuss concrete datasets or tasks where token-wise separateness and consistent labeling approximately hold, and comment on what happens when they fail?**
>
> Firstly, we would like to note that the $(r,\delta)$-separateness (see Assumption 4.1) is a standard assumption in the study of data memorization [9,10,11]. Since in the task of data memorization, we consider a discrete dataset with finite samples, and any discrete dataset is naturally $(r,\delta)$-separated, so this assumption is loose. In terms of the "consistently labeled" assumption in Theorem 4.1 and 4.2, we note that this assumption is necessary since a Transformer is a permutation equivariant function (please refer to claim 1 in [4] for detailed proof). We say a function $f: \mathbb{R}^{d\times n}\rightarrow \mathbb{R}^{d\times n}$ is permutation equivalent if for any permutation matrix $P$, we have $f(XP) = f(X)P$. In other words, if we permute the columns of the input $X$, the columns of the output $f(X)$ are permuted in the same way. Let $(Z^{(1)},y^{(1)})$ and $(Z^{(2)},y^{(2)})$ be two input-label pairs and $Z^{(1)}=Z^{(2)}$ up to permutations. Suppose that $Z_{:,i}^{(1)}=Z_{:,j}^{(2)}$ for some $i,j\in [n]$ while $y_{:,i}^{(1)}\neq y_{:,j}^{(2)}$, we can prove that there exists no Transformer $f$ such that $f(Z_{:,i}^{(1)})= y_{:,i}^{(1)}$ and $f(Z_{:,j}^{(2)}) = y_{:,j}^{(2)}$, which means that any Transformer fails to memorize $(Z^{(1)},y^{(1)})$ and $(Z^{(2)},y^{(2)})$. In [10], they define a special positional embedding to break the permutation equivariant limitation. It is still unclear that whether their method can be integrated into our work. We will provide a discussion about this in our final version.
>
> > **W3. Dense notation, many nested definitions, and the absence of illustrative diagrams or worked examples make the paper hard to follow for non-experts. Moreover, the intricate prompt design is not accompanied by a step-by-step example, which would help readers replicate the construction.**
>
> Thank you for your suggestions, which we believe will make our paper clearer and easier to follow. In the revised version, we will provide a toy example to show how these prompts are constructed step-by-step, and a notation table in the Appendix to facilitate understanding. Briefly speaking, the prompts we designed store the information of the ReLU feed-forward to be simulated and the position information, which decides the interaction between the prompts and data to be memorized.
>
> > **Q3. A small table contrasting your new upper bounds with those of Kajitsuka & Sato (2024), Hu et al. (2024), and Nakada et al. (2025) would help readers quantify the improvement across different regimes.**
>
> Our problem setting is different from all existing works. [1] studied the data memorization ability of Transformers with prompts. However, they results can only achieve $\varepsilon$ error instead of zero error and they assume that there exists a sequence-to-sequence continuous function $f$ such that $f(Z^{(i)}) = y^{(i)}$ for each input-label pair $(Z^{(i)},y^{(i)})$, which is a stronger assumption that ours. [2] was the first to show that a Transformer with different prompts can simulate/ exactly compute different ReLU feed-forward neural networks, and did not consider Transformers' ability to deal with sequential data. Besides, [3] studied the problem of how efficiently Transformers can memorize datapoints with a minimum number of parameters. Their results show that the number of parameters is upper bounded by $\tilde{O}(\sqrt{nN})$, which aligns with the prompt length in our work. Their work is not based on the prompt tuning technique, and the Transformers they constructed rely mostly on large feed-forward layers while only one self-attention layer is used, making it hard to isolate the role of the attention mechanism itself. Table 1 in [3] provides a detailed comparison among existing works with respect to the number of parameters that are needed to memorize $N$ datapoints. Since our work does not focus on the number of parameters, we can not find a place for our work in that table. But one thing is certain is that [1] only provides a lower bound of prompt length needed for data memorization, we are the first to prove an upper bound.
>
> > **Q4 Why does accuracy peak at prompt length 50 but drop at 60 as observed in Table 8? Please clarify why a longer prompt does not monotonically improve accuracy. A short analysis or ablation would make the empirical trend more convincing.**
>
> We provide a possible explanation here: The performance of prompt tuning depends on many aspects: model size, prompt initialization, optimization, task complexity ……. In our work, we theoretically prove that Transformers with prompts can memorize any set of datapoints if the prompts are sufficiently long. Our proof is constructive rather than prescriptive for training, meaning that standard gradient-based methods may not always efficiently find the required weight configuration so the gap between theory and practice is inevitable. Besides, long prompts may introduce extra information and disturb the optimization process. Figure 3(a) in [4] shows that longer prompts may result in a decline in performance and models with different number of parameters present different tolerance to long prompts. Similar results can be found in Figure 4 in [5] and Table 2 in [6]. In order to fully understand how prompt tuning works, it is necessary to consider other factors, like optimization.
>
> > **Q5. On lines 208–209 the displayed equation should end with a comma, and the following where should start with a lowercase “w” because it continues the same sentence.**
>
> We apologize for this typo, and this will be corrected in our revised version.
>
> ## References
>
> [1] Hu, Jerry Yao-Chieh, et al. Fundamental limits of prompt tuning transformers: Universality, capacity and efficiency.
>
> [2] Nakada, Ryumei, et al. A Theoretical Framework for Prompt Engineering: Approximating Smooth Functions with Transformer Prompts.
>
> [3] Kajitsuka, Tokio, and Issei Sato. On the optimal memorization capacity of transformers.
>
> [4] Lester, Brian, Rami Al-Rfou, and Noah Constant. The power of scale for parameter-efficient prompt tuning.
>
> [5] Li, Xiang Lisa, and Percy Liang. Prefix-tuning: Optimizing continuous prompts for generation.
>
> [6] Joon-Young Choi, Junho Kim, Jun-Hyung Park, Wing-Lam Mok, and SangKeun Lee. SMoP: Towards Efficient and Effective Prompt Tuning with Sparse Mixture-of-Prompts.

---

> > ### Comment · Reviewer_cRMN · 2025-08-05
> >
> > Thank you for the response. Your clarifications are helpful. As noted by reviewer xVXV and by me, the paper needs reorganization to make the theory and experiments clear. I will maintain my score.

---

> > > ### Author Response · Authors · 2025-08-06
> > >
> > > Your thoughtful feedback not only helped us address the concerns you raised but also deepened our understanding of the data memorization problem. We are deeply grateful for your valuable insights and for giving us the opportunity to refine and strengthen our work.
> > >
> > > Thank you once again for your constructive comments.

---

### Official Review · Reviewer_QVye · 2025-07-03

**Clarity:** 2
**Significance:** 4
**Originality:** 4
**Rating:** 5
**Confidence:** 2

**Summary:**

This theory paper studies the data memorization capabilities of the prompt tuning technique on Transformers. It builds on prior work which, under some conditions, emulates ReLU nets using transformers to finite precision. The authors extend on this work by providing novel theoretical results to narrower ReLU nets, an upper bound on the prompt length for which a constant-size transformer can memorize data, and empirical evidence that the theoretical results work in practice. They further elucidate that this prompt length can be shortened if the ReLU weights are low rank.

**Questions:**

- What practical insights can we take away from the theoretical findings here that would be useful to the average prompt tuning user?
- Can the authors elaborate more on the practical implications of the assumptions?

**Ethical Concerns:**

["NO or VERY MINOR ethics concerns only"]

**Final Justification:**

The paper proposes a novel theoretical insight on data memorization in prompt tuning, extending on past work. During the discussion period, the authors sufficiently addressed concerns, particularly those around practicality and empirical evaluations.

**Limitations:**

yes

**Quality:**

3

**Strengths And Weaknesses:**

Strengths:
- the paper marks a notable and novel advancement over prior work
- there is experimentation provided to support the theoretical results
- the authors propose an upper bound which, to my knowledge, has not been proposed before in this line of work
- the authors investigate additional cases of low-rank ReLU nets which has not been studied before in this line of work

Weaknesses:
- This introduction of this paper is written as if the reader is already familiar with prior work. This can be quite confusing and additional context should be given to motivate the work. Furthermore, a significant portion of contributions are in the appendix which should have otherwise been in the main body.
- It is unclear what, if any, practical takeaway there is here for those who are still using prompt tuning today.
- this paper makes assumptions (such as in 4.1). The extent to which these assumptions further deviate the practicality of this work is unclear and should be discussed.

While I am not an expert here, this paper reads rigorous and clearly marks an advance over prior theory, hence my tepid accept. I would strongly recommend that the authors make a case for why this work is relevant and practical outside of theory.

---

> ### Author Rebuttal · Authors · 2025-07-25
>
> Thank you for taking the time to review our paper and sharing your insightful comments. We will address your questions below.
> > W1. This introduction of this paper is written as if the reader is already familiar with prior work. This can be quite confusing and additional context should be given to motivate the work. Furthermore, a significant portion of contributions are in the appendix which should have otherwise been in the main body.
>
> Our work is closely related to two fields: Data Memorization and Prompt Tuning. Data Memorization aims to study the ability of neural networks to achieve zero loss on discrete datasets while Prompt Tuning requires designing or learning different prompts that can help Transformers adapt to downstream tasks. Data Memorization is important both from a privacy perspective [1], and as a stepping stone towards quantifying the model's ability to generalize to new data [2, 3]. Besides, to theoretically understand the superior performance of Transformers, some works also focus on their approximation ability [4]. Since data memorization is usually a key component to derive approximation results, it is reasonable to study Transformers' data memorization ability to verify their  representational capability. Prompt tuning tries to enable fixed pretrained Transformers to handle different tasks with corresponding prompts. Here, we combine Data Memorization and Prompt Tuning to provide a theoretical analysis of the success of prompt tuning.
>
> > W2 and Q2. What practical insights can we take away from the theoretical findings here that would be useful to the average prompt tuning user?
>
> - The first practical insight is that longer prompts are more informative. Since each prompt in our work stores the information of the target ReLU feed-forward neural networks, whose width and depth decide the prompt length. This means that longer prompts are more powerful because of the expressive ability of wider or deeper ReLU neural networks [5]. Some empirical studies [6,7] are consistent with our results.
> - Fixed Transformers with different prompts can adapt to different tasks. In Proposition 3.1,  the architecture and parameters of the constructed Transformer only depend on $\max(W,p)$, where $W$ represents the width of the ReLU feed-forward network that will be simulated and $p$ is the data dimension. Moreover, we show that a small constant-width ReLU feed-forward network can achieve data memorization for non-sequential data, which leads to the data memorization ability of a fixed Transformer. Here, a fixed Transformer can be viewed as a pretrained Transformer with the parameters frozen, which aligns with the real-world setting of prompt tuning.
> - Design prompts with large spectral norm. According to the definition of prompts in (2.6), $S$ is a large enough positive constant, which means that the constructed prompts have a large spectral norm. This phenomenon has been observed by [8].
>
> - It is important to note that our theoretical results are all constructive rather than prescriptive, meaning that other factors like the optimization dynamics and computational resources should also be considered in practice.
>
> > W3 and Q3. Can the authors elaborate more on the practical implications of the assumptions?
>
> Firstly, we would like to note that the $(r,\delta)$-separateness (see Assumption 4.1) is a standard assumption in the study of data memorization [9,10,11]. Since in the task of data memorization, we consider a discrete dataset with finite samples, and any discrete dataset is naturally $(r,\delta)$-separated, so this assumption is loose. In terms of the "consistently labeled" assumption in Theorem 4.1 and 4.2, we note that this assumption is necessary since a Transformer is a permutation equivariant function (please refer to claim 1 in [4] for detailed proof). We say a function $f: \mathbb{R}^{d\times n}\rightarrow \mathbb{R}^{d\times n}$ is permutation equivalent if for any permutation matrix $P$, we have $f(XP) = f(X)P$. In other words, if we permute the columns of the input $X$, the columns of the output $f(X)$ are permuted in the same way. Let $(Z^{(1)},y^{(1)})$ and $(Z^{(2)},y^{(2)})$ be two input-label pairs and $Z^{(1)}=Z^{(2)}$ up to permutations. Suppose that $Z_{:,i}^{(1)}=Z_{:,j}^{(2)}$ for some $i,j\in [n]$ while $y_{:,i}^{(1)}\neq y_{:,j}^{(2)}$, we can prove that there exists no Transformer $f$ such that $f(Z_{:,i}^{(1)})= y_{:,i}^{(1)}$ and $f(Z_{:,j}^{(2)}) = y_{:,j}^{(2)}$, which means that any Transformer fails to memorize $(Z^{(1)},y^{(1)})$ and $(Z^{(2)},y^{(2)})$. In [10], they define a special positional embedding to break the permutation equivariant limitation. We will provide a discussion about this in our final version.
>
>
>
>
> ## References
>
> [1] Carlini and Nicholas. Extracting training data from large language models.
>
> [2] Chiyuan Zhang, Samy Bengio, Moritz Hardt, Benjamin Recht, and Oriol Vinyals. Understanding deep learning requires rethinking generalization.
>
> [3] Preetum Nakkiran, Gal Kaplun, Yamini Bansal, Tristan Yang, Boaz Barak, and Ilya Sutskever. Deep double descent: where bigger models and more data hurt.
>
> [4] Yun, Chulhee, et al. Are transformers universal approximators of sequence-to-sequence functions?.
>
> [5] Lu, Jianfeng, et al. Deep network approximation for smooth functions.
>
> [6] Brown, T., Mann, B., Ryder, N., Subbiah, M., Kaplan, J. D., Dhariwal, P., Neelakantan, A., Shyam, P., Sastry, G., Askell, A., et al. Language models are few-shot learners.
>
> [7] Min, S., Lyu, X., Holtzman, A., Artetxe, M., Lewis, M., Hajishirzi, H., and Zettlemoyer, L. (2022). Rethinking the role of demonstrations: What makes in-context learning work?
>
> [8] Wang, Yihan, et al. Universality and limitations of prompt tuning.
>
> [9] Kajitsuka and Sato, Are Transformers with One Layer Self-Attention Using Low-Rank Weight Matrices Universal Approximators? ICLR 2024.
>
> [10]  Kim et al., Provable Memorization Capacity of Transformers. ICLR 2023.
>
> [11] Vardi et al., On the Optimal Memorization Capacity of ReLU Neural Networks. ICLR 2021.

---

> > ### Comment · Reviewer_QVye · 2025-08-01
> >
> > I thank the authors for their detailed response, which properly addressed my questions. I will maintain my score.

---

> > > ### Author Response · Authors · 2025-08-05
> > >
> > > Your thoughtful feedback not only helped us address the concerns you raised but also deepened our understanding of the data memorization problem. We are deeply grateful for your valuable insights and for giving us the opportunity to refine and strengthen our work.
> > >
> > > Thank you once again for your constructive comments.

---

### Note · Authors · 2025-08-12

- **Paper organization, motivation, and contribution**: Several reviewers mentioned that the organization of our work has caused unnecessary ambiguity. We apologize for this again. Since this work mainly aims to provide theoretical results, we did not focus too much on experiments. We hope all the reviewers, ACs, SACs can understand that the gap between theory and practice is hard to bridge sometimes, especially when theorems are all constructive. Actually, the real-world models are derived by gradient descent, which means that they can not always find the same weight configuration as that constructed in our theorems. Nevertheless, this does not diminish our contribution: when researchers seek to understand what kinds of computations a trained neural network can perform and what its weight matrices represent, we provide a deterministic result showing that prompt tuning is indeed capable of memorizing data points, which proves the expressive capacity of prompt tuning and establishes a foundation for other fields like data privacy. Moreover, our experiments offer unexpected empirical evidence supporting our theorems, even in scenarios where no such agreement is guaranteed.  Finally, we made an effort to connect theory with reality by giving a possible explanation of the efficiency of prompt tuning according to the observation that short prompts are already powerful, whereas long prompts may degrade the performance. We will follow all the helpful suggestions to reorganize this paper such that each theorem matches its corresponding experiments and provide more information in the main text. We believe that this is not a tricky problem, and it is easy to solve when someone points it out. Thanks for your comments again.

- **Closing gratitude:** During the rebuttal, we received many insightful comments. Reviewer OVye wondered what practical insights we can provide, which is consistent with our motivation to explain real-world phenomena with theoretical tools. Reviewer cRMN
 noticed that in our experiments, the performance declines when the prompt length is large. Reviewer URsx proposed that what kind of assumption on data points is needed to get a low-rank structure. Reviewer xVXV connects the initialization of the original models with prompt tuning.  All these gifted us a spark of hope when the days felt darkest. The reviewers‘ comments opened a door to a new world, one that holds infinity within. Now I see why this community, and AI itself, continue to thrive.

---

### Decision · Program_Chairs · 2025-09-17

**Decision:**

Accept (poster)

**Comment:**

In this work, the authors show that a single-layer randomly initialized Transformer with prompts possess competitive data memorization ability compared with models trained from scratch. The work builds on prior work to provide novel theoretical results to narrower ReLU nets, an upper bound on the prompt length for which a constant-size transformer can memorize data.

Most reviewers were positive about this work. Post rebuttal they indicated that
- The paper provides theoretical insight on data memorization in prompt tuning, extending on past work. The reviewer was satisfied iwth the answers provided by the authors during the rebuttal, addressing concerns regarding practicality and empirical evaluations.
- The paper provided the first upper bound on prompt length for constant-size Transformers and explaining the trade-off between prompt length and efficiency. The reviewer also indicated that the paper readability could be improved.
- The paper extends prior results in a meaningful, albeit limited way. The reviewer also found the practical relevance and assumptions credible post rebuttal discussion.

The less positive reviewer raised their score (to borderline reject), indicating that they would like to see the paper being reorganised to be more easy to follow. One of the more positive reviewers echoed that. I'd encourage the authors to revise the paper to make it more accessible. The reviewer also remained critical about the framing and suggested studying the relationship between model initialization schemes given the current experimental setup instead. While this a valid direction to consider, it is IMO beyond the scope of the current submission.